# The HIV capsid mimics karyopherin engagement of FG-nucleoporins

C. F. Dickson[1,2,7], S. Hertel[1,2,7], A. J. Tuckwell[1,2], N. Li[1,2], J. Ruan[3], S. C. Al-Izzi[2,4], N. Ariotti[5], E. Sierecki[1,2], Y. Gambin[1,2], R. G. Morris[2,4], G. J. Towers[6], T. Böcking[1,2] & D. A. Jacques[1,2✉]

HIV can infect non-dividing cells because the viral capsid can overcome the selective barrier of the nuclear pore complex and deliver the genome directly into the nucleus[1,2]. Remarkably, the intact HIV capsid is more than 1,000 times larger than the size limit prescribed by the diffusion barrier of the nuclear pore[3]. This barrier in the central channel of the nuclear pore is composed of intrinsically disordered nucleoporin domains enriched in phenylalanine–glycine (FG) dipeptides. Through multivalent FG interactions, cellular karyopherins and their bound cargoes solubilize in this phase to drive nucleocytoplasmic transport[4]. By performing an in vitro dissection of the nuclear pore complex, we show that a pocket on the surface of the HIV capsid similarly interacts with FG motifs from multiple nucleoporins and that this interaction licences capsids to penetrate FG-nucleoporin condensates. This karyopherin mimicry model addresses a key conceptual challenge for the role of the HIV capsid in nuclear entry and offers an explanation as to how an exogenous entity much larger than any known cellular cargo may be able to non-destructively breach the nuclear envelope.

All retroviruses establish infection by converting their single-stranded RNA into double-stranded DNA and integrating it into host chromatin in the nucleus. For most retroviruses, mitosis provides the opportunity to interact with chromosomal DNA during nuclear envelope breakdown. However, lentiviruses, which include HIVs, have evolved to infect non-dividing cells. To do so, these viruses must navigate the gatekeeper of nuclear entry, the nuclear pore complex (NPC).

The human NPC is a 110 MDa complex composed of 30 different nucleoporin proteins (Nups), which occur in stoichiometries ranging from eight to 48 copies per NPC[5]. Movement through the NPC central transport channel is restricted by size. The vast majority of proteins larger than around 40 kDa must recruit karyopherins to facilitate their nuclear transport[3]. The selectivity barrier in the central channel gating nuclear entry is formed from intrinsically disordered Nup domains that are enriched in phenylalanine–glycine (FG) motifs. These FG-repeat domains are found in around one third of Nups (FG-Nups) and contribute in excess of 5,800 individual FG motifs to a single NPC.

The ability of HIV to infect non-dividing cells maps to the capsid protein (CA)[1]. CA self-assembles into a metastable lattice to form a closed 'shell' of around 40 MDa that contains the genomic RNA and viral enzymes. Although the CA lattice was originally thought to disassemble shortly after entry into the host cell, it is now broadly accepted that its integrity is vital for cellular trafficking and reverse transcription, and for protecting the viral genome from cytoplasmic nucleic acid sensors and nucleases[6–8]. Furthermore, recent imaging studies have captured intact capsids as they transit the NPC[2] and have demonstrated intact capsids in the nucleus[9–13].

Critically, it has been unclear how the HIV capsid can overcome the selectivity barrier of the NPC when it is more than 1,000 times greater in size than the passive diffusion limit. Transporters in the karyopherin family can carry large cargoes through the NPC by specifically interacting with the FG motifs of the diffusion barrier. The efficiency of this cargo transport relies on the presence of multiple FG-binding sites on the karyopherin. The interactions are highly specific but weak and with rapid exchange kinetics, enabling the karyopherin and its bound cargo to partition into the selectivity barrier and rapidly diffuse through the NPC[4]. Cocrystal structures have shown that the HIV capsid exterior also possesses an FG-binding pocket, which recruits the cellular cofactors Sec24C[14], Nup153 (ref. 15), and cleavage and polyadenylation specificity factor subunit 6 (CPSF6)[16]. Although these proteins are structurally distinct and found in different cellular compartments, each of their interactions with the HIV capsid depends on an FG motif that buries itself into a pocket in the CA amino-terminal domain.

## HIV capsids bind FG-repeat Nups

Given that each CA molecule carries an FG-binding site, complete capsids carry more than 1,200 such sites and therefore have a high capacity to interact with proteins carrying multiple FG motifs. We hypothesized that many CA:FG-Nup interactions may occur in addition to those previously described (Supplementary Table 1) and that even weak interactions at this site have the potential to be significant owing to multivalency. We recently developed an in vitro technique based on fluorescence fluctuation spectroscopy (FFS) to screen for interactions

[1]Department of Molecular Medicine, School of Biomedical Sciences, University of New South Wales, Sydney, New South Wales, Australia. [2]EMBL Australia Node in Single Molecule Science, School of Biomedical Sciences, University of New South Wales, Sydney, New South Wales, Australia. [3]Electron Microscope Unit, Mark Wainwright Analytical Centre, University of New South Wales, Sydney, New South Wales, Australia. [4]School of Physics, University of New South Wales, Sydney, New South Wales, Australia. [5]Institute for Molecular Bioscience, University of Queensland, Brisbane, Queensland, Australia. [6]Infection and Immunity, University College London, London, UK. [7]These authors contributed equally: C. F. Dickson, S. Hertel. ✉e-mail: d.jacques@unsw.edu.au

between the HIV capsid and cellular proteins[17]. In brief, putative binders are expressed as GFP fusions in a cell-free system. After the addition of fluorescent capsid-like particles (CLPs; CA_{A204C} cross-linked assemblies labelled with Alexa Fluor 568), binding is determined by two-colour coincidence detection as individual CLPs pass through a confocal volume generating fluorescence fluctuations (for gel electrophoresis images of cell-free and recombinant proteins, see Supplementary Figs. 1 and 2; for sequences of all proteins used, see Supplementary Table 2; and for cryo-electron tomography (cryo-ET) images of all CLPs used, see Extended Data Fig. 1). Observations are made directly in the cell-free extract, which we have found to be a powerful approach for screening difficult-to-purify putative capsid binders such as the highly disordered FG-Nups. Furthermore, the use of CLPs as the binding platform has three critical advantages: (1) CLPs show a mix of cone, tube and sphere morphologies bearing both CA hexamers and pentamers, and therefore they present all possible CA interfaces; (2) the presence of a CA lattice gives high sensitivity to the measurement by physically concentrating binders on the CLP, thereby enabling detection of interactions well below the dissociation constant ($K_d$); and (3) the CLP lattice can engage binders that rely on multivalent CA contacts to make detectable capsid interactions. This assay therefore provides a platform for systematic investigation of potentially weak but specific CA:Nup interactions for all FG-Nups across the NPC.

To 'dissect' the NPC, we individually expressed GFP fusions of the ten FG-repeat NPC components (Nup42, Nup50, Nup54, Nup58, Nup62, Nup98, Nup153, Nup214, Nup358 and Pom121; each with 5–42 FG motifs), along with Nup35, Nup88 and Nup133, which do not have FG-repeat domains (for details, see Extended Data Table 1 and Supplementary Information). We also included known CA binders CPSF6, CypA and the cyclophilin domain of Nup358 (Nup358_{3044–3224}). For Nups that did not express as full-length proteins, we either truncated the transmembrane domain (Pom121_{266–1249}) or expressed only the FG-repeat domain (Nup98_{1–499}, Nup214_{1210–2090}). Nups either remained as monomeric protein (as exemplified by POM121, Fig. 1a, flat blue trace) or spontaneously oligomerized (as exemplified by Nup98, Fig. 1b, fluctuating blue trace). This latter behaviour was not unexpected, given that FG-repeat proteins are known to phase separate in vitro[18]. Binding was identified by the recording of 'mirror plots' after the addition of CLP (Fig. 1a,b and Extended Data Fig. 2, orange traces, represented as negative values for clarity), showing that capsid and Nup were colocated. The degree of coincidence is reported as the 'Nup/CLP intensity ratio'[17]. The positive controls (CPSF6, CypA and Nup358_{3044–3224}) were recruited to the CA lattice as expected (Extended Data Fig. 2), whereas the non-FG-repeat Nups (Nup35, Nup88 and Nup133) showed no detectable binding (Fig. 1c,d and Extended Data Fig. 2). Of the monomeric Nups, those with the most FG motifs in unstructured FG domains, Nup58 (14 FGs), Pom121 (24 FGs) and Nup214 (42 FGs), showed the clearest CLP binding (Fig. 1c and Extended Data Fig. 2), whereas Nup62 (six FGs) also bound. A similar trend was observed for the oligomeric Nups, with Nup42 (12 FGs), Nup153 (29 FGs) and Nup98 (40 FGs) all binding (Fig. 1d and Extended Data Fig. 2). When we accounted for stoichiometry, the seven Nups that showed CLP binding were found to contribute 83% of the total FG-repeat motifs in the NPC (Fig. 1e). Furthermore, these Nups were found to be distributed throughout the NPC, including in the cytoplasmic filaments (Nup42, Nup62, Nup98 and Nup214), the central transport channel (Nup58, Nup62, Nup98 and Pom121) and the nuclear basket (Nup153) (Fig. 1f). The clear coincidence between CLPs and Nup98 was particularly striking, as this Nup is present at 48 copies per NPC, making it the most significant contributor of FGs to the diffusion barrier.

## FG repeats bind a specific pocket on CA

FG-containing peptides are known to have different capsid-binding modes. Previous crystal structures have shown that the short peptides

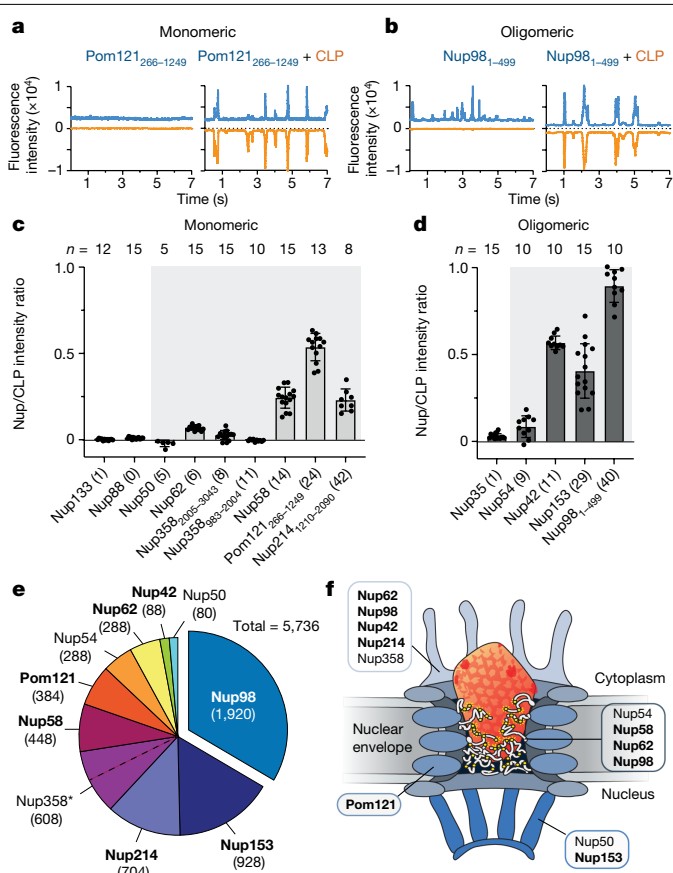

**Fig. 1 | FG-Nups bind to the HIV-1 capsid. a,b,** FFS traces of GFP-Pom121 (**a**) and GFP-Nup98 (**b**), alone or mixed with AF568 CLPs (blue, Nup channel; orange, CLP channel). **c,d,** Nup/CLP fluorescence intensity ratios calculated for monomeric (**c**) and oligomeric (**d**) Nups (grey background denotes canonical FG-repeat Nups with the number of FG motifs per Nup in parentheses). Error bars indicate the standard deviation of the mean. **e,** Relative FG motif contributions to the diffusion barrier based on published Nup stoichiometries (see also Extended Data Table 1). *Due to its size, the FG domain of Nup358 was expressed as two fragments, as denoted by the dotted line. **f,** HIV capsid interacts with Nups distributed throughout the NPC. Nups identified to bind to CLPs are highlighted in bold. $n$, number of scans per sample.

Nup153_{1407–1423} and CPSF6_{313–327} bury their respective FG motifs in the CA N-terminal domain. Although these FG motifs are conformationally identical, the peripheral interactions differ. Nup153_{1407–1423} adopts a linear conformation that bridges two CA monomers, whereas CPSF6_{313–327} forms a more compact structure that packs predominantly against a single CA protomer (Fig. 2a). Notably, in both cases, CA residue N57 forms two critical hydrogen bonds with the main-chain NH and O of the buried phenylalanine. Mutation of CA N57 therefore disrupts binding of both cofactors. However, mutation of residue N74 or A77 only disrupts CPSF6 binding, leaving Nup153 unaffected[16]. CLP mutants (N57D, N74D or A77V) reproduced these binding specificities in our FFS assay (Fig. 2b,c and Extended Data Fig. 3), whereas binding of a CypA control was unaffected by these mutations, demonstrating these CLPs to be otherwise competent for assembly and binding (Extended Data Fig. 3).

To investigate how the capsid recognizes FG repeats, we examined interactions of CA mutants with our identified binders (Nup42_{full-length}, Nup58_{full-length}, Nup62_{full-length}, Nup98_{1–499}, Pom121_{266–1249} and Nup214_{1210–2090}). These FG-Nups demonstrated similar profiles to the Nup153-peptide (Fig. 2d–i and Extended Data Fig. 4). In each case, binding was significantly reduced by N57D but unaffected (and possibly even

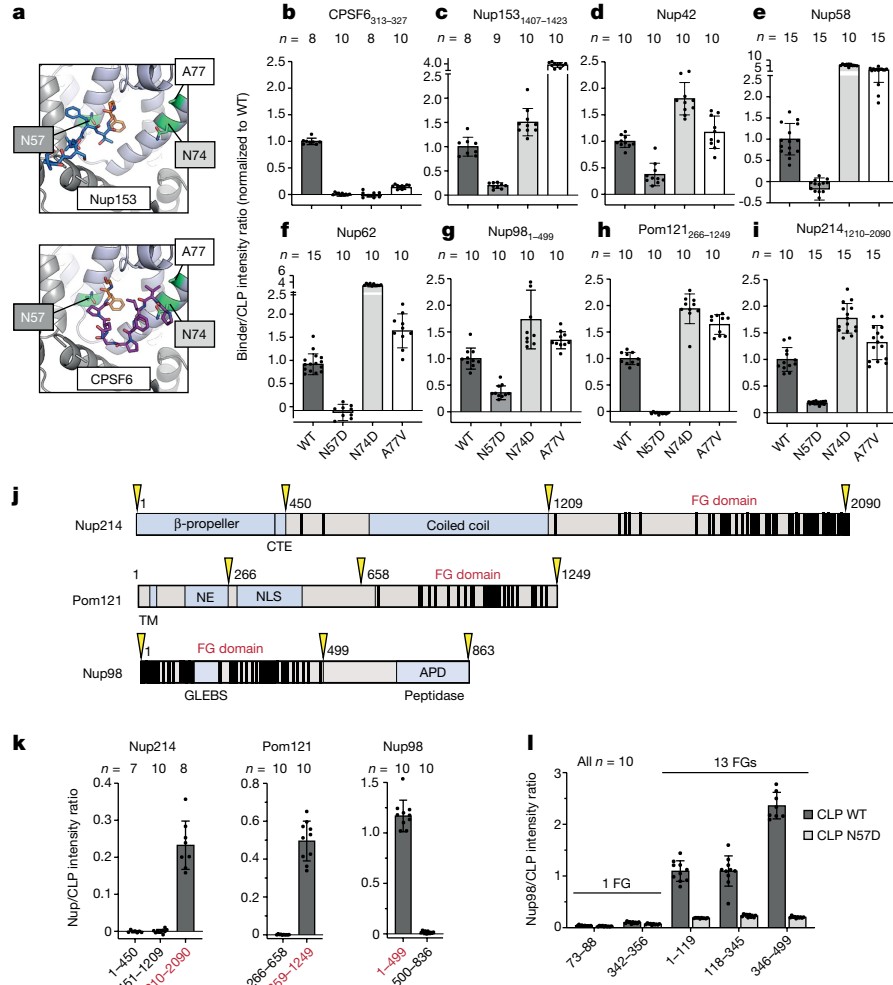

**Fig. 2 | HIV-1 CA specifically recruits FG-repeat domains. a**, Hydrophobic pocket of CA (cartoon) with Nup153 (top, blue; PDB 4U0C) and CPSF6 (bottom, purple; PDB 4U0B) peptides bound. Buried FG motifs are shown in orange, with key CA residues in green. **b–i**, Binding of CLPs (wild-type or mutant) to CPSF6$_{313–327}$ (**b**), Nup153$_{1407–1423}$ (**c**), Nup42 (**d**), Nup58 (**e**), Nup62 (**f**), Nup98$_{1–499}$ (**g**), Pom121$_{266–1249}$ (**h**) or Nup214$_{1210–2090}$ (**i**), as measured by FFS. **j**, Domain architectures of Nup214, Pom121 and Nup98. FG motifs are shown as black bars.

CTE, C-terminally extended peptide; TM, transmembrane domain; NE, inner nuclear envelope-binding region; NLS, nuclear localization signal; GLEBS, Gle2-binding sequence; APD, autoproteolytic domain. Domain boundaries of truncation constructs are shown in yellow. **k**, Binding of truncated constructs of Nup214, Pom121 and Nup98 to CLPs as measured by FFS. FG-repeat domains are shown in red. **l**, Binding of Nup98 peptides and fragments of Nup98 to CLPs. Error bars show standard deviation of the mean; $n$, number of scans per sample.

enhanced) by N74D or A77V. The results obtained with these CLP mutations indicate that the FG-Nups interact with CA N57 in the FG-binding pocket, and that the binding footprints more closely resemble that of Nup153$_{1407–1423}$ than that of CPSF6$_{313–327}$. These observations may explain why capsid N74D and A77V (CPSF6-binding mutants) retain the ability to infect non-dividing cells[8,15,19–21], as they maintain FG interaction and use of the NPC, whereas N57 mutants (FG-binding mutants) are dependent on cell division for maximal infectivity[15,20].

Nup98, Pom121 and Nup214 have distinct domain architectures, each possessing a clearly defined FG-repeat domain with 44, 24 and 40 FG motifs, respectively (Fig. 2j). To determine whether the FG domains were solely responsible for CA binding, we performed side-by-side FFS measurements comparing the FG and non-FG domains. Strikingly, binding was observed for all three FG domains, whereas there was no detectable interaction for any non-FG domain (Fig. 2k and Extended Data Fig. 5). To probe whether binding was driven predominantly by specific FG-containing motifs (as is the case for Nup153 (refs. 15,16)), we chose the strongest binder, Nup98, and further dissected the FG domain into three regions, each containing 13 FGs. Each of the three FG-repeat fragments bound robustly to the wild-type capsid but not N57D (Fig. 2l and Extended Data Fig. 5). CLP interaction with representative Nup98

peptides containing only a single FG motif (representing both GLFG and FGFG types) was below the detection limit for FFS (Fig. 2l), indicating that capsid binding to Nup98 may be driven by weak but FG-specific interactions enhanced by multivalency.

## CA exhibits karyopherin-like properties

Direct interaction between CA and the FG-Nups indicates that the HIV capsid may engage with the diffusion barrier of the NPC. Furthermore, these interactions are reminiscent of the low-affinity multivalent FG binding exhibited by karyopherins, which are uniquely capable of crossing the FG-rich diffusion barrier along with their bound cargoes.

A key advance in the understanding of karyopherin-mediated transport came with the observation that isolated FG-Nups spontaneously form liquid–liquid phase-separated condensates that retain the selective properties of the diffusion barrier of the NPC[18,22]. Although seminal for a mechanistic understanding of nucleocytoplasmic transport, these FG-Nup condensates have not previously been used to study viral nuclear entry. We therefore sought to produce them to explore the interplay between the HIV capsid and the NPC diffusion barrier. Nup98 was chosen as our model system, as it plays a vital part in maintenance

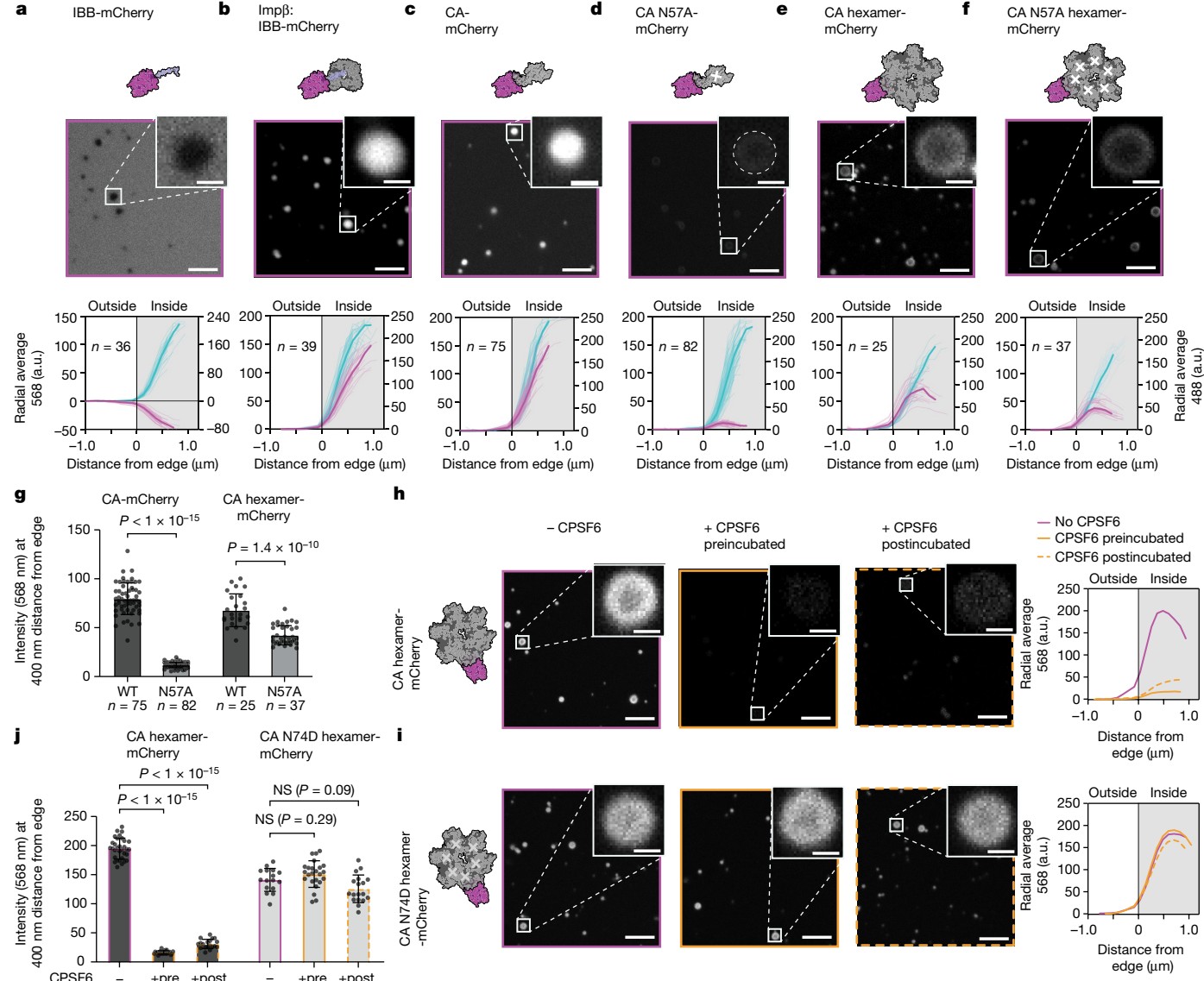

**Fig. 3 | The FG-binding pocket mediates HIV-1 CA entry into Nup98 condensates. a**–**f**, Single *z*-plane images (568 nm) of mCherry-labelled controls or CA constructs (top) and background-subtracted radially averaged fluorescence intensity condensate depth profiles (bottom; cyan, Nup98 channel; magenta, protein-mCherry channel): IBB-mCherry (**a**), Impβ:IBB-mCherry (**b**), CA-mCherry (**c**), CA N57A-mCherry (**d**), CA hexamer-mCherry (**e**), CA N57A hexamer-mCherry (**f**). Mean intensity curves are shown in bold. **g**, Intensity values (568 nm) for CA-mCherry and CA$_{hexamer}$-mCherry WT and N57A at 400 nm from the condensate edge. **h**,**i**, Single *z*-plane images (568 nm)

for CA$_{hexamer}$-mCherry (**h**) and CA N74D$_{hexamer}$-mCherry (**i**) in the absence of CPSF6 (magenta), preincubated with CPSF6 (orange) or postincubated with CPSF6 (dashed orange). Mean fluorescence intensity depth profiles are shown for all conditions. **j**, Intensity values for CA$_{hexamer}$-mCherry and CA N74D$_{hexamer}$-mCherry in the absence and presence of CPSF6 at 400 nm from the condensate edge. All images: scale bar for main, 5 μm; scale bar for inset, 1 μm. *n*, number of condensates analysed per sample. Error bars show the standard deviation of the mean. Statistical tests: two-tailed *t*-test (**g**) and one-way analysis of variance (**j**). Impβ, importin-β.

of the integrity of the NPC[23] and contributes the most FGs (Fig. 1e), and, among the phase-separating FG-Nups, it produces condensates that most accurately recapitulate the selectivity properties of the diffusion barrier in vitro[18,24,25]. Notably, of the Nups, Nup98 was also the clearest CA binder in our FFS assay (Fig. 1b,d).

We produced the unstructured FG domain of Nup98 (residues 1–499) under denaturing conditions and observed spontaneous formation of spherical, phase-separated condensates (average diameter 1.3 μm) after shock dilution into aqueous buffer (see Methods for details). We tested these condensates for their selective properties by mixing them immediately after formation (when they still maintained liquid-like characteristics) with mCherry fused to the importin-β-binding domain of importin-α (IBB-mCherry). IBB-mCherry (33 kDa) was excluded from

the Nup98 phases (Fig. 3a and Extended Data Fig. 6) but was transported into the condensates within minutes of addition of importin-β (Fig. 3b and Extended Data Fig. 6). These results demonstrate that our Nup98 condensates exhibit two essential functions of the NPC permeability barrier, passive exclusion and karyopherin-facilitated transport. Furthermore, the addition of importin-β did not result in uptake of wild-type mCherry (27 kDa; Extended Data Fig. 6), showing that the presence of importin-β does not affect the size-selective properties of the Nup98 condensates. To test whether CA could also enter the FG-Nup phase, we fused mCherry to the carboxyl terminus of CA (CA-mCherry, 52 kDa). At the concentrations used in our assay (45 μM), we expected CA-mCherry to exist in equal amounts as monomer and dimer ($K_d$ approximately 40 μM)[26], with the latter carrying two

FG-binding sites. Strikingly, CA-mCherry rapidly partitioned into the Nup98 condensates (Fig. 3c and Extended Data Fig. 6), whereas diffusion into the condensate was almost entirely abolished after N57A mutation (Fig. 3d,g and Extended Data Fig. 6). Next, we produced a cysteine cross-linked hexamer (CA$_{hexamer}$-mCherry)[27] carrying an average of one CA-mCherry per hexamer. At 182 kDa, this construct is well beyond the size limit for passive diffusion across the NPC but presents six FG-binding sites appropriately positioned relative to their neighbouring CA protomers. Again, we observed CA$_{hexamer}$-mCherry partitioning into the condensates (Fig. 3e and Extended Data Fig. 6), an effect that was again significantly reduced by N57A (Fig. 3f,g and Extended Data Fig. 6).

To further probe the functional relevance of the FG-binding pocket, we sought to test whether known cofactors could compete for CA binding and therefore influence Nup penetration. The recently developed drug lenacapavir binds the FG pocket but was not amenable to study in our systems as it resulted in CA clustering, consistent with its known overassembly effect[28]. As an alternative, we used the host cofactor CPSF6, which had no such confounding effects. CPSF6$_{313-327}$ peptide preferentially binds to CA$_{hexamer}$ ($K_d = 50 \mu M$) over unassembled CA ($K_d$ approximately 700 $\mu M$)[16]. As expected, pretreatment with 500 $\mu M$ CPSF6$_{313-327}$ had no observable effect on CA-mCherry (Extended Data Fig. 7), whereas it abolished the ability of CA$_{hexamer}$-mCherry to partition into the Nup98 condensates (Fig. 3h,j and Extended Data Fig. 7). This effect was specific to the interaction with the FG pocket, because CPSF6$_{313-327}$ had no effect on CA(N74D)$_{hexamer}$-mCherry, which does not bind CPSF6$_{313-327}$ (Fig. 3i,j and Extended Data Fig. 7). Posttreatment with CPSF6$_{313-327}$ also removed CA$_{hexamer}$-mCherry from the Nup98 condensates. These effects of CPSF6 support the role of the FG-binding pocket in mediating capsid entry into Nup98 condensates; moreover, the pretreatment and posttreatment results indicate that competing cofactors may be able to influence both NPC engagement and disengagement. Indeed, these results are consistent with the proposed 'ratchet' mechanism by which CPSF6 extracts the capsid from the nuclear face of the NPC[9].

Whereas these results indicate that CA may be able to enter the NPC autonomously, previous studies have suggested that the karyopherins importin-α3 (ref. 29), transportin-1 (TNPO1)[30] and transportin-3 (TNPO3)[31] each play a part in HIV-1 nuclear entry through a direct interaction with the capsid. However, we observed no specific binding between capsid and importin-α, TNPO1 or TNPO3 by FFS (Extended Data Fig. 8). Furthermore, rather than enhancing CA entry into Nup98 condensates, these karyopherins reduced entry under our experimental conditions (Extended Data Fig. 8), consistent with the prediction that CA and karyopherins would compete for FG binding and therefore have a similar mechanism of action. Notably, a fully assembled capsid with more than 1,000 FG-binding sites is likely to be a much stronger competitor for the FG motifs than CA hexamers. Thus, we argue that our observation of CA penetrating FG condensates and facilitating the entry of an mCherry pseudo-cargo demonstrates that the HIV capsid protein possesses intrinsic karyopherin-like properties and can function independently of the canonical nucleocytoplasmic transport factors.

## HIV capsids enter Nup98 condensates

To chaperone the viral genome and enzymes across the nuclear envelope, the HIV capsid must be capable of entering the NPC intact. To investigate the FG-Nup-penetration properties of complete capsids, we induced coassembly of CA(A204C) with CA(A204C)-mCherry at a ratio of 200:1. These fluorescent CLPs formed the expected heterogeneous mix of lattice structures that model the pleiomorphic capsids found in HIV virions (Extended Data Fig. 1). Confocal microscopy showed that after the addition of CLPs, fluorescent puncta were recruited to the periphery of the Nup98 condensates (Fig. 4a and Extended Data Fig. 9) but could not resolve whether these structures had penetrated

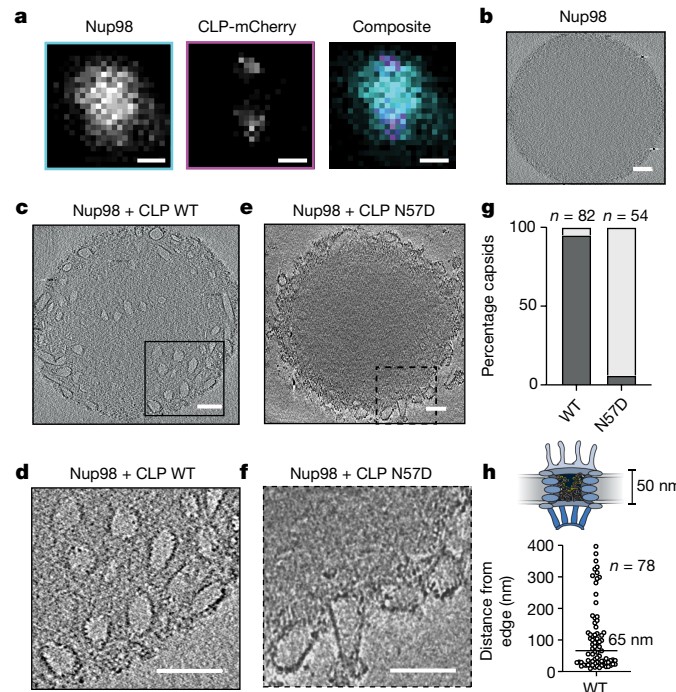

**Fig. 4 | Intact HIV-1 CLPs penetrate Nup98 condensates via FG-binding pocket. a**, CLP-mCherry assemblies (punctae) recruited to Nup98 condensates. Single $z$-plane images for 488 nm (cyan), 568 nm (magenta) and composite. Scale bar, 500 nm. **b,c,e**, Representative slices from cryo-ET images of Nup98 alone (**b**), in the presence of CLP WT (**c**) and or CLP N57D (**e**). Scale bars, 100 nm. **d,f**, Zoomed images from **c** (**d**) and **e** (**f**), showing an average of five slices with a total thickness of 5.2 nm. Scale bars, 100 nm. **g**, Populations ($n$) of WT and N57D CLPs excluded from (light grey) or partitioned into (dark grey) condensates. **h**, Penetration depths of ($n$) CLP WT, measured from the edge of eight unique condensates.

the condensates, nor whether the structural integrity of closed CA cones or spheres was maintained after entry.

To examine whether complete HIV-1 capsids can penetrate the Nup98 condensates, we turned to cryo-ET, a technique that (in contrast to previous measurements) also allowed us to study CA-Nup98 condensate interactions in a label-free context. We ensured that the condensates were electron transparent by reducing the Nup98 concentration, resulting in smaller condensates (diameter 400 nm to 1 $\mu m$), which we imaged with and without CLPs introduced immediately before freezing. In the absence of CA, Nup98 condensates had a uniform, featureless internal texture, consistent with a phase-separated, intrinsically disordered protein (Fig. 4b and Extended Data Fig. 9). Strikingly, after addition of wild-type CLP to preformed Nup98 condensates, we observed internalized pleomorphic substructures, including conical structures similar to authentic HIV capsids (Fig. 4c,d and Extended Data Fig. 9). These structures carried an electron-dense exterior with evidence of a repetitive lattice structure. Internal negative space confirmed that wild-type CLPs were empty, showing that the integrity of the CLPs was not affected even when they were fully immersed in the FG-motif-dense condensates. Conversely, CLPs carrying the N57D mutation were not internalized and were observed either at the surface of the Nup98 condensates or in the bulk solution (Fig. 4e–g). Furthermore, the median penetration depth of wild-type CLPs was 65 nm (Fig. 4h), which is greater than the 50 nm required to cross the central transport channel of the NPC[32]. These results show that complete HIV capsids have properties that enable them to sufficiently penetrate and enter the key material of the diffusion barrier of the NPC, specifically via the FG-binding pocket, while preventing disordered Nups from infiltrating the capsid interior.

## Discussion

There are several reasons the HIV:NPC interaction has remained poorly understood. In cellulo approaches can be confounded by both the essentiality of the Nup genes and the complex interactome of the NPC. Protein depletion approaches may be incomplete and/or result in changes in the abundance and/or mislocalization of other Nups, whereas pulldown assays can yield false positives owing to Nup–Nup interactions[19]. Nevertheless, more than one third of Nups have previously been implicated in various aspects of the HIV lifecycle using these methods[15,19,20,33–37] (Supplementary Table 1). Despite similar approaches, the consistency with which specific Nups are identified varies considerably. By performing an in vitro screen of the CA:Nup interactions, we have circumvented the cellular challenges to show that the capsid specifically engages with the multitude of FG motifs found within the NPC diffusion barrier. Notably, we observe interactions with Nups located in the cytoplasmic filaments, throughout the central transport channel and in the nuclear basket, implying that capsid:Nup interactions can, in principle, occur at all points along the translocation pathway.

FG repeats have been classified into two main types based on the residues immediately upstream, FxFG/FG and GLFG. We observe that the HIV capsid interacts with both. The two types are thought to segregate in the NPC, with FxFG/FG repeats enriched at the cytoplasmic and nuclear peripheries, and GLFG repeats enriched in the central channel where they function as the main component of the diffusion barrier[5]. The only GLFG-containing Nup in the human NPC is Nup98, which is thus a crucial barrier that any foreign entity must overcome to enter the nucleus. It is therefore striking that Nup98 is the clearest CA binder in our assays and that Nup98 condensates readily take up HIV capsids.

Structural studies of karyopherins have shown that they possess multiple surface pockets for binding FG dipeptides, and it is these FG interactions that enable karyopherins and their bound cargoes to selectively cross the diffusion barrier. Several karyopherins have been shown to interact directly with FG motifs of Nup42, Nup62, Nup98, Nup153 and Nup214 (refs. 38–42); notably, all of these Nups have been identified as CA binders in our study. The remarkable structural and functional similarities between CA:Nup and karyopherin:Nup interactions imply that rather than engaging with host nuclear transport receptors, the HIV capsid has evolved to become the karyopherin for the protected HIV genome.

The reductionist system of Nup98 condensates allows us to demonstrate the principle of FG-mediated phase partitioning and probe the specific sites on the capsid that permit it to be a client of these phases. We acknowledge that the human NPC is one of the most complex macromolecular assemblies known. The diffusion barrier itself has defied a consensus physicochemical description owing to its intrinsic disorder, multicomponent composition and tethered nano-environment[43–45]. Indeed, although it is at present one of the best-known examples of a demixed aqueous environment in the cell, the nuanced complexity means that experimental models are always approximations and limitations should be recognized when extrapolating precise details concerning nuclear entry[25]. In the case of HIV, the concept that the capsid has the ability to partition into FG-containing liquid–liquid phases has several important implications for viral infection. In such a framework, molecular weight does not represent a barrier to nuclear entry. Any assembly should be able to enter the nucleus provided that its dimensions do not exceed those of the central transport channel and that it is not excluded from the FG-Nup phase. Notably, recent cryo-ET images of intact HIV-1 capsids travelling through nuclear pores showed that the central channel can become wider than previously thought to accommodate the width of the complete capsid[2]. On the basis of a combination of these cryo-ET studies with our work presented here, the HIV capsid satisfies both criteria of appropriate size and penetration capability.

Furthermore, the collective effects of FG interactions can be recast in terms of surface energies and, in the simple case of the threefold intersection between capsid–condensate, capsid–cytosol and condensate–cytosol interfaces, a wetting angle. This is important because alongside the geometries of the capsid and the central transport channel, the surface energies determine the capillary forces that act on the capsid[46], with potential to shed light on how capsids orient relative to the NPC, and possibly explain why, among the retrovirus family, only those that traverse the NPC (that is, the lentiviruses) adopt a conical capsid morphology.

The HIV capsid is also likely to encounter other cellular compartments that could be described as phase separated. One of the best-characterized capsid binders, CPSF6, is known to partition into membraneless compartments in the nucleus, where it plays an essential part in targeting HIV integration[6,47,48]. Impairment of the CA:CPSF6 interaction (through A77V, N74D mutations or CPSF6 knockdown[9,11,13,49]) results in arrest of CA at the nuclear membrane and integration at the nuclear periphery, indicating that the capsid may not be able to properly disengage from the NPC. Furthermore, cytoplasmic mislocalization of CPSF6 (through either ectopic expression of CPSF6 lacking a nuclear localization signal or depletion of the dedicated CPSF6 karyopherin, TNPO3) prevents the HIV capsid from entering the nucleus[6,50]. Our data are consistent with a model in which cytoplasmic CPSF6 competes for FG-Nup binding, thereby preventing nuclear entry, while nuclear CPSF6 helps to extract the capsid out of the NPC into the nucleus and directs it towards the sites of active transcription before integration. It is worth noting that endogenous karyopherins similarly require assistance to disengage from the NPC. Importin-β, for example, is released from the NPC by Ran-GTP[51]. Thus, nuclear CPSF6 probably has an analogous release function in the karyopherin mimicry model of capsid-mediated nuclear entry. We also note that Nups can be found in other compartments of the cell, such as annulate lamellae, and in condensates in the nucleus[52]. Although their relevance to HIV is yet to be established, our data indicate that non-canonical Nup function could potentially influence early infection.

Whether the HIV capsid maintains its structural integrity while embedded in FG-containing phases (the NPC or CPSF6-rich compartments such as speckle-associated domains) remains an open question. Furthermore, it has been proposed that the capsid ultrastructure may remodel after transit of the NPC[2,11,53]. Although we have not explicitly addressed this question with our cross-linked capsid model, the high concentration of FGs in the NPC (more than 100 mM[4]) indicates that many FG pockets are likely to be occupied simultaneously. High occupancy of the FG pocket with capsid-targeting drug lenacapavir or related compound PF74 affects capsid integrity and CA lattice stability[54]. It is therefore conceivable that effects observed only near saturation (such as changes to the capsid ultrastructure) may indeed take place during NPC transit and/or after arrival in CPSF6-containing nuclear compartments. The role of phase partitioning in HIV uncoating has not previously been considered and will probably provide key insights into this enigmatic process.

In summary, we propose a model in which the HIV capsid mimics the karyopherin mechanism of transiting the NPC by solubilizing in the diffusion barrier through specific, multivalent FG interactions. This mechanism of nuclear entry is likely to be conserved in other medically important viruses, which may prove to be treatable by a strategy analogous to the use of lenacapavir in HIV. Certainly, further study of the HIV capsid will continue to provide a paradigm for viral infection and nuclear transport mechanisms and reveal opportunities for improving lentiviruses as gene delivery vectors, along with new therapeutic approaches that may be extended to unrelated viruses.

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

# Methods

## Constructs

Nup42, Nup50, Nup54, Nup62, Nup88, Nup133 and Nup214 in the pDONR221 gateway master vector were purchased from DNASU and cloned into the cell=free expression vector pCellFree_03 (ref. 55), using the Gateway method, to produce fusion proteins with an N-terminal GFP tag (see Supplementary Table 2 for details). Nup35, Nup58, Nup98, Nup358 and Pom121 were cloned from complementary DNA (cDNA) prepared from HEK293T cells. Briefly, RNA was extracted using an Isolate II RNA mini kit (Bioline), and cDNA was synthesized using oligo(dT) primers and SuperScript IV (Thermo Fisher Scientific). Target genes were then cloned into pCellFree_03 using the Gibson assembly protocol (New England Biolabs). Nup98 and Pom121 fragments were purchased as gBlocks from IDT and cloned into pCellFree_03. pET28a vectors containing mCherry2-Nup98 fragments, as well as pGEX-6P-3 vectors containing importin-α, TNPO1 and TNPO3 were purchased from GenScript. Cell lines (HEK239T) were only used for cDNA preparation in this study, with gene identity subsequently confirmed by Sanger sequencing.

## Protein expression and purification

**HIV-1 CA proteins.** HIV-1 CA proteins (K158C, A204C, R18G/A204C, N57D/A204C, N74D/A204C, N77V/A204C) were expressed in *Escherichia coli* C41 or Rosetta2 cells and purified as previously described[56]. HIV-1 CA K158C was labelled with Alexa Fluor 568-C5-maleimide (AF568, Thermo Fisher Scientific, A20341) and mixed in an approximately 1:200 ratio with unlabelled HIV-1 CA A204C before assembly. CA lattice assembly was carried out as described by Lau et al. [17] with the modification of a 15 min incubation at 37 °C and no overnight incubation. $CA_{hexamer}$, $CA_{hexamer}$-mCherry and CA-mCherry were purified based on previously described protocols[27]. Briefly, $CA_{hexamer}$, $CA_{hexamer}$-mCherry (pOPT) and CA-mCherry (pET21a) were expressed in *E. coli* C41 cells. Cells were grown at 37 °C, protein expression was induced with 0.5 mM isopropyl β-D-1-thiogalactopyranoside (IPTG) at an optical density of 0.6, and cell growth was continued overnight at 18 °C. Cells were collected (4,000*g*, 10 min, 4 °C), and the cell pellets were resuspended in lysis buffer (purification buffer 50 mM Tris pH 8, 50 mM NaCl, 2 mM dithiothreitol (DTT); +cOmplete EDTA-free Protease Inhibitor (Roche)) and lysed by sonication. The lysate was cleared by centrifugation (16,000*g*, 60 min, 4 °C), and an ammonium sulfate (20% w/v) precipitation was carried out on the soluble fraction. After stirring for 30 min at 4 °C, the precipitated material was pelleted (16,000*g*, 20 min, 4 °C). $CA_{hexamer}$ was further purified by resuspending the pellet in 100 mM citric acid pH 4.5, 2 mM DTT and dialysed against the same buffer three times. The precipitated protein was pelleted (16,000*g*, 20 min, 4 °C), and the soluble fraction was collected for further purification. $CA_{hexamer}$-mCherry and CA-mCherry could not be purified with citric acid precipitation owing to a possible loss of mCherry fluorescence. The ammonium-sulfate-precipitated pellets were resuspended in 8 ml purification buffer per litre culture, run over a 20 ml Hi-TRAP Q column in purification buffer and eluted in a 1–100% gradient with buffer B (50 mM Tris pH 8, 500 mM NaCl). Protein-containing fractions were pooled.

CA-mCherry was further purified by gel filtration (Superdex 200 in purification buffer), and fractions containing clean protein were pooled and snap frozen to be stored at −80 °C.

To assemble the final cross-linked hexameric $CA_{hexamer}$-mCherry, $CA_{hexamer}$ and $CA_{hexamer}$-mCherry were mixed in a 5:1 ratio and assembled by means of the following dialysis steps: twice against 50 mM Tris pH 8, 1,000 mM NaCl, 2 mM DTT; twice against 50 mM Tris pH 8, 1,000 mM NaCl; and twice against 50 mM Tris pH 8, 40 mM NaCl. The assembled $CA_{hexamer}$-mCherry was finally purified with gel filtration (S200), and fractions containing hexameric $CA_{hexamer}$-mCherry carrying one mCherry were pooled and snap frozen to be stored at −80 °C.

**Cell-free expression.** Cell-free expression of GFP fusion proteins was performed as described by Lau et al.[17]. Briefly *Leishmania tarentolae* extract was supplemented with RnaseOUT (1:1,000, Invitrogen) on ice. pCellFree_03-Nup plasmids (150 ng μl$^{-1}$) were then added to the expression mix, followed by incubation at 27 °C for 3 h. Following incubation, the undiluted extract was mixed 1:1 with NuPAGE LDS loading buffer (Thermo Fisher Scientific), 10 mM DTT, and the purity and degradation of the expressed Nup were assessed by sodium dodecyl sulfate polyacrylamide gel electrophoresis in-gel fluorescence.

**Importin-β, IBB-mCherry, Nup98.** The mCherry protein was kindly provided by J. Goyette. Importin-β and IBB-mCherry in pET11a (GenScript) were expressed as His-fusion proteins in *E. coli* Rosetta2 cells. Cells were grown at 37 °C, protein expression was induced with 0.5 mM IPTG at an optical density of 0.6, and cell growth was continued overnight at 18 °C. Cells were collected (4,000*g*, 10 min, 4 °C), and the cell pellets were resuspended in lysis buffer (purification buffer: 20 mM Tris pH 7.5, 150 mM NaCl, 2 mM DTT; +cOmplete EDTA-free Protease Inhibitor (Roche)) and lysed by sonication. The lysate was cleared by centrifugation (16,000*g*, 45 min, 4 °C), and the supernatant was bound to Ni-beads (Ni Sepharose High Performance, Cytiva) preincubated in purification buffer; then, imidazole was added to the protein–Ni slurry to a final concentration of 10 mM, and the mixture was incubated for 2 h at 4 °C. The beads were washed with 20 column volumes of wash buffer (purification buffer + 20 mM imidazole) and eluted in 2 ml fractions with elution buffer (purification buffer + 500 mM imidazole). Protein-containing fractions were pooled, and cleavage of the His-tag was achieved with TEV (IBB-mCherry) or SUMO (importin-β) in overnight dialysis against TEV-cleavage buffer (50 mM Tris pH 7.5, 1 mM DTT) or SUMO-cleavage buffer (50 mM Tris pH 7.5, 150 mM NaCl, 1 mM DTT). Cleaved proteins were added to Ni-beads, and the flow-through was collected. Proteins were finally purified by size-exclusion chromatography (SEC; Hi Load16/60 Superdex 200) in purification buffer. Clean protein fractions were pooled and snap frozen to be stored at −80 °C. Nup98 in pET28a was purified as described above with the following changes: the buffers used were purification buffer (6 M GuHCl, 50 mM TrisHCl pH 7.5, 2 mM DTT), wash buffer (6 M GuHCl, 20 mM imidazole pH 8) and elution buffer (500 mM GuHCl, 500 mM imidazole pH 8). After induction with IPTG, cells were grown further for 3 h at 30 °C before being collected. Sufficient purity of the protein was achieved with affinity chromatography; consequently, no SEC was performed. The His-tag was not removed. The protein–Ni slurry was stored at 4 °C for phase-separation assays.

**Importin-α, TNPO1, TNPO3.** Importin-α, TNPO1 and TNPO3 were expressed as GST-fusion proteins from *E. coli* Rosetta2 cells, as described above for importin-β and IBB-mCherry. Cell pellets were then resuspended in 25 mM Tris pH 7.5, 500 mM NaCl, 1 mM DTT, 10% glycerol and 1× cOmplete EDTA-free Protease Inhibitor (Roche) and lysed by sonication. Cell lysates were subjected to centrifugation (18,000*g*, 60 min, 4 °C). Clarified lysates were bound to GSH-resin and washed with 25 mM Tris pH 7.5, 150 mM NaCl, 1 mM DTT, 10% glycerol. On-column cleavage of the GST-tag was achieved with PreScission Protease. Cleaved proteins were subjected to SEC over a Superdex 200 26/600 (Cytiva) and equilibrated in 50 mM Tris pH 7.5, 200 mM NaCl, 1 mM DTT, 10% glycerol. Eluted proteins were concentrated, snap frozen and stored at −80 °C.

## Fluorescence fluctuation spectroscopy

FFS was performed as described by Lau et al.[17]. Briefly, cell-free expressed GFP-Nup proteins (approximately 50 nM; equivalent to approximately 2,500 photon count) or AF488 peptides (100 nM) were mixed with AF568-HIV-1 CA assemblies (12 μM) in 50 mM Tris, pH 8, 150 mM NaCl. Fluorescence traces were recorded for 15 s per trace in

1 ms bins using a scanning stage operated at $1 \mu m s^{-1}$. Typically, measurements were repeated 10–15 times per sample.

## Determining the number of FGs in relative solvent accessibility

FG motif accessibility was determined by calculating per-residue relative solvent accessibility (RSA) from AlphaFoldDB structures to determine order and disorder[57]. RSA-based order and disorder for all Nups except Nup358 were accessed through Mobi-DB[58].

Owing to the length of Nup358, AlphaFold2 predictions were performed as three FG-containing sections (982–2004, 2005–3043 and 3058–3224), with their per-residue RSA-based disorder propensity calculated locally. Binary designations of order and disorder were assigned using a threshold optimized on Critical Assessment of Protein Intrinsic Disorder (CAID) data (0.581)[57].

## Phase separation

For phase-separation assays, the Nup98-saturated Ni slurry ('Protein Expression and Purification'; Nup98 binding capacity Ni-beads 40 mg ml$^{-1}$ medium) was mixed with Alexa Fluor 488-C5-maleimide (AF488, Thermo Fisher Scientific; final concentration 10 μM), followed by incubation for 5 min. We aimed for as little labelling with AF488 as possible with enough signal to noise to avoid confounding effects from the fluorophore for phase separation. The slurry was washed with 10 column volumes of wash buffer 2 (500 mM GuHCl, 20 mM imidazole pH 8) + 2 mM DTT, followed by a second wash step with wash buffer 2 without DTT to remove the DTT and residual fluorophore. The labelled protein was eluted with elution buffer for a final protein concentration of approximately 4 mg ml$^{-1}$ (estimated from sodium dodecyl sulfate polyacrylamide gel electrophoresis against a bovine serum albumin standard), and phase separation of Nup98 was induced by 1:10 shock dilution into assay buffer (50 mM Tris pH 7.5, 150 mM NaCl). We observed the most 'fluid' condensates using shock dilution; these condensates 'hardened' within minutes of forming, as tested with an importin-β:IBB-mCherry control. This resulted in limited equilibration of the condensates with some fluorescent substrates (for example, as seen for CA$_{hexamer}$-mCherry). Fluorescent substrates were added to the phase-separated Nup98 condensates and immediately imaged.

## Confocal microscopy

Imaging was performed with a Zeiss LSM 880 inverted laser scanning confocal microscope using a ×63 oil immersion objective (numerical aperture = 1.4) (Leica). The substrate–Nup98 reaction mixes were transferred into a 12-well silicone chamber (Ibidi) on a 170 ± 5 μm cover slide. Z-stacks were taken around the centres of the phase-separated Nup98 condensates (position with highest diameter) with sequential imaging at 488 and 568. Images were processed using ImageJ2 v.2.9.0/1.53t, MATLAB R2020a and Prism v.9.4.1.

Nup98 condensate-CA experiments were performed with mCherry-labelled CA (CA-mCherry and CA$_{hexamer}$-mCherry), as it has previously been reported that fluorescent dyes such as Alexa568 can non-specifically interact with Nup condensates[59].

Experimental details for Fig. 3a–f and Extended Data Figs. 6 and 7: Nup98 condensates were immediately mixed with the following proteins: mCherry and IBB-mCherry (200 μM), importin-β:IBB-mCherry and importin-β + mCherry (both 100 μM), CA-mCherry and CA N57A-mCherry (45 μM), CA$_{hexamer}$-mCherry and CA N57A$_{hexamer}$-mCherry (65 μM). The mixed Nup98–protein samples were imaged after around 5 min incubation on the coverslip.

Experimental details for Fig. 3h,i and Extended Data Figs. 8–10: for the CPSF6$_P$ preincubation sample, CA-mCherry, CA$_{hexamer}$-mCherry and CA N74D$_{hexamer}$-mCherry (65 μM final) were incubated with CPSF6 (500 μM final) for 10 min, followed by mixing with Nup98 condensates and imaging after around 5 min incubation on the coverslip. The control sample and the CPSF6 postincubation sample were incubated with MQW for 10 min, followed by mixing with Nup98 condensates and imaging after around 5 min incubation on the coverslip. For the CPSF6 postincubation sample, CPSF6 was added to the sample on the coverslip and imaged after 5, 10, 15 and 20 min.

Experimental details for Extended Data Fig. 8c–j: CA-mCherry and CA$_{hexamer}$-mCherry (final 25 μM) were incubated with importin-α:importin-β, TNPO1 and TNPO3 (all final 2.5 μM), respectively, for 10 min, followed by mixing with Nup98 condensates and imaging after around 5 min incubation on the coverslip.

Experimental details for Fig. 4 and Extended Data Fig. 9b–d: CLP-mCherry was assembled with a ratio of 1:100 CA-mCherry (final 0.4 μM) to CA (final 40 μM) by addition of NaCl (final 1 mM) and 15 min incubation at 37 °C. After assembly, the sample was spun down hard (18,000g, 7 min) to separate the assemblies from non-assembled monomer. The pellet was resuspended in assay buffer (50 mM Tris pH 7.5, 150 mM NaCl; resuspension volume the same as the original sample volume), followed by a slow spin of the resuspended pellet (4,000g, 5 min) to remove aggregates. Nup98 condensates were immediately mixed with CA-mCherry assemblies (1 μM monomer concentration) and imaged after around 5 min incubation on the coverslip.

## Radial intensity profiles

Averaged radial intensity profiles were obtained using the plugin Radial Profile in ImageJ. Z-slices were chosen at the centres of the phase-separated Nup98 condensates (position with largest diameter). Profiles were background subtracted. The edge of a condensate was defined from the 488 channel as the first intensity value greater than 5. Then, 200 nm was subtracted from this value to account for point spread function of the fluorescent pixel. This point was defined as point 0 μm in the radial averaged intensity graphs (Fig. 3 and Extended Data Figs. 6–8).

## Cryo-electron microscopy

Nup98 condensates were prepared as for confocal microscopy with the following modifications. To induce the formation of smaller condensates, the unlabelled Nup98 was diluted two-fold in 500 mM GuHCl, 500 mM imidazole pH 8, before a 1:10 shock dilution into assay buffer (50 mM Tris pH 7.5, 150 mM NaCl).

Unlabelled CLPs were prepared as described above. A two-step centrifugation protocol was performed as for confocal microscopy. Equal volumes of CLP suspension were mixed with freshly shock-diluted condensates, immediately before to plunge freezing to minimize sample aggregation.

**Frozen-hydrated sample preparation.** A mixture (4.5 μl) of freshly mixed Nup98 condensate and capsid with protein A-gold (10 nm) was applied onto a glow-discharged Quantifoil R2/2 copper grid (Quantifoil Micro Tools). The grid was blotted in the front for 2.5 s at 15 °C with 90% relative humidity and then plunged into liquid ethane using a Lecia EM GP device (Lecia Microsystem). The vitrified grids were then stored in liquid nitrogen before cryo-ET imaging.

**Cryo-ET imaging and reconstruction.** The grids were imaged on a Talos Arctica electron microscope (Thermo Fisher Scientific) operated at 200 kV acceleration voltage. Cryo-ET data were collected with single-axis tilt on a Falcon III direct electron detector (Thermo Fisher Scientific) in linear mode at a magnification of ×28,000 with a pixel size of 5.23 Å. Tilt series were collected using the dose-symmetric scheme[60] from −60 to 60° at 3° intervals using Tomography software (Thermo Fisher Scientific) with the defocus value set to −10 μm. Total dose for each tilt series ranged from 60–70 e/Å$^2$. Images of tilt series were binned two-fold before tomograms were reconstructed. Three-dimensional reconstructions from tilt series were generated with the IMOD package[61]. Fiducial tracking was used to align the stack of tilted images.

## Statistics and reproducibility

Confocal microscopy images and graphs in Fig. 3 and Extended Data Figs. 6–8 show representative data from at least three independent experiments with similar results. Confocal microscopy images in Fig. 4a and Extended Data Fig. 9 are representative of three independent experiments with similar results. Cryo images and graphs in Fig. 4b–h and Extended Data Fig. 9 show representative data from two independent experiments with similar results.

## Reporting summary

Further information on research design is available in the Nature Portfolio Reporting Summary linked to this article.

## Data availability

The experimental data that support the findings of this study are available at Dryad with the following identifier: https://doi.org/10.5061/dryad.b2rbnzsm0. Source data are provided with this paper.

## Code availability

FFS traces were analysed using custom software TRISTAN freely available at https://github.com/lilbutsa/Tristan.

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

**Acknowledgements** We thank J. Stear and J. Walsh for critical reading of this manuscript and R. Yu for assistance with electron microscopy data collection. This work was supported by a National Health and Medical Research Council Ideas Grant (GNT2013215; D.A.J., T.B.) and Wellcome Trust Collaborator Award (214344/Z/18/Z; D.A.J., T.B., G.J.T.). C.F.D. was supported by a NHMRC Early Career Fellowship (GNT1110116). D.A.J. was supported by a UNSW Scientia Fellowship. The confocal imaging component of this study was carried out using instruments situated in and maintained by the Katharina Gaus Light Microscopy Facility at UNSW. We acknowledge the use of the Cryo Electron Microscopy Facility through the Victor Chang Cardiac Research Institute Innovation Centre, funded by the New South Wales government, and the Electron Microscope Unit at UNSW Sydney. We also acknowledge the use of the Recombinant Products Facility and the Structural Biology Facility in the Mark Wainwright Analytical Centre – UNSW, funded in part by the Australian Research Council Linkage Infrastructure, Equipment and Facilities Grant: ARC LIEF 190100165. We acknowledge the Bedegal people of the Eora nation, the traditional custodians of the land upon which this research took place. D.A.J. thanks the late V. Jacques for all her support and encouragement.

**Author contributions** C.F.D. and S.H. contributed equally to this work. C.F.D. was responsible for design and execution of fluorescence fluctuation experiments, and S.H. was responsible for design and execution of Nup phase separation and confocal microscopy experiments. C.F.D., S.H., J.R., N.A. and N.L. acquired and analysed cryo-electron microscopy data. A.J.T. contributed useful discussions around experimental design and contributed to FFS and confocal measurement analysis. E.S. and Y.G. contributed to FFS experimental design, cell-free protein expression and data collection. S.C.A-I. and R.G.M. contributed to condensate experimental design and data interpretation. G.J.T. and T.B. contributed to experimental design and data interpretation. D.A.J. supervised all aspects of the project. C.F.D, S.H., A.J.T. and D.A.J. wrote the manuscript with input from all authors.

**Funding** Open access funding provided through UNSW Library.

**Competing interests** The authors declare no competing interests.

**Additional information**
**Correspondence and requests for materials** should be addressed to D. A. Jacques.

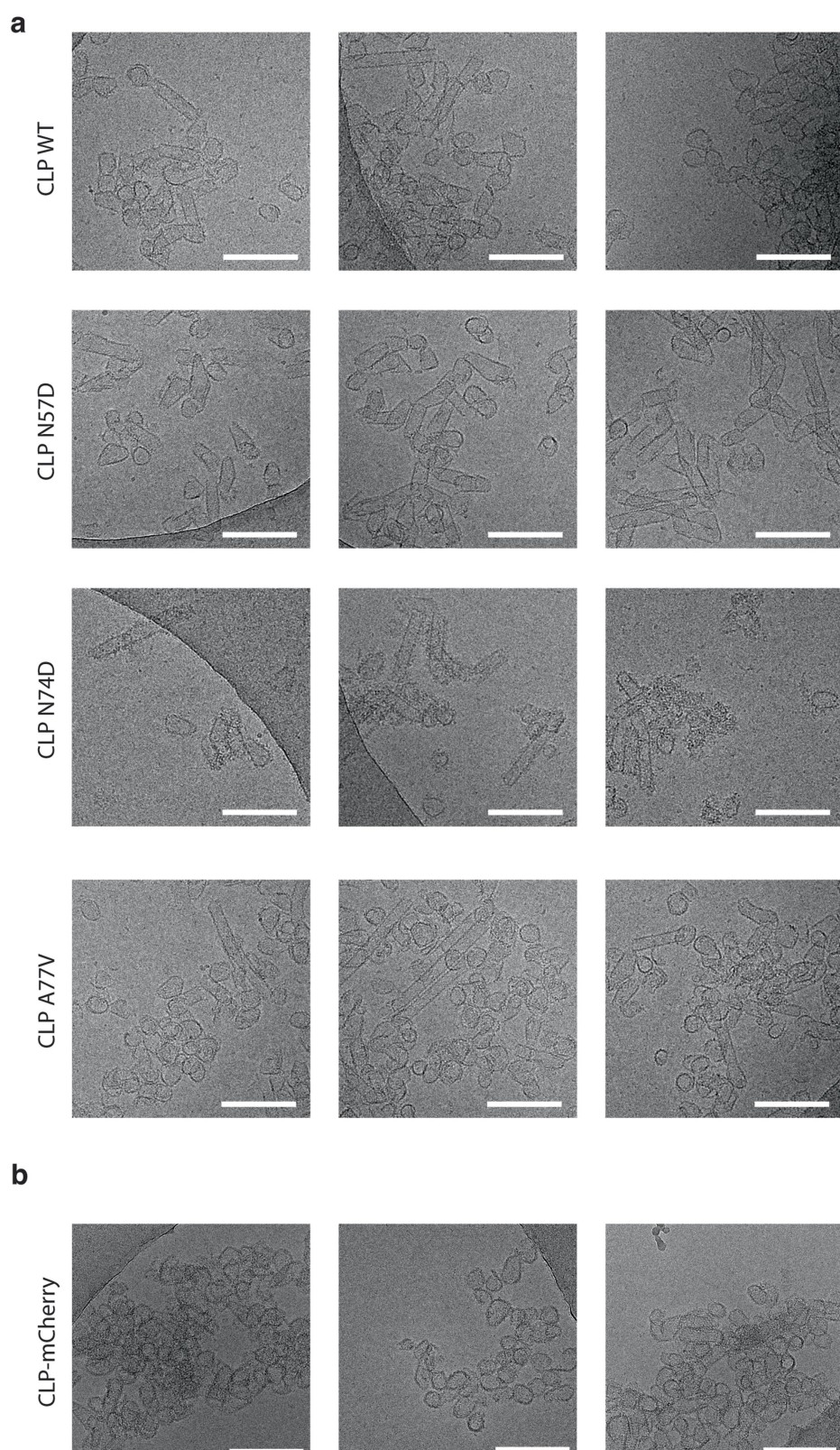

**Extended Data Fig. 1 | Cryo-electron micrographs of CLPs used in this study. a**, CLPs used for FFS. **b**, CLPs used for confocal microscopy imaging. Scale bar = 200 nm.

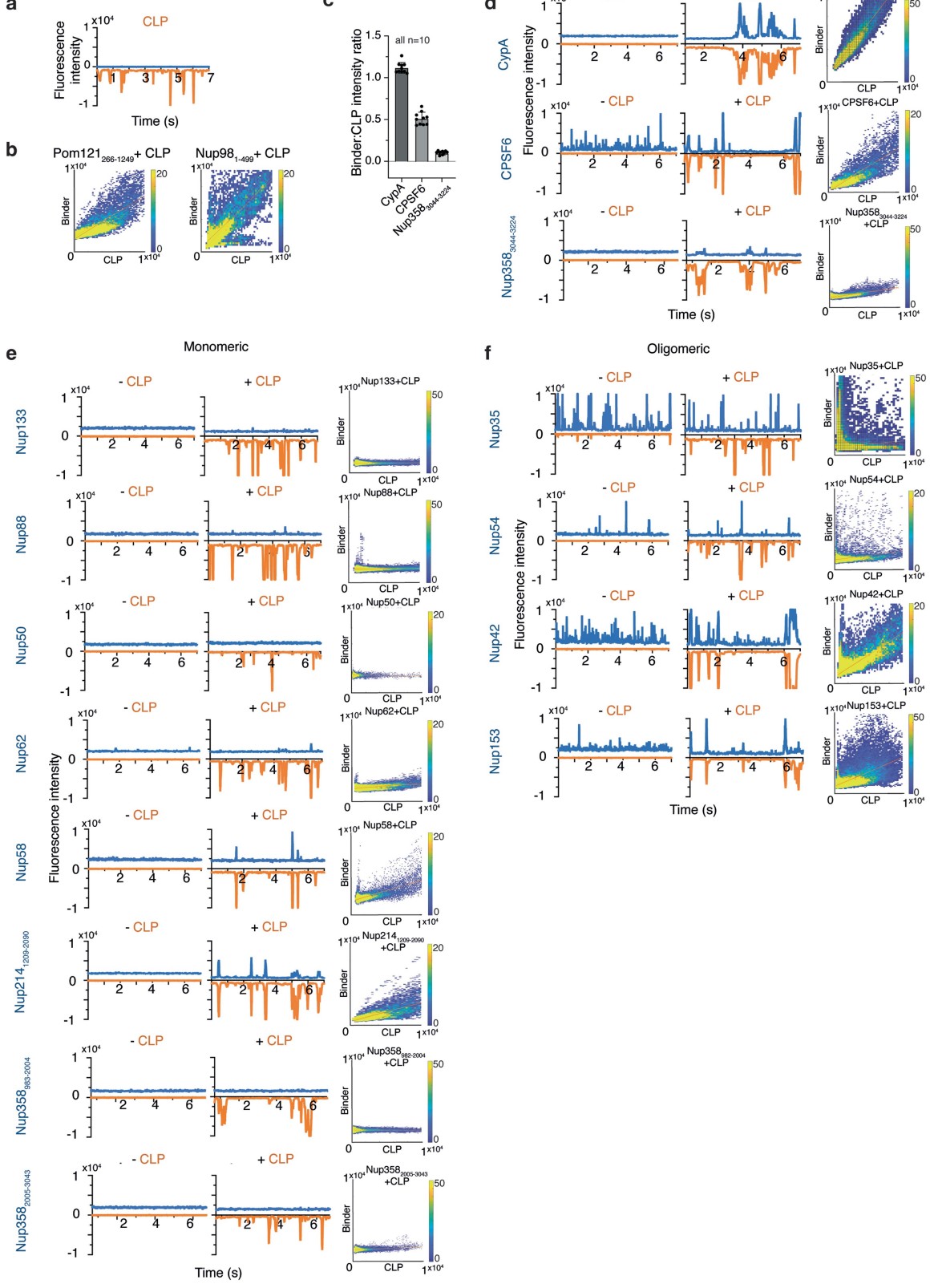

**Extended Data Fig. 2 | FFS data for CLPs and interactors. a**, FFS trace of AF568-CLP (orange). **b**, Coincidence heatmaps used to calculate NUP:CLP fluorescence intensity ratios for Pom121 and Nup98 (Fig. 1c and d). Representative FFS traces are shown in Fig. 1a and b. **d**, **e** and **f**, FFS traces for GFP-binders in the absence and presence of AF568-CLP and heat maps of binder to CLP fluorescence intensity used to calculate fluorescence intensity ratios in **c**, and Fig. 1c and d. Blue, Nup channel; orange, CLP channel. Error bars show standard deviation of the mean. N=number of scans per sample.

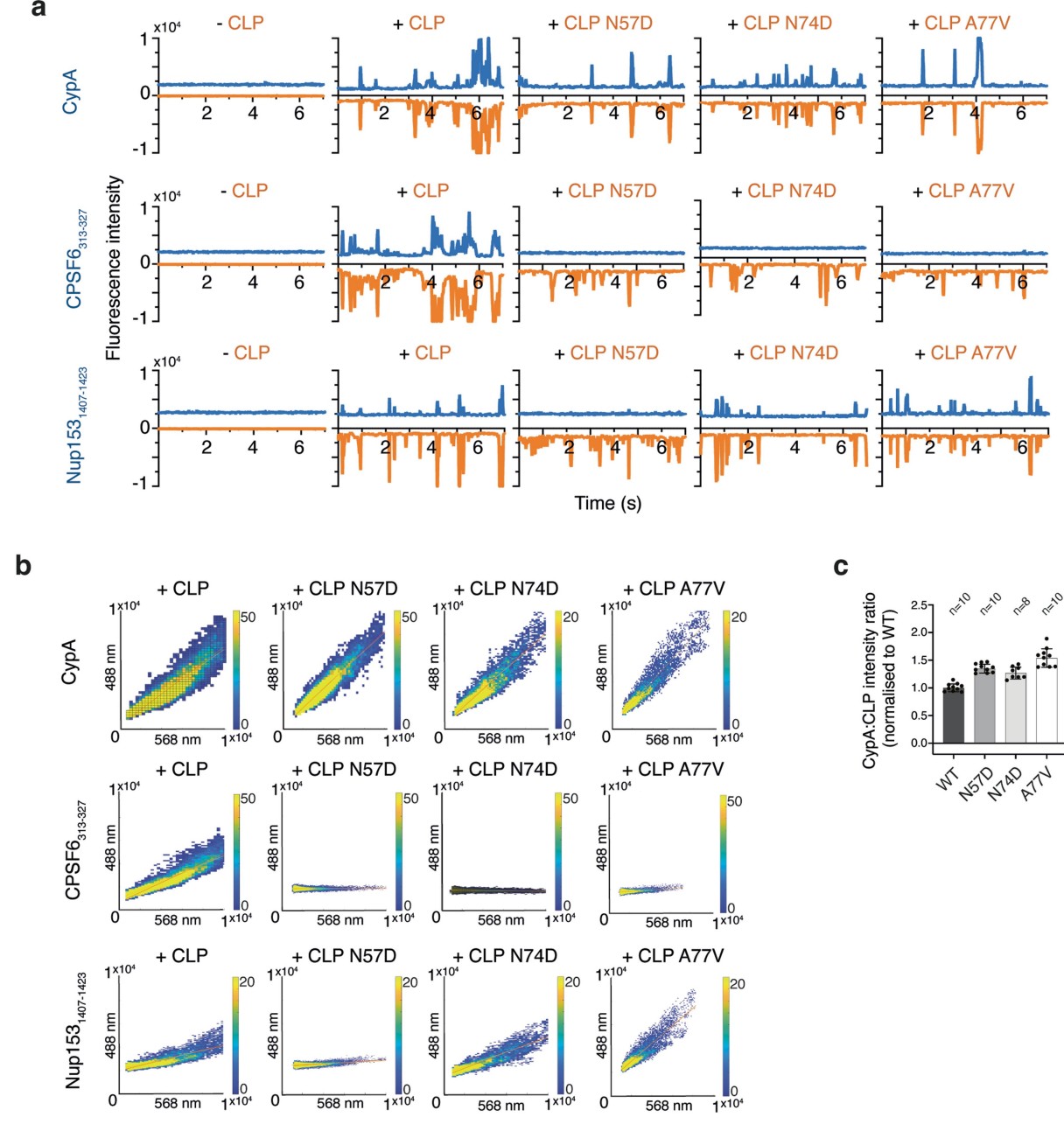

**Extended Data Fig. 3 | FFS data for known cofactors with wild-type and mutant CLPs. a**, FFS traces for GFP-controls in the absence and presence of wild-type and mutant AF568-CLPs. Blue, Nup channel; orange, CLP channel. **b**, Corresponding coincidence heatmaps used to calculate NUP:CLP fluorescence intensity ratio in **c** and Fig. 2b, c. Error bars show standard deviation of the mean. N=number of scans per sample.

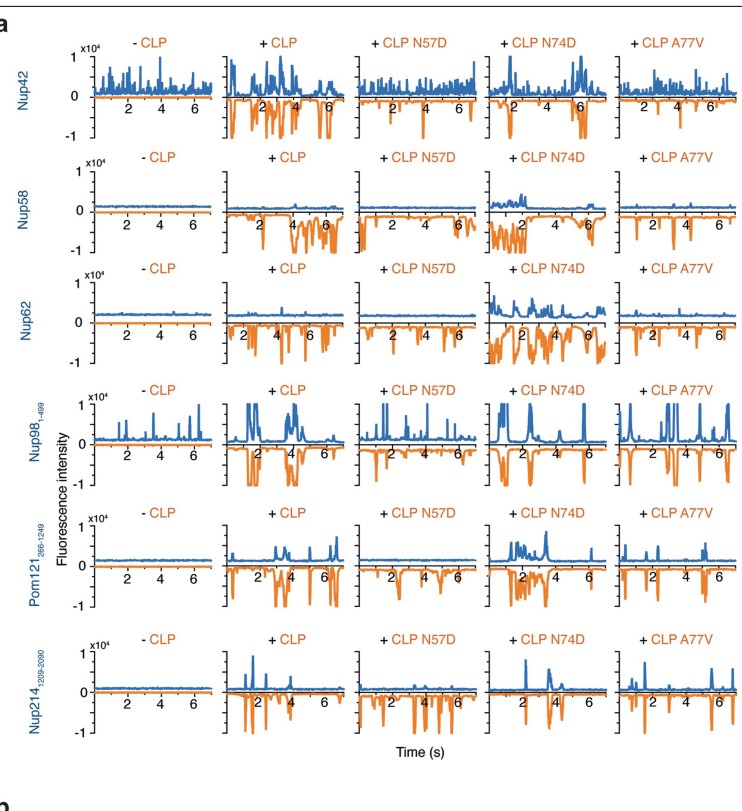

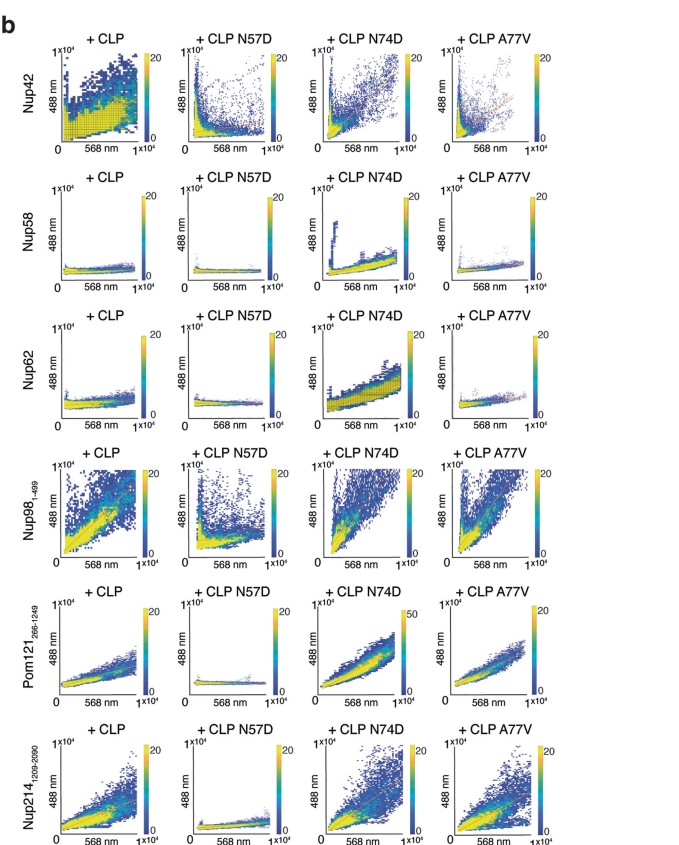

**Extended Data Fig. 4 | FFS data for Nup binders with wild-type and mutant CLPs. a**, FFS traces for GFP-Nups in the absence and presence of wild-type and mutant AF568-CLPs. Blue, Nup channel; orange, CLP channel. **b**, Corresponding coincidence heatmaps used to calculate NUP:CLP fluorescence intensity ratio in Fig. 2d, e, f, and g.

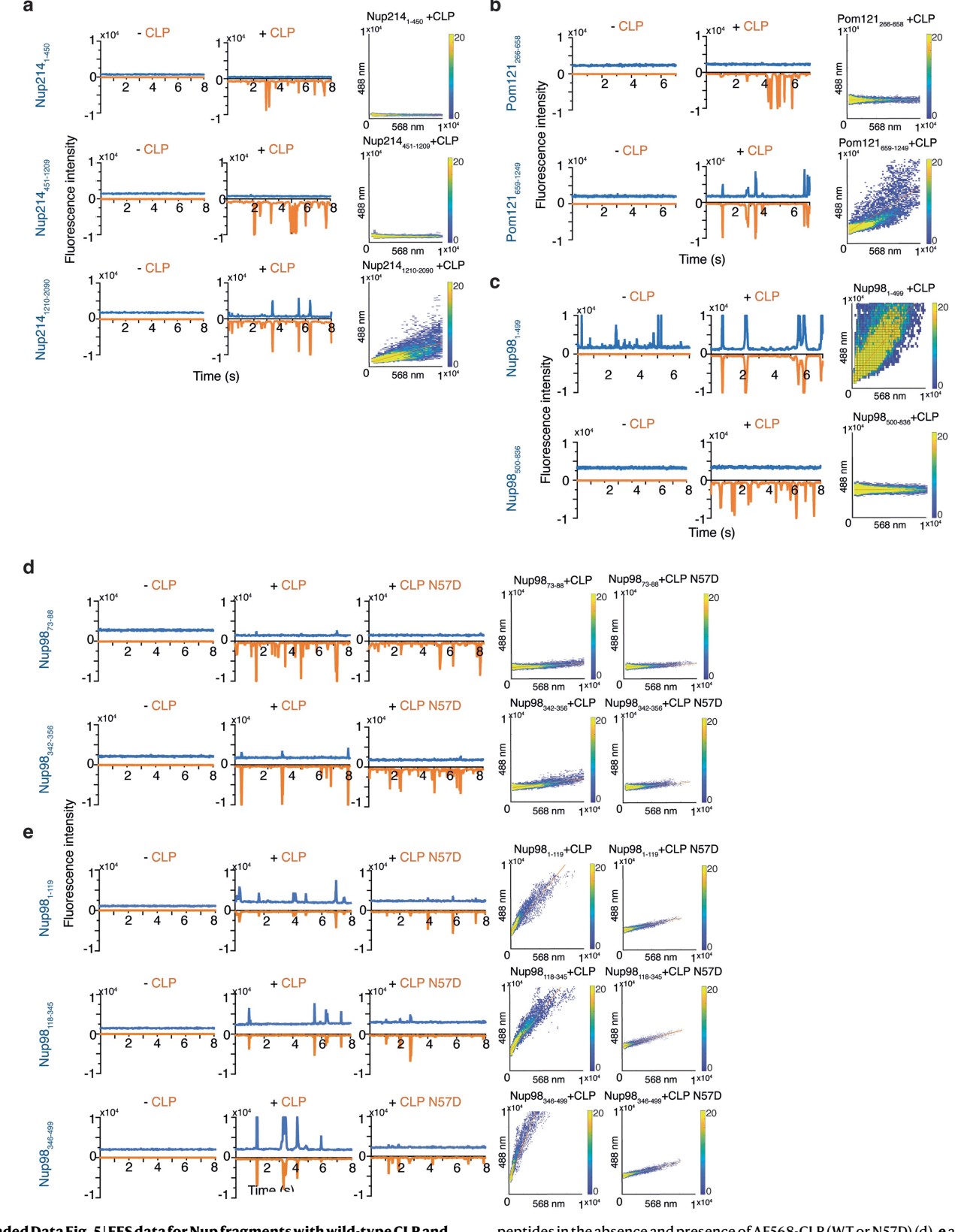

**Extended Data Fig. 5 | FFS data for Nup fragments with wild-type CLP and CLP N57D. a-d**, FFS traces and associated coincidence heatmaps for truncation constructs of GFP-Nup214 (a), GFP-Pom121 (b), GFP-Nup98 (c), and AF488-Nup98 peptides in the absence and presence of AF568-CLP (WT or N57D) (d). **e** as a-d, with mCherry2-Nup98 fragments and AF569-CLP. For all FFS traces: blue, Nup channel; orange, CLP channel.

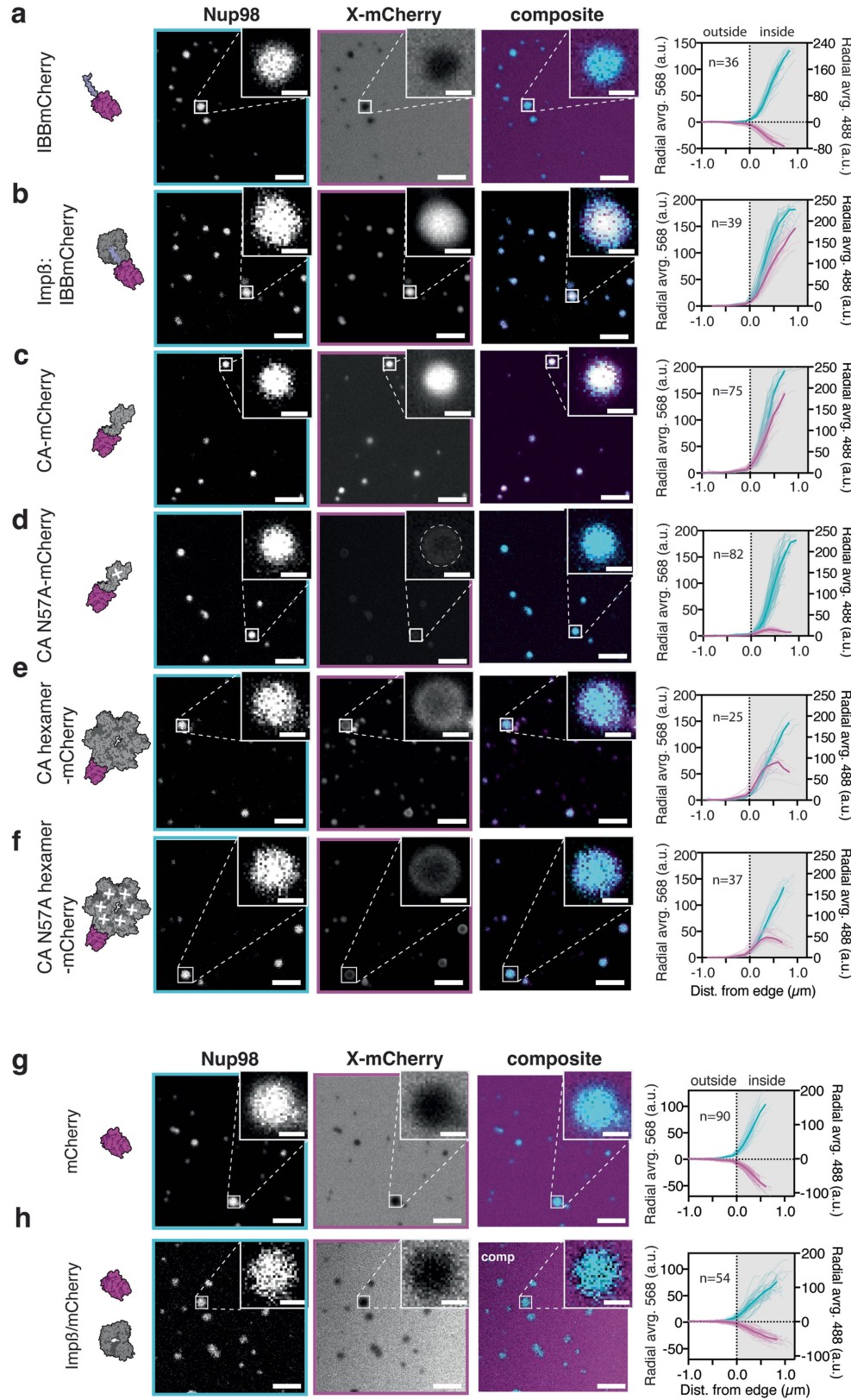

**Extended Data Fig. 6 | Confocal fluorescence microscopy (all channels) with Nup98 condensates and mCherry labelled controls or CA-constructs. a-h,** Single z-plane images (cyan, 488-Nup channel; magenta, protein-mCherry channel) of control fusion protein IBBmCherry (a), control complex Importin-β:IBBmCherry (b), unassembled CA-mCherry (c), unassembled CA-mCherry carrying FG pocket mutation N57A (d), cross-linked CA hexamer-mCherry (e), cross-linked CA hexamer-mCherry carrying N57A (f), mCherry (g) and control Importin-β and mCherry (h). Background subtracted radially averaged fluorescence intensity across condensates for 488-Nup channel (cyan) and protein-mCherry channel (magenta). Mean intensity curves shown in bold. Scale bar 5 µm (main), 1 µm (inset). N = number of condensates analysed per sample.

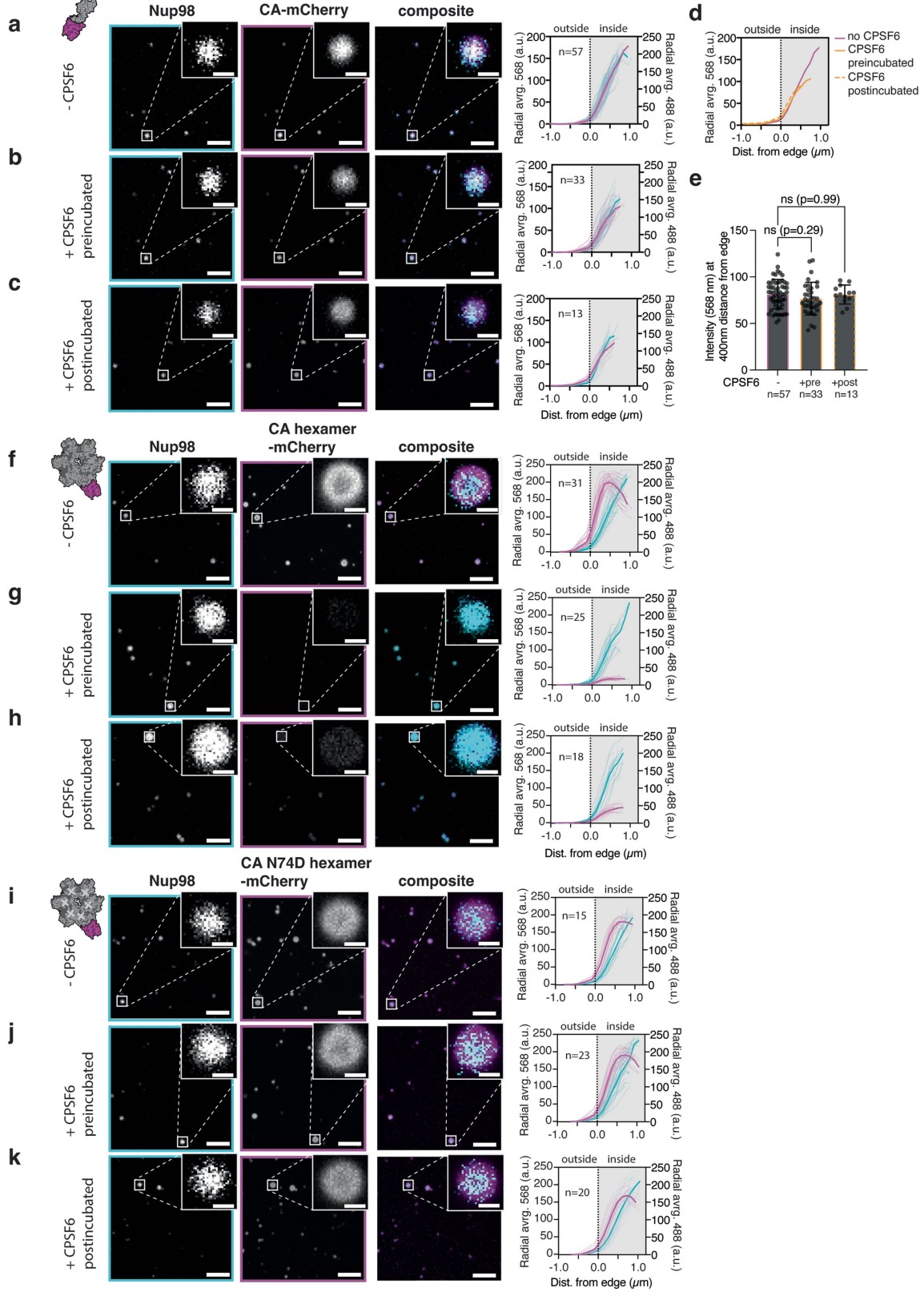

**Extended Data Fig. 7** | See next page for caption.

**Extended Data Fig. 7 | Confocal fluorescence microscopy (all channels) of Nup98 condensates and CA-mCherry, CA$_{hexamer}$-mCherry or CA (N74D)$_{hexamer}$-mCherry in the presence of CPSF6. a-c**, Single z-plane images (cyan, 488-Nup channel; magenta, protein-mCherry channel) of CA-mCherry without CPSF6 (a), with CPSF6 pre-incubated for 10 min (b), with CPSF6 post-incubated 20 min (c). **d**, Background subtracted radially averaged fluorescence intensity across condensates for 488-Nup channel (cyan) and protein-mCherry channel (magenta). Mean intensity curves shown in bold. Scale bar 5 μm (main), 1 μm (inset). **e**, Intensity values (568) at 400 nm from edge of condensates.

**f-k**, Single z-plane images (cyan, 488-Nup channel; magenta, protein-mCherry channel) of CA$_{hexamer}$-mCherry (f) or CA (N74D)$_{hexamer}$-mCherry (l) without CPSF6, with CPSF6 pre-incubated 10 min (g and j), with CPSF6 post-incubated 20 min (h and k). Background subtracted radially averaged fluorescence intensity across condensates for 488-Nup channel (cyan) and protein-mCherry channel (magenta). Mean intensity curves shown in bold. Scale bar 5 μm (main), 1 μm (inset). N = number of condensates analysed per sample. Error bars show standard deviation of the mean. Statistical test for panel e is one-way ANOVA.

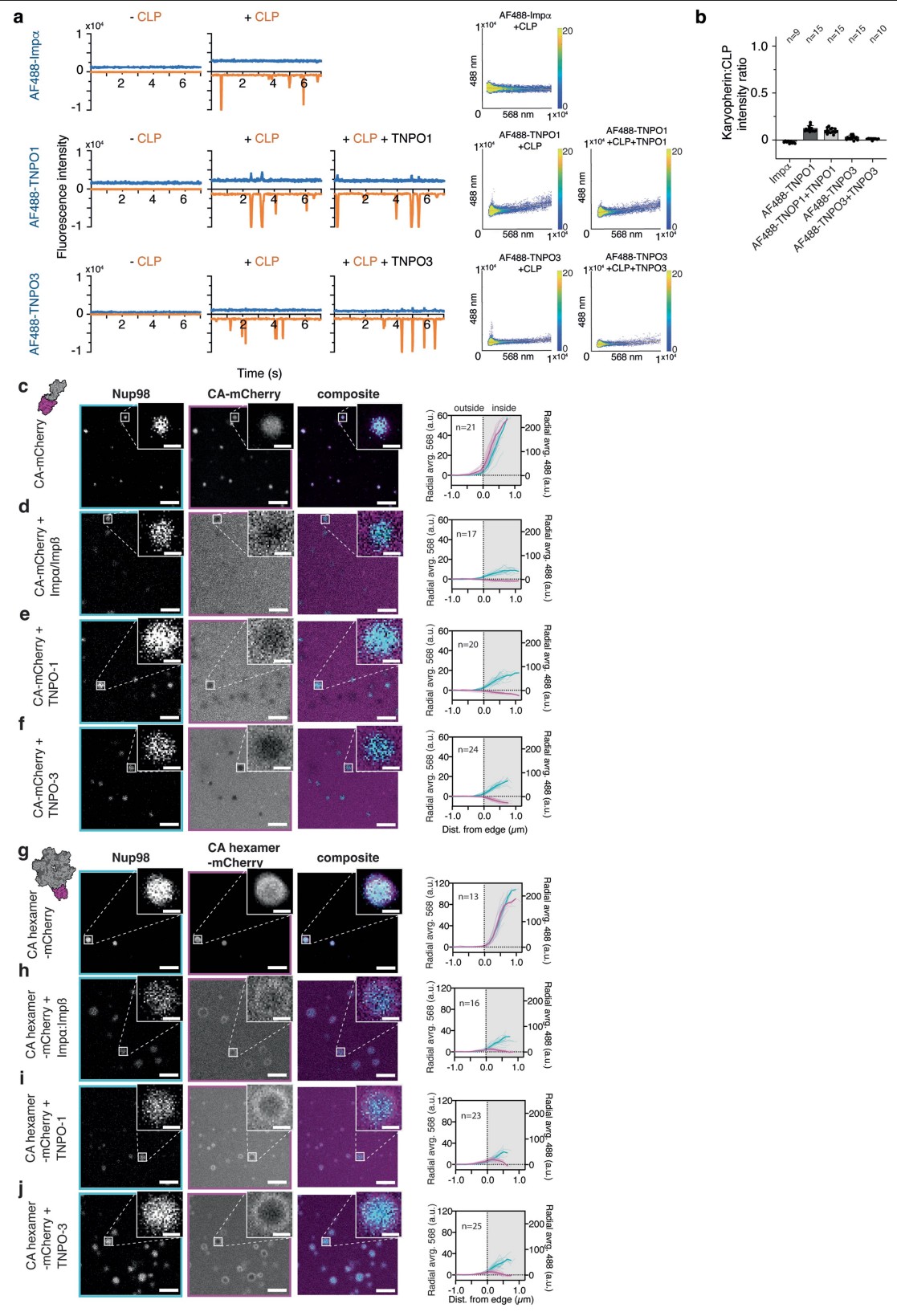

**Extended Data Fig. 8** | See next page for caption.

**Extended Data Fig. 8 | FFS of CLPs and confocal fluorescence microscopy (all channels) of CLPs and Nup98 condensates in the presence of the karyopherins Importin-α:Importin-β, TNPO-1 and TNPO-3. a**, FFS traces (left) of AF488-labelled karyopherins in absence or presence of AF568-CLP and an excess of unlabelled karyopherin; associated coincidence heatmaps (right) of karyopherin to CLP fluorescence intensity used to calculate **b**, fluorescence intensity ratios. Error bars show the standard deviation of the mean. N=number of scans per sample. **c-f**, Single z-plane images (cyan, 488-Nup channel; magenta, protein-mCherry channel) of CA-mCherry (c), CA-mCherry and Importin-α:Importin-β (d), CA-mCherry and TNPO-1 (e), CA-mCherry and TNPO-3 (f). **g-j**, Single z-plane images (cyan, 488-Nup channel; magenta, protein-mCherry channel) of $CA_{hexamer}$-mCherry (g), $CA_{hexamer}$-mCherry and Importin-α:Importin-β (h), $CA_{hexamer}$-mCherry and TNPO-1 (i), $CA_{hexamer}$-mCherry and TNPO-3 (j). Background subtracted radially averaged fluorescence intensity across condensates for 488-Nup channel (cyan) and protein-mCherry channel (magenta). Mean intensity curves shown in bold. Nup98 signal was reduced in the presence of karyopherins, suggesting that binding of karyopherins to FG-motifs reduced the compactness of the condensates. Selective properties of the condensates were still maintained (compare Extended Data Fig. 6). Scale bar 5 μm (main), 1 μm (inset). N = number of condensates analysed per sample.

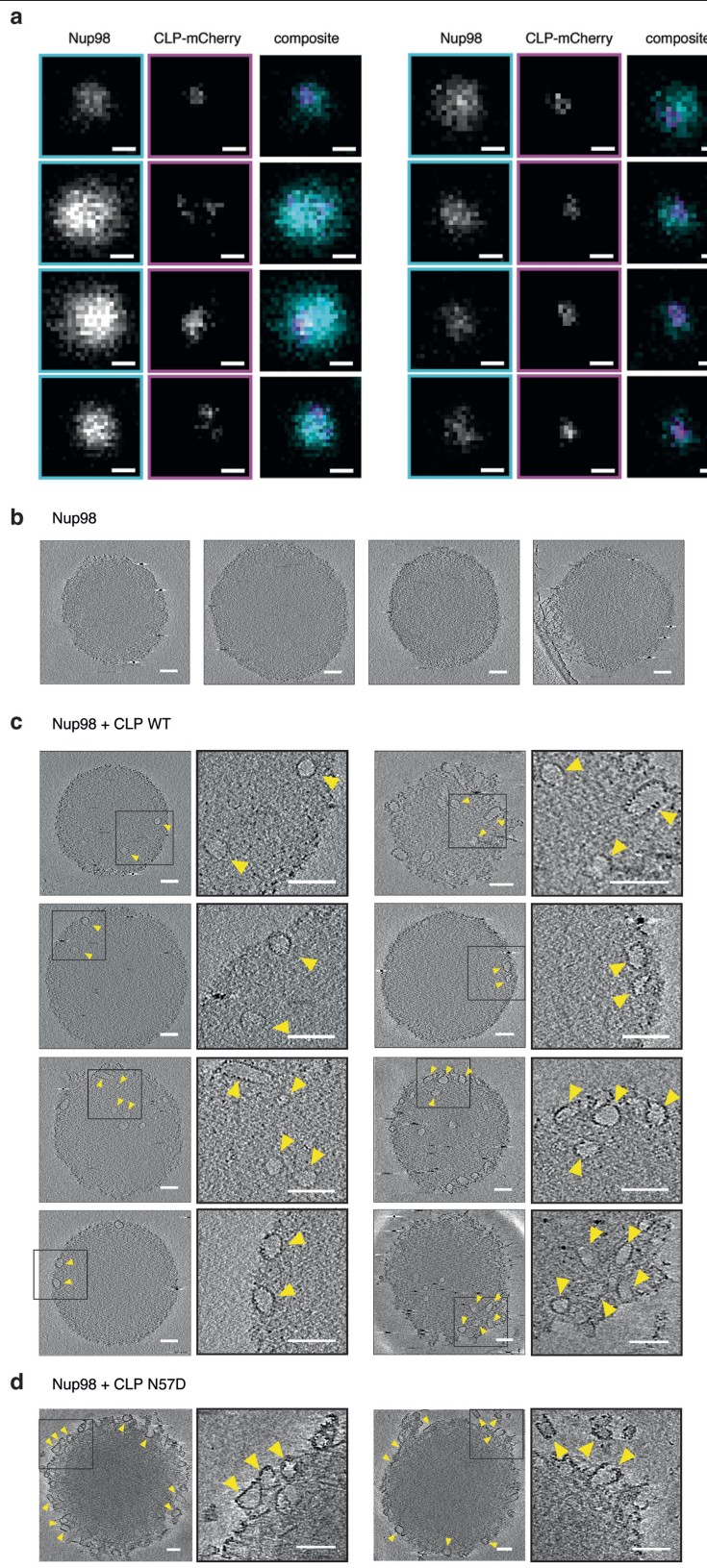

**Extended Data Fig. 9 | Confocal fluorescence microscopy (all channels) and cryo-electron tomography of CLPs bound and partitioned into Nup98 condensates. a**, Confocal fluorescence microscopy images of CLPs (puncta) recruited to Nup98 condensates. Single z-plane images for 488 nm (cyan), 568 nm (magenta) and composite. Scale bar 500 nm. **b-d**, Representative slices from cryo-electron tomograms of Nup98 alone (b), in the presence of CLP WT (c), or CLP N57D (d). Scale bars 100 nm. Tomograms are binned by 2 and displayed as an average of 5 slices with a total thickness of 5.2 nm.

**Extended Data Table 1 | Total Number of FGs and accessible FGs, calculated using AlphaFold2 RSA (relative solvent accessibility), per Nup tested in this study**

| Name | Uniprot Acc | FG per Nup | Accessible FGs (Alphafold-RSA) | Stoichiometry | Total Accessible FGs |
|---|---|---|---|---|---|
| Nup98 | P52948 | 40 | 40 | 48 | 1920 |
| Nup153 | P49790 | 29 | 29 | 32 | 928 |
| Nup214 | P35658 | 44^ | 44 | 16 | 704 |
| Nup358* | P49792 | 22 | 19* | 32 | 608 |
| Nup58 | Q9BVL2 | 14 | 14 | 32 | 448 |
| Pom121 | Q96HA1 | 24 | 24 | 16 | 384 |
| Nup54 | Q7Z3B4 | 9 | 9 | 32 | 288 |
| Nup62 | P37198 | 6 | 6 | 48 | 288 |
| Nup42 | O15504 | 12 | 11 | 8 | 88 |
| Nup50 | Q9UKX7 | 5 | 5 | 16 | 80 |
| Nup35 | Q8NFH5 | 3 | 1 | 32 | 32 |
| Nup133 | Q8WUM0 | 3 | 1 | 32 | 32 |
| Nup88 | Q99567 | 3 | 0 | 16 | 0 |

| * | NUP358 | FG per Nup | Accessible FGs (Alphafold-RSA) |
|---|---|---|---|
| | Nup358_0982-2004 | 11 | 11 |
| | Nup358_2005-3043 | 8 | 8 |
| | Nup358_3058-3224 | 3 | 0 |
| | Total | 22 | 19 |

# Reporting Summary

## Statistics

For all statistical analyses, confirm that the following items are present in the figure legend, table legend, main text, or Methods section.

| n/a | Confirmed | |
|---|---|---|
| ☐ | ☒ | The exact sample size (*n*) for each experimental group/condition, given as a discrete number and unit of measurement |
| ☐ | ☒ | A statement on whether measurements were taken from distinct samples or whether the same sample was measured repeatedly |
| ☐ | ☒ | The statistical test(s) used AND whether they are one- or two-sided *Only common tests should be described solely by name; describe more complex techniques in the Methods section.* |
| ☒ | ☐ | A description of all covariates tested |
| ☒ | ☐ | A description of any assumptions or corrections, such as tests of normality and adjustment for multiple comparisons |
| ☐ | ☒ | A full description of the statistical parameters including central tendency (e.g. means) or other basic estimates (e.g. regression coefficient) AND variation (e.g. standard deviation) or associated estimates of uncertainty (e.g. confidence intervals) |
| ☐ | ☒ | For null hypothesis testing, the test statistic (e.g. *F*, *t*, *r*) with confidence intervals, effect sizes, degrees of freedom and *P* value noted *Give P values as exact values whenever suitable.* |
| ☒ | ☐ | For Bayesian analysis, information on the choice of priors and Markov chain Monte Carlo settings |
| ☒ | ☐ | For hierarchical and complex designs, identification of the appropriate level for tests and full reporting of outcomes |
| ☒ | ☐ | Estimates of effect sizes (e.g. Cohen's *d*, Pearson's *r*), indicating how they were calculated |

*Our web collection on statistics for biologists contains articles on many of the points above.*

## Software and code

Policy information about availability of computer code

| | |
|---|---|
| Data collection | Fluorescence Fluctuation Spectroscopy data were collected using LabVIEW 14.0f1<br>Confocal Microscopy Data were collected using Zen 2.1 from Zeiss<br>CryoEM tilt series were collected using Tomography (Thermo Fisher Scientific, version 5.12.0.4776REL) |
| Data analysis | Fluorescence Fluctuation Spectroscopy data were analysed using TRISTAN v0_2 (Two Reagents Incident Spectroscopic Analysis, freely available on https://github.com/lilbutsa/Tristan<br>Averaged radial intensity profiles were obtained using the plugin Radial Profile in ImageJ Version 2.9.0/1.53t<br>Three-dimensional reconstructions from cryoEM tilt seris were generated with IMOD 4.11.19<br>Matlab Version R2019b and GraphPad Prism 9.4.1 were used for statistical analysis and data presentation |

For manuscripts utilizing custom algorithms or software that are central to the research but not yet described in published literature, software must be made available to editors and reviewers. We strongly encourage code deposition in a community repository (e.g. GitHub). See the Nature Portfolio guidelines for submitting code & software for further information.

## Data

Policy information about availability of data

All manuscripts must include a data availability statement. This statement should provide the following information, where applicable:
- Accession codes, unique identifiers, or web links for publicly available datasets
- A description of any restrictions on data availability
- For clinical datasets or third party data, please ensure that the statement adheres to our policy

The experimental data that support the findings of this study are available at Dryad with the identifier: https://doi.org/10.5061/dryad.b2rbnzsm0

## Research involving human participants, their data, or biological material

Policy information about studies with human participants or human data. See also policy information about sex, gender (identity/presentation), and sexual orientation and race, ethnicity and racism.

| Reporting on sex and gender | N/A |
|---|---|
| Reporting on race, ethnicity, or other socially relevant groupings | N/A |
| Population characteristics | N/A |
| Recruitment | N/A |
| Ethics oversight | N/A |

Note that full information on the approval of the study protocol must also be provided in the manuscript.

# Field-specific reporting

Please select the one below that is the best fit for your research. If you are not sure, read the appropriate sections before making your selection.

☒ Life sciences ☐ Behavioural & social sciences ☐ Ecological, evolutionary & environmental sciences

For a reference copy of the document with all sections, see nature.com/documents/nr-reporting-summary-flat.pdf

# Life sciences study design

All studies must disclose on these points even when the disclosure is negative.

| Sample size | This is a biophysical study on purified chemicals and did not involve predetermined sample sizes of individual participants or animals. Fluorescence Fluctuation Spectroscopy is a single molecule technique in which thousands to millions of individual events are recorded in a single experiment, with experiments repeated at least twice giving statistical power. Confocal microscopy observed tens to hundreds of individual nucleoporin condensates over multiple fields of view per experiment. Null hypothesis testing gave p values < 10^-10 demonstrating statistical power. Tens of Cryo-electron tomography tilt series were collected per sample to illustrate nanoscale capsid behavior, with the sample size chosen to optimise use of electron microscope time. |
|---|---|
| Data exclusions | For Fluorescence Fluctuations Spectroscopy data were excluded if systematic baseline variation occurred (indicating instument drift and/or misalignment). For confocal microscopy analysis, condensates too small to appropriately analyse were not considered, intensity values which significantly differed from the average value were discarded |
| Replication | Results were replicated at least twice. |
| Randomization | Randomization was not relevant to the study of capsids interacting with nucleoporins as this is a biophysical study on purified chemicals, and does not involve individual human or animals that require randomization following assessment of study eligibility. |
| Blinding | Randomization was not relevant to the study of capsids interacting with nucleoporins as this is a biophysical study on purified chemicals, and there are no individual participants that can be influenced by information regarding the experiment. |

# Reporting for specific materials, systems and methods

We require information from authors about some types of materials, experimental systems and methods used in many studies. Here, indicate whether each material, system or method listed is relevant to your study. If you are not sure if a list item applies to your research, read the appropriate section before selecting a response.

## Materials & experimental systems

| n/a | Involved in the study |
|-----|----------------------|
| ☒ ☐ | Antibodies |
| ☐ ☒ | Eukaryotic cell lines |
| ☒ ☐ | Palaeontology and archaeology |
| ☒ ☐ | Animals and other organisms |
| ☒ ☐ | Clinical data |
| ☒ ☐ | Dual use research of concern |
| ☒ ☐ | Plants |

## Methods

| n/a | Involved in the study |
|-----|----------------------|
| ☒ ☐ | ChIP-seq |
| ☒ ☐ | Flow cytometry |
| ☒ ☐ | MRI-based neuroimaging |

## Eukaryotic cell lines

Policy information about cell lines and Sex and Gender in Research

| | |
|---|---|
| Cell line source(s) | HEK293T cells were obtained from ATCC |
| Authentication | Live cells were not used in this study and were only used as a source of specific cDNAs, which were verified by Sanger sequencing |
| Mycoplasma contamination | None |
| Commonly misidentified lines (See ICLAC register) | N/A |

## Plants

| | |
|---|---|
| Seed stocks | N/A |
| Novel plant genotypes | N/A |
| Authentication | N/A |

