## [Peer Review File · Nature]

Manuscript Title: The HIV capsid mimics karyopherin engagement of FG-nucleoporins

Reviewer Comments & Author Rebuttals

Reviewer Reports on the Initial Version:

Referees' comments:

Referee #1 (Remarks to the Author):

Dickson and coworkers report a groundbreaking study of nucleoporin interactions with the HIV 1 capsid and their role in nuclear entry. The authors used fluorescence fluctuation spectroscopy to study binding of recombinant Nup fusion proteins expressed in vitro to capsid-like particles (CLPs). Specific binding of Nups to CLPs was demonstrated using CA mutants that selectively alter Nup153 binding and CPSF6 binding. They demonstrated penetration of monomeric and hexameric CA proteins into condensates formed by shock dilution of a purified Nup98 protein, which contains a high density of FG repeats. CA accumulation in the condensates was prevented by the N57A substitution in the FG binding pocket in CA and by addition of a CPSF6 peptide that binds to the pocket. The latter result is intriguing considering a model that nuclear CPSF6 aids in releasing the viral core at the nuclear side of the pore by coating the capsid. Lastly, the authors show that CLPs accumulate within Nup98 condensates.

The study is highly innovative and advances our understanding of how the intact HIV-1 core can enter the nucleus. It will be of substantial interest to the fields of virology and cell biology. The experimental data are of high quality, with appropriate numbers of replicates and statistical analysis.

The experiments were all performed with recombinant proteins, so the conclusions regarding nuclear entry are largely based on the effects of the mutations and would be strengthened by testing mutant CLPs in the assay shown in Fig. 4. Nonetheless, the conclusions are well reasoned. The assays also create opportunities for additional studies in this field. The manuscript is clearly written and appropriately references the prior work in this field.

The study could be improved by addressing the following:

1. How does the Nup binding specificity demonstrated in Fig. 1 relate to the reported effects of specific Nup depletion on HIV-1 infection? This can be summarized in a graph or table.
2. In Fig. 4, assays of mutant CLPs and inhibitors would help to validate the biological relevance of the result. For example, does the N57D mutation prevent entry of CLPs into Nup98 condensates? Do mutations elsewhere in the capsid that also reduce nuclear entry, e.g. E45A, affect CLP entry into condensates?
3. Does PF74 inhibit CLP entry into condensates at a concentration that inhibits nuclear entry but not reverse transcription?
4. Does addition of the CPSF6 peptide promote exit of CLPs from Nup98 condensates?
5. Do native HIV-1 cores enter Nup98 condensates?

Technical matter:

The authors employed N57D mutant CLPs in their experiments but the mutant monomeric and hexameric CA proteins were N57A. Why the difference?

Minor:

The title may suggest that the problem of HIV nuclear entry is now solved. I suggest something more nuanced, e.g., "Involvement of karyopherin mimicry in HIV nuclear entry"

Referee #3 (Remarks to the Author):

This manuscript proposes a new model for HIV infection by which HIV capsids enter the nuclear pore using a similar mechanism to that of karyopherins. Through in vitro experiments, the authors suggest that HIV capsids, and therefore other lentiviruses, can breach the diffusion barrier of the nuclear pore complex through specific FG repeat interactions, and therefore is efficient even in post-mitotic cells. Typically, virus incorporation into a genome occurs upon mitosis, but the proposed model indicates how HIV and lentiviral packages persist despite division.

Using fluorescence fluctuation microscopy (FFS) and confocal microscopy on in vitro systems, the authors give insight into how HIV directly interacts with components of the nuclear pore complex. Overall, the "karyopherin mimicry" model is an interesting concept and addition to the current thinking about how viruses transport across the nuclear pore. This and the accompanying manuscript are important studies and should be published after addressing the following comments.

Comments:

1. The "dissection" of the NPC seems inconsistent. For proteins that did not express fully, only the FG domain was used. However, size is thought to sometimes play a role in binding affinity or uptake and recruitment. Perhaps a good control would be to compare across all FG domains only and control for size, especially since another argument made was that size does not play a contributing factor in HIV penetration of the nuclear pore. If this were true, then the difference between the wild-type protein and just FG domain would be minimal.
2. The authors choose Nup98 as their model system due to its important role in nuclear pore integrity and FG rich structure. They find that it binds capsid protein (CA) the best out of their Nup screen and test the selectivity properties and mimicry of their Nup98 condensate model system. However, it is unclear to me how we can be sure that this shows actual penetration of CA through the nuclear pore. Certainly, capsid particles partition into Nup98 condensates, but in vivo Nup98 and other components of the nuclear pore can be pulled out of the NPC to nuclear bodies or to assist in RNA transport. Thus, if we imagine this in a living system, would this show capsids in the cytoplasm dynamically recruited to Nup98 and other FG domains in the nuclear pore, and subsequently entering the nucleus? Or would we instead see that wherever Nup98 or other top binders from this study are, the capsids follow? Surely these are predominantly located within the NPC, but to eliminate uncertainty of the model being proposed, this might need to be confirmed.
3. It may be useful to pick another promising Nup (i.e. other than Nup 98) from the initial screen that

has rich FG content but differing function from Nup98 to confirm these experiments with another component and further validate such claims.

4. Throughout the manuscript I was left wondering if there is a way to show this in dynamic fashion. For instance, it is alluded to that this happens over time and then the HIV capsid is recycled and the process continues to then infect cells. Is there a way to show that this model not only holds by showing partitioning patterns, but also that, in vitro or in vivo, it holds over time if we watch the capsids move and “breach” the condensates?

Referee #4 (Remarks to the Author):

This manuscript by Dickson et al. claims the main component of the HIV capsid protein (CA) shows a karyopherin-like behavior, being able to establish multivalent, weak and specific interactions with the FG repeat domains in the NPC permeability barrier. This direct interaction would be enough to mediate the overcome of the permeability barrier, allowing the viral capsid to traverse the NPC and reach the inside of the nucleus in non-dividing human cells. They show how monomeric and oligomeric CA directly interacts with several flavors of FG domains in vitro and that such interaction is dependent on the integrity of the CA FG-binding pocket; The authors then use in vitro reconstituted FG domain condensates to verify that capsid-like arrangements of CA effectively interact with the FG components of the condensates and could actually even penetrate these condensates for distances that would account for the length of the NPC central channel. The interaction and merging of the viral-like particles with the condensates does not destabilize the CA oligomers and does not require the presence of nuclear transport factors, nor it is regulated by RanGTP. The authors propose that the karyopherin-like arrangement of CA into the capsid lattice provides the capsid with self-translocating properties, explaining the ability of the virus to access the nucleus of non-dividing cells.

However, a great deal has been recently published revealing an already significant understanding of how the HIV capsid traverses the NPC, and the manuscript provides only a very limited conceptual advance, compared with what is already known from multiple other labs. It is already well understood that CA interacts directly with FG domains as shown by a number of groups, e.g. Kane et al., 2018, and reviewed e.g. in Guedan, 2021; in fact, even structures showing exactly how CA binds FG repeats have been published by other groups (e.g. Price et al., 2014; Stacey et al., 2023), where a specific pocket binds to the FG repeat. It is unsurprising that mutation of this pocket prevents this interaction. Thus, we already know the molecular recognition mechanism by which the capsid interacts with FG repeats to enter the NPC. CA is well understood to interact with a host of other cellular proteins that mediate interaction with the NPC. The FG binding pocket is known to bind several other non-Nup proteins such as CPSF6, and Sec24C. CA also interacts with the nuclear transport factor TNPO3 (e.g. Zhou et al., 2011), and with the nucleoporin Nup358 not through its few FG repeats but directly with its cyclophilin-like domain. Although the authors acknowledge these interactions are known, they do not account for all these other host co-factors that have been shown to be involved in the actual mechanism of capsid entry to the NPC. Also, given the known dynamic nature of these interactions, the argument that their addition to the entire capsid would form a shell too big to pass through the NPC is an unproven assumption, with many other possible

arrangements and substoichiometric associations allowing for this passage, in agreement with the fact that these other proteins are known to be involved in capsid passage into the NPC. There is evidence that structural flexibility of the intact CA is required for entry, although whether this is linked to disassembly or to distortion is unresolved {Guedan, 2021, 34543344}, discussed below. As stated by Guedan, 2021: “thus, the viral core should be considered as a dynamic structure that binds numerous cellular proteins on its path through the NPC”. This contrasts rather with the picture presented in this manuscript.

The other key questions in the field are also left unaddressed by this work. These include: what is the sequence of events, does the NPC have to dilate or be otherwise re-structured in any way, what do the nuclear basket, Nup88 complex and cytoplasmic filaments do, (how) are viral entry to the NPC and uncoating coupled, and it is unclear if intact capsids go all the way through the NPC into the nucleus or they disassemble during passage through the NPC. Zila et al., 2021 showed that the capsid orients narrow end first towards the NPC and that the NPC may dilate as the capsid enters to accommodate it, pointing to an organized mechanism of entry and so interaction with the NPC; their EM analyses also showed that the capsids appeared to undergo major remodeling as it enters the NPC, an observation supported by other groups' work. None of these key steps are addressed here. Thus, at the very least, the model presented here – based on all this other work – is undoubtedly very incomplete.

The study also contains significant flaws, causing the conclusions not to be fully supported by the data presented here. Taken together, this work is not suitable for publication in Nature at this stage. My other main general concerns are the following:

1. The authors work with the assumption that FG macroscopic condensates recapitulate the properties and behavior of the nanoscale NPC permeability barrier, when this is not a generally accepted concept, deriving as it does from in vitro studies such as these. This assumption and this in vitro system has been contradicted in a recent publication in Nature itself (Yu et al., 2023) and in BioRxiv paper (Kozai et al., 2023), showing that the behavior of FG domains in condensates does not recapitulate the state and properties of native, in situ, NPC anchored FG Nups. Thus, many of the conclusions in this paper – based on this other work – could be incorrect. The authors fail to address this serious issue.

2. In my opinion, the most parsimonious and better supported explanation for the data presented here would be that the CA capsid lattice might mediate the docking of the HIV capsid into the NPCs, but none of the manuscripts show compelling evidence that the HIV capsids or even the smaller 40 nm capsid-like spheres do traverse the NPC in vivo. The authors themselves discuss this option as a side idea, instead of considering it their main conclusion, with the Kap-like trafficking through the NPC as a potential, but not proven, possibility. The incorporation into FG condensates is no proof, any more than the ability to make omelets from eggs proves that chickens arise from omelets. For the reasons stated below, and it would be nice to show it in an actual NPC.

3. Karyopherins establish multivalent, weak and transient interactions with the FGs in the NPC. It is not clear from the evidence presented here that such is the mode of interaction of the CA multimers in the HIV capsid, so without further data and controls, the identification of CA to a karyopherin,

although evocative, might not be an accurate description of the actual behavior of the CA lattice and could be quite misleading to a general reader.

4. It is also not clearly stated throughout the manuscript the actual time length of the experiments performed with these condensates, which are, acknowledged by the authors themselves, prone to “aging”, gelation and formation of potentially aberrant beta-structures, a circumstance that is hard to believe happens in a living NPC permeability barrier. How would the “aging” of the condensates affect the different experiments shown in these manuscripts?

5. It is worrying that the presence of karyopherins massively outcompete the CA protein or hexamer in its interaction with FG condensates. That would precisely be a closer situation to what a HIV capsid would find in a living NPC: hundreds of copies of the different types of karyopherins and other nuclear transport factors (carrying RNPs, Ran, etc) also interacting and passing through the NPC, together with their cargoes and other molecules, forming a heterogeneous, complex, and dynamic nano-environment. The capsid would never find pure and homogeneous FG condensates. How do the authors reconcile these apparently conflicting observations?

6. Considering that the authors are trying to establish an analogy between the HIV capsid and karyopherins, it is surprising that they did not measure the interaction between any of the importins shown in Fig. S3 and the Nup98 condensates using their FFS measurement setup. Such an additional control/example would be desirable to be able to establish a more truthful comparison between the behaviors of CA and karyopherins.

7. The authors reason that, because the binding of a single FG is below the limit of detection in their FFS measurements, a “weak, but FG specific, interaction enhanced by avidity” is suggested. However, another interpretation would be that their result might just probably reveal an intrinsic limitation of their method and not necessarily a behavior analogous to a karyopherin. If that’s the case, their analogy is unsupported and highly speculative.

8. In figure S3-1, I agree that the N57A mutation has an effect on the behavior of the constructs, but, unless I am misinterpreting something, I can clearly see in the pictures that the signal from the mCherry channel significantly concentrates at the rim of the condensates even in the N57A mutant. Would this indicate CA has a certain avidity for protein condensates independently of its FG-binding pocket, but it is not able to move across those condensates? Could the authors include some kind of control to show that CA and its hexamer do not have just a non-specific avidity to protein condensates?

9. In the section “FG-repeat domains are specifically recruited to the FG-binding pocket on the HIV capsid” to formally demonstrate this point it would be interesting to show that a phenylalanine mutant in the FG repeats does not interact with CA, completely discarding that the interaction could be helped or be mediated by other residues.

10. Line #95 “FG-repeat proteins are known to phase-separate”, please add here the caveat “when overexpressed and are not assembled into the NPC”, otherwise you are misleading the reader.

11. Line #192, please correct “an affect” with “an effect”

Referee #5 (Remarks to the Author):

Dickson and colleagues present a powerhouse, biochemistry-based study that significantly informs the mechanism of HIV nuclear transport. Leveraging in vitro translation and fluorescently-labeled capsid like particles (CLPs) made from purified CA protein, they detect Nup-CA binding via fluorescence fluctuation spectroscopy (FFS). From this they rank ability of various human Nups to bind CLPs. They furthermore leverage Nup98 condensates to gauge penetration distance of various substrates (CA, karyopherins), culminating in rather clear images of internalized CLPs. Much of the work is carefully controlled by the use of CA/CLPs mutants as well as peptide competition. The overall conclusion that the HIV capsid mimics human karyopherin behavior to gain NPC entry/passage is novel and will predictably be of interest to wide, general readership. The study overall was quite impressive.

There is no virology data provided. Nevertheless, appropriate prior studies were in general cited, and the writing well-framed the important advancements of this in vitro work to the larger body of literature.

The paper will be improved by incorporating some pertinent references and by providing additional transparency in terms of proteins analyzed and related images. Given the full in vitro nature of the work, it is critical to fully inform readers of analyte quality.

Please note the paper lacked page and line numbers – always a pain for reviewers. I assigned 1 to first page.

1) Was Nup358_3058-3224 protein tested by FFS? Due to the C-terminal Cyp-homology domain, one would predict binding, and it would be informative to compare this to the utilized CypA control. Perhaps I missed this data. If you don't have it, no need to do the experiment. In this case, either remove Nup358_3058-3224 from Table S1 or add footnote to say it was only analyzed by AlphaFold and not studied as protein.

2) The paper uses numerous proteins, many made via in vitro translation. Although Table S1 provides some information, the paper will be improved by providing additional transparency. Either greatly expand Table S1 or add separate Table that lists all proteins and peptides used in the paper (at construct level). Although Uniprot Acc is useful, additional work is required to track down specific isoforms, which applies for many of the proteins. Please add a separate column indicating aa content of corresponding full length (FL) proteins.

3) Please add SDS-PAGE images of all in vitro translated proteins. Although I realize FFS affords internal label proximity control, it nevertheless is important to inform readers of analyte quality. If the fluorescent proteins are not discernable via Coomassie blue, please perform appropriate western blots to highlight these alongside Coomassie images.

- 4) Along these lines, please provide negative stain images of assemblies CLPs (WT, N57D, N74D, A77V; Fig 2b-g).
- 5) Intro first paragraph. It is overstatement to say all genera outside lentiviruses require M phase. While this is well-established for Mo-MLV, other gammaretroviruses as well as betaretroviruses are claimed to effectively infect growth arrested cells. While these studies may be one-offs, more careful work with alpharetroviruses indicate these lay somewhere between HIV and Mo-MLV. Please rephrase to more accurately reflect the literature.
- 6) Page 2 first full paragraph “cleavage and polyadenylation specificity factor 6”
- 7) Ending sentence, refs 28, 29 established that elements of an adjacent CTD contribute to the FG binding pocket.
- 8) Three lines from bottom. Pom121C was also studied as fragment; please add. Table S1 reports Nup214 coordinates as 1209-2090. Please fix.
- 9) Page 3 bottom full paragraph line 3. Truncations correct? Nup42 in these experiments appears to be FL.
- 10) “These observations may explain why...” overstates. N74D is defective for macrophage infection (ref 68).
- 11) Page 4 line 6, place (Fig 2j and S2-3) at end of sentence.
- 12) Normalized FFS data indicates CypA the most robust binder. Please amend last sentence of second full paragraph.
- 13) The N57A mutation had less of an affect on CA hexamer versus CA-mCherry radial avg (Fig 3c-f; S3-1c-f). Although perhaps maybe 2-fold, page 5 line 4 describes the CA hexamer-mCherry affect as “significantly reduced”. Please calculate and report these p values.
- 14) Page 5 second full paragraph line 3 cites 46, 47. My reads indicate while ref 47 showed evidence for direct binding, ref 46 claimed a genetic as compared to biochemical interaction.
- 15) Many labs have measured HIV infection in Nup knockdown cells, and results of many of these studies are fully consistent with this current work. Discussion line 2, “cannot be easily manipulated”
- 16) Line 5, “any molecular details” is disingenuous. E.g., an important Nup153 FG motif was identified 10 years ago (refs 27-29, 35). Moreover, a recent NSMB paper partially reconstituted HIV nuclear import using Nup-docked DNA origami (PMID: 36807645). While the current paper surely adds important dimensionalities, the NSMB paper could be cited. Moreover, this paper revealed context dependent CA-Nup62 binding. It would be informative if you could frame your own Nup62 results to what’s been previously published. Afterall, it’s the only Nup that fell outside the subset

shown to directly engage karyopherins (bottom page 6).

17) Paragraph 4 line 5 should cite PMID: 26586435, the first to show this.

18) Bottom paragraph line 2: the predominant CPSF6 isoform harbors a single FG. While CPSF6 may phase separate (this has not been strictly demonstrated for the FL protein), it would seem incorrect to refer to this as “FG-phase”.

19) Please fix ref 39 citation.

20) Fig S3-3e lacks x-axis indicators.

Author Rebuttals to Initial Comments:

Below is a detailed point-by-point summary of our responses to each of the referees. For ease of reading and cross-referencing, we have signposted each referee comment by the referee number and comment in the order they were presented (eg. '1-2' is Referee #1, comment 2). Our responses are in blue text, and we have also highlighted the manuscript in blue text where it has been revised.

Referee #1:

Dickson and coworkers report a groundbreaking study of nucleoporin interactions with the HIV 1 capsid and their role in nuclear entry. The authors used fluorescence fluctuation spectroscopy to study binding of recombinant Nup fusion proteins expressed in vitro to capsid-like particles (CLPs). Specific binding of Nups to CLPs was demonstrated using CA mutants that selectively alter Nup153 binding and CPSF6 binding. They demonstrated penetration of monomeric and hexameric CA proteins into condensates formed by shock dilution of a purified Nup98 protein, which contains a high density of FG repeats. CA accumulation in the condensates was prevented by the N57A substitution in the FG binding pocket in CA and by addition of a CPSF6 peptide that binds to the pocket. The latter result is intriguing considering a model that nuclear CPSF6 aids in releasing the viral core at the nuclear side of the pore by coating the capsid. Lastly, the authors show that CLPs accumulate within Nup98 condensates.

The study is highly innovative and advances our understanding of how the intact HIV-1 core can enter the nucleus. It will be of substantial interest to the fields of virology and cell biology. The experimental data are of high quality, with appropriate numbers of replicates and statistical analysis.

We thank the referee, and we appreciate reading that our data and analyses are considered high quality.

The experiments were all performed with recombinant proteins, so the conclusions regarding nuclear entry are largely based on the effects of the mutations and would be strengthened by testing mutant CLPs in the assay shown in Fig. 4. Nonetheless, the conclusions are well reasoned. The assays also create opportunities for additional studies in this field. The manuscript is clearly written and appropriately references the prior work in this field.

We thank the referee for stating that they expect that our work will seed additional studies in the future. This was our intention. (The CLP comment is addressed in point 1-2 below).

1-1 How does the Nup binding specificity demonstrated in Fig. 1 relate to the reported effects of specific Nup depletion on HIV-1 infection? This can be summarized in a graph or table.

We thank the referee for this suggestion. We have summarised prior Nup depletion and Nup:CA interaction studies in a new table (Table S1). While there is variation between screens, Nup98, Nup153, Nup214, and Nup358 are strongly represented as having effects in depletion studies, and CA interaction by binding studies. We comment on this in the discussion.

1-2 *In Fig. 4, assays of mutant CLPs and inhibitors would help to validate the biological relevance of the result. For example, does the N57D mutation prevent entry of CLPs into Nup98 condensates? Do mutations elsewhere in the capsid that also reduce nuclear entry, e.g. E45A, affect CLP entry into condensates?*

N57D does prevent CLP entry into Nup98 condensates. We thank the referee for the suggestion, and have incorporated the supporting cryoEM data to figure 4, which we think greatly strengthens the manuscript. Inhibitors have the confounding effect of causing overassembly (effectively aggregation) of CLPs *in vitro* preventing reliable interpretation of such an experiment.

E45A has a complex phenotype. The E45A capsid is hyperstable, which could influence NPC transit (if the capsid is required to remodel or retain a degree of structural plasticity during transit) and likely influences uncoating (PMID 12692245). It retains some dependence on Nup153, suggesting it still uses the NPC (PMID 21593146). While infectivity is reduced by about 1-log upon aphidicolin treatment, it is fully restored with cyclosporine treatment (PMID 19625401), suggesting that this mutant can enter the nucleus via the NPC, but that CypA plays a role. To avoid conflating these effects, we have not studied this mutant in the current manuscript. We anticipate that it will feature prominently in future studies aimed at elucidating the effect of condensates on uncoating as well as the role played by CypA in nuclear entry.

1-3 *Does PF74 inhibit CLP entry into condensates at a concentration that inhibits nuclear entry but not reverse transcription?*

The inhibition of nuclear import at low concentrations of PF74 (2 uM) is modest (only ~50%), while higher concentrations lead to capsid breakage (PMID 30567984). Thus, it is not possible to obtain conditions where entry into the NPC is effectively blocked. Since the assays measuring CLP entry into Nup98 condensates are not quantitative, it is not possible to show partial inhibition of CLP entry into the Nup phase.

1-4 *Does addition of the CPSF6 peptide promote exit of CLPs from Nup98 condensates?*

This is a very interesting question but is technically challenging to answer. There are three complicating factors to interpreting CLP internalisation: resolution (by confocal microscopy we can't confirm that CLPs are internalised), adsorption (CLPs adsorb to the Nup98 condensate surface even when they cannot penetrate – see 1-2 above), condensate aging (CLP are large and diffuse slowly, while the Nup98 condensates lose fluidity on a shorter timescale essentially trapping the CLP inside). By confocal microscopy, we do see ~40% reduction (pre-treated with CPSF6) and ~25% reduction (post-treated) in CLP associated with Nup condensates, which is remarkable considering the adsorption issues. We have not included these data, as a thorough treatment of the role of CPSF6 in nuclear entry requires careful consideration, including CPSF6's role in other phase separated compartments.

1-5 *Do native HIV-1 cores enter Nup98 condensates?*

At this stage we don't know whether capsid immersion in a Nup98 phase forms part of the uncoating trigger. If it does, then it might be expected that on the timescale of our experiments, cores would disintegrate. This might lead to the erroneous conclusion that they do not enter the phase. This process needs to be studied carefully as it could prove to be non-trivial. We intend to follow this up in a subsequent study focusing on the interplay between condensates and uncoating.

1-6 *The authors employed N57D mutant CLPs in their experiments but the mutant monomeric and hexameric CA proteins were N57A. Why the difference?*

The difference arose innocently due to parallel experiments performed by separate individuals preparing their own reagents. Both N57A and N57D have been validated to disrupt FG-binding, and have indistinguishable phenotypes, and we observe identical behaviour by FFS, so choice of either is an appropriate control. During revision, we identified an error in labelling of Fig 3g, which has been corrected.

1-7 *The title may suggest that the problem of HIV nuclear entry is now solved. I suggest something more nuanced, e.g., "Involvement of karyopherin mimicry in HIV nuclear entry"*

We are willing to consider alternative titles but disagree with this specific suggestion as we wish to convey the concept that it is the HIV capsid that is mimicking the structure and functional properties of karyopherins. An alternative title that we propose is '*The HIV capsid employs karyopherin mimicry to penetrate nuclear pores*'.

Referee #3:

This manuscript proposes a new model for HIV infection by which HIV capsids enter the nuclear pore using a similar mechanism to that of karyopherins. Through in vitro experiments, the authors suggest that HIV capsids, and therefore other lentiviruses, can breach the diffusion barrier of the nuclear pore complex through specific FG repeat interactions, and therefore is efficient even in post-mitotic cells. Typically, virus incorporation into a genome occurs upon mitosis, but the proposed model indicates how HIV and lentiviral packages persist despite division.

Using fluorescence fluctuation microscopy (FFS) and confocal microscopy on in vitro systems, the authors give insight into how HIV directly interacts with components of the nuclear pore complex. Overall, the “karyopherin mimicry” model is an interesting concept and addition to the current thinking about how viruses transport across the nuclear pore. This and the accompanying manuscript are important studies and should be published after addressing the following comments.

This is an accurate account of our study. We appreciate the referee stating that it should be published.

3-1 *The “dissection” of the NPC seems inconsistent. For proteins that did not express fully, only the FG domain was used. However, size is thought to sometimes play a role in binding affinity or uptake and recruitment. Perhaps a good control would be to compare across all FG domains only and control for size, especially since another argument made was that size does not play a contributing factor in HIV penetration of the nuclear pore. If this were true, then the difference between the wild-type protein and just FG domain would be minimal.*

In the discussion we comment that, in our model ‘...molecular weight does not represent a barrier to nuclear entry’. By this statement we are referring to the size of the cargo, not the size of the nucleoporin, about which we make no claim. In terms of controlling for the size of the FG, this is non-trivial as each FG-repeat domain has different numbers of FG’s and a different FG-density. In some cases (such as Nup358), the FG’s are not present in one distinct domain. Nevertheless, we believe we have controlled for FG size as best as is practical. Three isolated FG-repeat domains (Nup214₁₂₁₀₋₂₀₉₀, POM121₆₅₉₋₁₂₄₉, and Nup98₁₋₄₉₉) are of comparable size and all bind to CLPs. Conversely, the other domains of these proteins do not (see figure 2j,k). Furthermore, POM121₂₆₆₋₁₂₄₉, which represents the soluble portion of the protein (the N-terminus housing transmembrane regions), binds CLP equally well despite being 66% larger than the isolate FG domain (compare 1c and 2h). This serves to demonstrate that the difference between wild-type and FG-domain is indeed minimal, and that size, *per se*, is not an influential factor in this context.

3-2 *The authors choose Nup98 as their model system due to its important role in nuclear pore integrity and FG rich structure. They find that it binds capsid protein (CA) the best out of their Nup screen and test the selectivity properties and mimicry of their Nup98 condensate model system. However, it is*

unclear to me how we can be sure that this shows actual penetration of CA through the nuclear pore. Certainly, capsid particles partition into Nup98 condensates, but in vivo Nup98 and other components of the nuclear pore can be pulled out of the NPC to nuclear bodies or to assist in RNA transport. Thus, if we imagine this in a living system, would this show capsids in the cytoplasm dynamically recruited to Nup98 and other FG domains in the nuclear pore, and subsequently entering the nucleus? Or would we instead see that wherever Nup98 or other top binders from this study are, the capsids follow? Surely these are predominantly located within the NPC, but to eliminate uncertainty of the model being proposed, this might need to be confirmed.

The notion that nucleoporins play roles outside the context of the nuclear pore complex itself and that they are in different rates of exchange is a fascinating area of research. Indeed, we are also currently investigating the role of certain nucleoporins in mediating capsid transport across the cell – a role that may be independent of their function in the NPC, or may be involved in recruitment of capsid cores to NPCs. We also note that nuclear Nup98 condensates have been observed in certain cancers, and are also involved in cytoplasmic annulate lamellae. These are interesting, but their relevance to HIV infection needs to be carefully determined and is beyond the scope of the current study. As this concept represents a good example of an area of research that could be seeded by this manuscript, we have added the following text to the discussion: *'We also note that nucleoporins can be found in other compartments of the cell, such as annulate lamellae and in condensates within the nucleus (PMID: 33336681). While their relevance to HIV is yet to be established, our data suggest that non-canonical Nup function could potentially influence early infection.'*

3-3 *It may be useful to pick another promising Nup (i.e. other than Nup 98) from the initial screen that has rich FG content but differing function from Nup98 to confirm these experiments with another component and further validate such claims.*

The accompanying manuscript by Fu *et al.* provides examples of alternative FG-domains in both bio-layer interferometry and FG-phase assays. This complementarity serves to strengthen our argument that the capsid is a general FG-binder.

3-4 *Throughout the manuscript I was left wondering if there is a way to show this in dynamic fashion. For instance, it is alluded to that this happens over time and then the HIV capsid is recycled and the process continues to then infect cells. Is there a way to show that this model not only holds by showing partitioning patterns, but also that, in vitro or in vivo, it holds over time if we watch the capsids move and "breach" the condensates?*

We do not mention recycling, and this is not something we expect the HIV capsid to do. This comment may refer to the accompanying Fu *et al.* manuscript. We share the referee's desire for a method that

allows real-time measurement of capsid movement/breaching nuclear pores. Our Nup98 condensates are a compelling model for demonstrating the physicochemical principle of partitioning into this phase, and therefore how HIV can enter the diffusion barrier (the aim of our study). However, we deliberately avoided kinetic measurements, as real nuclear pores have added layers of complexity (including the points raised above about nucleoporin-mediated movement in the cell) that will influence kinetics and are not accounted for in the condensate model system. Nevertheless, we do have thoughts about modelling capsid transit kinetics, and they will be the subject of future studies.

Referee #4:

This manuscript by Dickson et al. claims the main component of the HIV capsid protein (CA) shows a karyopherin-like behavior, being able to establish multivalent, weak and specific interactions with the FG repeat domains in the NPC permeability barrier. This direct interaction would be enough to mediate the overcome of the permeability barrier, allowing the viral capsid to traverse the NPC and reach the inside of the nucleus in non-dividing human cells. They show how monomeric and oligomeric CA directly interacts with several flavors of FG domains in vitro and that such interaction is dependent on the integrity of the CA FG-binding pocket; The authors then use in vitro reconstituted FG domain condensates to verify that capsid-like arrangements of CA effectively interact with the FG components of the condensates and could actually even penetrate these condensates for distances that would account for the length of the NPC central channel. The interaction and merging of the viral-like particles with the condensates does not destabilize the CA oligomers and does not require the presence of nuclear transport factors, nor it is regulated by RanGTP. The authors propose that the karyopherin-like arrangement of CA into the capsid lattice provides the capsid with self-translocating properties, explaining the ability of the virus to access the nucleus of non-dividing cells.

MISREPRESENTATION OF THE MANUSCRIPT – We do not test monomeric or ‘oligomeric’ CA in our FFS assay. We only test capsid-like particles (CLPs). We also do not use of the word ‘oligomeric’ when referring to CA, as this is an ambiguous term that could be misinterpreted to refer to hexamers, pentamers, tubes, spheres or CLPs.

MISREPRESENTATION OF THE MANUSCRIPT - The comment that we claim that condensates do not destabilise the CA oligomers is false. In our discussion we state ‘*Whether the HIV capsid maintains its structural integrity whilst embedded within an FG-phase (be it NPC or CPSF6) remains an open question.*’ Furthermore we also state that ‘*...we have not explicitly addressed this question with our crosslinked capsid model.*’

COMMENT DIRECTED AT ANOTHER MANUSCRIPT – We present no data on RanGTP. This comment appears to be addressed to the accompanying Fu *et al.* manuscript.

MISREPRESENTATION OF THE MANUSCRIPT – The term ‘self-translocating properties’ is not one that we use, and may indicate that the referee has misinterpreted our claims. We are not claiming that the HIV capsid ‘self-translocates’ in the sense that it can mediate nuclear entry entirely independent of other factors. What we are claiming is that the ‘karyopherin mimicry’ model of the capsid specifically engaging with the FG-rich disordered proteins in the diffusion barrier enables the capsid to penetrate NPC. In other words, the NPC diffusion barrier is not a barrier to the HIV capsid. Indeed, we comment that our CPSF6 results are ‘*consistent with the proposed ‘ratchet’ mechanism by which CPSF6 extracts the capsid from the nuclear face of the NPC*’, and we anticipate that other factors are likely to be required to complete translocation. We also acknowledge that there is still debate in the HIV field as

to whether the capsid indeed does enter the nucleus intact or remodel during NPC transit (see our penultimate paragraph in the discussion).

However, a great deal has been recently published revealing an already significant understanding of how the HIV capsid traverses the NPC, and the manuscript provides only a very limited conceptual advance, compared with what is already known from multiple other labs. It is already well understood that CA interacts directly with FG domains as shown by a number of groups, e.g. Kane et al., 2018, and reviewed e.g. in Guedan, 2021; in fact, even structures showing exactly how CA binds FG repeats have been published by other groups (e.g. Price et al., 2014; Stacey et al., 2023), where a specific pocket binds to the FG repeat. It is unsurprising that mutation of this pocket prevents this interaction. Thus, we already know the molecular recognition mechanism by which the capsid interacts with FG repeats to enter the NPC.

MISREPRESENTATION OF THE LITERATURE – None of the four papers presented supports the referee's assertion that the molecular mechanisms are already known.

Kane *et al.* 2018 performed pull-down assays with CA-tubes and cell lysate in 2M NaCl. Reviewer comments are available with this paper, with one commenting that the assay was '*conducted in physiologically irrelevant high salt (2M) concentration.*' At these concentrations of salt, hydrophobic effects would be enhanced and potentially lead to false-positive interactions by pull-down. Additionally, this paper gives no details of the interactions that are made, which we reveal in Figures 1 and 2. The authors themselves state '*It should be noted that there are numerous interactions between individual Nups; therefore, binding to CA tubes by individual Nups in this assay may not necessarily reflect direct binding in all cases.*'

Guedan, 2021, is a review article on the state of the understanding of capsid nuclear entry within the HIV field. The article only refers to Nup358 and Nup153 specifically, as they are the best characterised in terms of structure and interaction with the HIV capsid. The only other mention of nucleoporins in general is to the above Kane *et al.* paper. This review serves to highlight how under-developed the thinking around capsid nuclear entry is within the HIV field.

We are intimately familiar with Price *et al.*, 2014 as David Jacques was co-first author on that paper, and it was its shortcomings that inspired the current manuscript. It reports, amongst other findings, the only crystal structure of an FG-Nup bound to the HIV capsid hexamer. The study is limited to a single FG-containing peptide (Nup153, residues 1407-1423) bound to the capsid hexamer. At the time, it was thought that this was a unique interaction motif. The structure was striking because it showed that two cofactors, Nup153 and CPSF6 bound with very different footprints on the capsid surface, which could be differentiated by CA mutation. We illustrate this in Figure 2a. It is disingenuous to say

that because of this work, it is now known ‘exactly how CA binds FG-repeats’. This work did not use FG-repeats, and highlighted that no two cofactors bound in the same way. With the more recent discovery of Sec24c (Rebensberg, *et al.* 2021, Nature Microbiology), we see that a third binder at this site adopts yet another conformation. It was entirely reasonable to question whether other FG-Nups make contacts consistent with any of these binding modes. Furthermore, our current study shows definitively that it is necessary to have the multitude of binding sites of the CA lattice and the whole FG-repeat regions in order to detect the majority of capsid:Nup interactions. Studies on CA hexamers and single FG-containing peptides have been insightful, but paint an incomplete picture.

Stacey, *et al.* 2023 performed a cryoEM study which included the Nup153 peptide (residues 1407-1429) bound to purified HIV cores. The results agree with the structures from Price *et al.*, 2014 and offers no additional insight with respect to FG-Nup:Capsid interactions beyond this one specific peptide.

MISREPRESENTATION OF THE MANUSCRIPT- The statement that it is ‘unsurprising that mutation of this pocket prevents interaction’ is bizarre. We use these defined mutations as controls in Fig 2b,c to contextualise our FFS assay with what was already known in the literature, and to demonstrate that the FFS assay is capable of identifying crucial interaction residues on the capsid surface. The results presented for Nup42, Nup58, Nup62, Nup98, Pom121, and Nup214 in Fig 2 and Fig 3 are all new observations that build from these controls.

CA is well understood to interact with a host of other cellular proteins that mediate interaction with the NPC. The FG binding pocket is known to bind several other non-Nup proteins such as CPSF6, and Sec24C. CA also interacts with the nuclear transport factor TNPO3 (e.g. Zhou et al., 2011), and with the nucleoporin Nup358 not through its few FG repeats but directly with its cyclophilin-like domain. Although the authors acknowledge these interactions are known, they do not account for all these other host co-factors that have been shown to be involved in the actual mechanism of capsid entry to the NPC.

UNSUBSTANTIATED CLAIM – The referee states that CA is well understood to interact with a host of other cellular proteins that mediate interaction with the NPC. We have found no evidence that the factors mentioned, CPSF6 and Sec24c, mediate interaction with the NPC. TNPO3 is disproven (see below), and Nup358 engages via a cyclophilin domain which recognises a surface-exposed loop on the capsid (Bichel *et al.* 2013). Nup358 forms the cytoplasmic filaments of the NPC, with the cyclophilin domain at the extremity. It likely binds the capsid as it approaches the NPC, but would not be able to facilitate the capsid crossing the diffusion barrier on its own. FG-binding and Nup358 binding are not mutually exclusive. We could add a comment along these lines, but it is unclear what concern the referee is raising.

MISREPRESENTATION OF THE LITERATURE – The interaction between CA and TNPO3 reported in Zhou *et al.* 2011 has been comprehensively disproven (Schaller *et al.* 2011, Tu *et al.* 2011, Ambrose *et al.* 2012, Maertens *et al.* 2014). Its effects on the HIV lifecycle are indirect and a result of its role in mediating the nuclear import of CPSF6. We also show data that TNPO3 does not bind CLPs (Fig S3-6), and we state in the discussion: ‘cytoplasmic mislocalisation of CPSF6 (either through ectopic expression of CPSF6 lacking a nuclear localisation signal, or depletion of the dedicated CPSF6 karyopherin, TNPO3) prevents the HIV capsid from entering the nucleus’.

MISREPRESENTATION OF THE MANUSCRIPT

It is disingenuous to state that we have not accounted for these interactions. We discuss them frequently throughout the manuscript, and even probe them experimentally as stated above.

Also, given the known dynamic nature of these interactions, the argument that their addition to the entire capsid would form a shell too big to pass through the NPC is an unproven assumption, with many other possible arrangements and substoichiometric associations allowing for this passage, in agreement with the fact that these other proteins are known to be involved in capsid passage into the NPC.

COMMENT DIRECTED AT ANOTHER MANUSCRIPT – We have made no claims about any factors forming a shell. This comment appears to be addressed to the accompanying Fu *et al.* manuscript.

There is evidence that structural flexibility of the intact CA is required for entry, although whether this is linked to disassembly or to distortion is unresolved {Guedan, 2021, 34543344}, discussed below. As stated by Guedan, 2021: “thus, the viral core should be considered as a dynamic structure that binds numerous cellular proteins on its path through the NPC”. This contrasts rather with the picture presented in this manuscript.

MISREPRESENTATION OF THE MANUSCRIPT – We investigate and/or discuss the relevant cellular proteins (see above) and dedicate the penultimate paragraph of the discussion to commentary on the possible capsid dynamics (remodelling and uncoating) that may be at play during nuclear pore transit. We even cite Guedan *et al.* 2021 in this very context. To say that we present a contrasting picture is non-sensical.

The other key questions in the field are also left unaddressed by this work. These include: what is the sequence of events, does the NPC have to dilate or be otherwise re-structured in any way, what do the nuclear basket, Nup88 complex and cytoplasmic filaments do, (how) are viral entry to the NPC and uncoating coupled, and it is unclear if intact capsids go all the way through the NPC into the nucleus

or they disassemble during passage through the NPC. Zila *et al.*, 2021 showed that the capsid orients narrow end first towards the NPC and that the NPC may dilate as the capsid enters to accommodate it, pointing to an organized mechanism of entry and so interaction with the NPC; their EM analyses also showed that the capsids appeared to undergo major remodeling as it enters the NPC, an observation supported by other groups' work. None of these key steps are addressed here. Thus, at the very least, the model presented here – based on all this other work – is undoubtedly very incomplete.

CRITICISM BEYOND THE SCOPE OF THE MANUSCRIPT – Our aim in this study was to test whether the HIV capsid's thousands of FG-binding sites confer on it the intrinsic ability to penetrate the diffusion barrier of the nuclear pore complex. We deliberately kept the study focused on a reduced system as this allows us to distil out the fundamental physicochemical properties involved. The model we propose represents a significant conceptual reframing of HIV nuclear entry, which has remained incompletely understood for 40 years. Each of the questions proposed by the referee have either already been answered or are serious undertakings well beyond the scope of the current manuscript:

-Sequence of events: it is unclear to what the referee is referring

-NPC dilation: Zila *et al.* 2021 reported that the NPC does dilate

-What do the nuclear basket, Nup88 complex and cytoplasmic filaments do: it is unclear exactly what the referee expects of these structures, but this could be subject of a follow-up study.

-How are NPC entry and uncoating coupled: we discuss this in the manuscript. An answer to the problem of HIV uncoating would be a major discovery, and it is disingenuous to trivialise this challenge as the referee has done. We show that if this is to be answered, it needs to consider a role for phase separation.

-Do capsids disassemble during NPC transit: this is the same point as above.

-Do intact capsids go all the way through the NPC: this is a debated area of HIV research, but there is considerable recent evidence that we have cited that support intact capsids in the nucleus (Burdick *et al.* 2020, Li *et al.* 2021, Müller *et al.* 2021).

MISREPRESENTATION OF THE MANUSCRIPT – Again, the referee ignores our discussion points on capsid remodelling, and the fact that we refer to Zila *et al.* 2021 paper when stating our premise, during our analysis, and in our discussion (reference 7).

UNSUBSTANTIATED CLAIM – The referee bases their concluding remark on 'all this other work' but has only presented one reference (Zila *et al.* which we have fully incorporated into our manuscript) alongside a list of their own out of scope questions. At this point in their review, they have not presented a single reference that supports their position.

The study also contains significant flaws, causing the conclusions not to be fully supported by the data presented here. Taken together, this work is not suitable for publication in Nature at this stage. My other main general concerns are the following:

4-1 *The authors work with the assumption that FG macroscopic condensates recapitulate the properties and behavior of the nanoscale NPC permeability barrier, when this is not a generally accepted concept, deriving as it does from in vitro studies such as these. This assumption and this in vitro system has been contradicted in a recent publication in Nature itself (Yu et al., 2023) and in BioRxiv paper (Kozai et al., 2023), showing that the behavior of FG domains in condensates does not recapitulate the state and properties of native, in situ, NPC anchored FG Nups. Thus, many of the conclusions in this paper – based on this other work – could be incorrect. The authors fail to address this serious issue.*

MISREPRESENTATION OF THE LITERATURE – The referee makes the criticism that our use of condensates to model the properties of the NPC permeability barrier is not generally accepted. They offer one recent peer reviewed article from Nature, Yu *et al.* 2023, in support of their position. This is a fascinating study of the dimensions and dynamics of Nup98 within the nuclear pore complex. In Figure 1 of the paper, the authors compare the functional Nup98 within the NPC with phase-separated condensates *in vitro*. They prepare Nup98 FG-domains using a similar approach to us:

‘Purified NUP98 FG domain was phase- separated in vitro by rapidly diluting a denatured highly concentrated stock solution into physiological buffer. The permeability of the droplet-like condensates formed was measured rapidly where they still obeyed liquid-like characteristics. The droplets recapitulated the function of the permeability barrier, as shown in the passive exclusion and facilitated transport assays.’ This paper, therefore, is in complete agreement with our approach to modelling the permeability barrier *in vitro*.

USE OF NON-PEER-REVIEWED MATERIAL – Kozai *et al.* is a preprint from the lab of Roderick Lim. Such a document is poor evidence for what may or may not be generally accepted in the field as it has not been subject to peer review. We also note that it was posted after the submission of our manuscript. Nevertheless, in Figure 4a they observe *in vitro* NPC-like transport selectivity of a phase comprising the yeast homologue of Nup98, showing that this system is valid for probing karyopherin-like properties.

We state in our manuscript that the condensates “*recapitulate the selectivity properties of the diffusion barrier*”. This is a reproducible observation, not an assumption, made in multiple studies over more than 15 years. Some examples of peer-reviewed studies include Frey and Görlich, 2007; Hülsmann et al., 2012; Labokha et al., 2013; Mohr et al., 2009; Schmidt and Görlich, 2015, Celetti et

al., 2020, Ng et al., 2021. These studies have also been cited in high-profile reviews including Aramburu and Lemke 2017; Lin and Hoelz, 2019; and Musacchio et al., 2022. This non-exhaustive list of primary literature and review articles demonstrates the reproducibility and broad acceptance of the FG-Nup phase separation model for understanding NPC transport selectivity.

4-2 *In my opinion, the most parsimonious and better supported explanation for the data presented here would be that the CA capsid lattice might mediate the docking of the HIV capsid into the NPCs, but none of the manuscripts show compelling evidence that the HIV capsids or even the smaller 40 nm capsid-like spheres do traverse the NPC in vivo. The authors themselves discuss this option as a side idea, instead of considering it their main conclusion, with the Kap-like trafficking through the NPC as a potential, but not proven, possibility. The incorporation into FG condensates is no proof, any more than the ability to make omelets from eggs proves that chickens arise from omelets. For the reasons stated below, and it would be nice to show it in an actual NPC.*

COMMENT DIRECTED AT ANOTHER MANUSCRIPT – We do not use spheres in our study. This comment may be directed at the accompanying Fu *et al.* manuscript. Regardless, as stated above, we have cited the papers reporting intact capsids in the nucleus, proving this is beyond the scope of the study.

4-3 *Karyopherins establish multivalent, weak and transient interactions with the FGs in the NPC. It is not clear from the evidence presented here that such is the mode of interaction of the CA multimers in the HIV capsid, so without further data and controls, the identification of CA to a karyopherin, although evocative, might not be an accurate description of the actual behavior of the CA lattice and could be quite misleading to a general reader.*

CONTRADICTION – The referee states that the evidence is not clear, and wants more controls. What controls do they want? We show controls in all figures. The evidence for weak interactions is presented in Figure 2I, where we lose binding as we further subdivide FG-repeat domains.

4-4 *It is also not clearly stated throughout the manuscript the actual time length of the experiments performed with these condensates, which are, acknowledged by the authors themselves, prone to “aging”, gelation and formation of potentially aberrant beta-structures, a circumstance that is hard to believe happens in a living NPC permeability barrier. How would the “aging” of the condensates affect the different experiments shown in these manuscripts?*

Nup98 condensates do indeed age, a circumstance which we experimentally controlled for in this study. We use condensates which still show liquid like characteristics and stated the timing in the

methods. To emphasise in the main text we added the following: ‘We tested these condensates for their selective properties *by mixing them immediately after formation (where they still maintain liquid-like characteristics)* with mCherry fused to the Importin-beta-binding domain of Importin-alpha (IBB-mCherry).’

4-5 *It is worrying that the presence of karyopherins massively outcompete the CA protein or hexamer in its interaction with FG condensates. That would precisely be a closer situation to what a HIV capsid would find in a living NPC: hundreds of copies of the different types of karyopherins and other nuclear transport factors (carrying RNPs, Ran, etc) also interacting and passing through the NPC, together with their cargoes and other molecules, forming a heterogeneous, complex, and dynamic nano-environment. The capsid would never find pure and homogeneous FG condensates. How do the authors reconcile these apparently conflicting observations?*

This is a consequence of experimental conditions (order of addition, relative stoichiometry, and relative affinities for FG-motifs). We don’t find the results ‘worrying’ as it makes logical sense that two species that compete for the same site will displace one another depending on relative concentrations. Within a real NPC during HIV infection, we expect that the volume of the capsid and its massive avidity for FG-motifs will likely gradually exclude other transport factors. Karyopherin-NPC interactions themselves are modulated by weak cellular competitors (PMID: 22357553). It is possible that a karyopherin competition modulates CA-NPC translocation. As we have no kinetic data, we haven’t commented on this possibility.

4-6. *Considering that the authors are trying to establish an analogy between the HIV capsid and karyopherins, it is surprising that they did not measure the interaction between any of the importins shown in Fig. S3 and the Nup98 condensates using their FFS measurement setup. Such an additional control/example would be desirable to be able to establish a more truthful comparison between the behaviors of CA and karyopherins.*

The FFS approach relies on the HIV CLPs condensing the putative binding partner on the CA lattice. This concentrates the binder, creating foci that pass through the confocal volume. A measurement of Nup98 binding to importins does not involve the CLPs. It may prove feasible to adjust the experimental parameters to assess whether importin and Nup98 coincident by this method, but the diffusion rates and relative intensities will differ from the CLP platform, and they may not be directly comparable. We also feel this is an unnecessary measurement as we see the relevant result in Figure 3b, which is also a confocal imaging method but contains more information than an FFS trace.

4-7 *The authors reason that, because the binding of a single FG is below the limit of detection in their FFS measurements, a “weak, but FG specific, interaction enhanced by avidity” is suggested. However, another interpretation would be that their result might just probably reveal an intrinsic limitation of*

their method and not necessarily a behavior analogous to a karyopherin. If that's the case, their analogy is unsupported and highly speculative.

UNSUBSTANTIATED CLAIM – Our model is based on multiple lines of inquiry. FFS shows the capsid only binds the FG-domains (Fig 2k). As the domains are subdivided, coincidence reduces (Fig 2l) to the point where a single FG peptide is below the detection limit (Fig 2l) for binding. This is exactly what would be expected for a multivalent/high-avidity interaction. The model is further reinforced by the karyopherin-like behaviour seen in the condensates (Figure 3). The referee's comment that our model is unsupported is false, and their alternative interpretation is not clear.

4-8 *In figure S3-1, I agree that the N57A mutation has an effect on the behavior of the constructs, but, unless I am misinterpreting something, I can clearly see in the pictures that the signal from the mCherry channel significantly concentrates at the rim of the condensates even in the N57A mutant. Would this indicate CA has a certain avidity for protein condensates independently of its FG-binding pocket, but it is not able to move across those condensates? Could the authors include some kind of control to show that CA and its hexamer do not have just a non-specific avidity to protein condensates?*

The hexamer represents a model CA system that is well-accepted in the HIV field, but presents internal surfaces that would not be exposed to the diffusion barrier in a closed capsid. It is possible that this construct therefore does have a degree of non-specific interaction with the condensate surface as the referee suggests. But the difference in capsid penetration is already controlled, and the statistical treatment of it relative to WT hexamer is presented in Figure 3g.

4-9 *In the section "FG-repeat domains are specifically recruited to the FG-binding pocket on the HIV capsid" to formally demonstrate this point it would be interesting to show that a phenylalanine mutant in the FG repeats does not interact with CA, completely discarding that the interaction could be helped or be mediated by other residues.*

We note that phenylalanine mutation was shown to abolish the interaction by Price *et al.* 2014. Taken together with the mutational data in figure 2, as well as the comparison of FG- and non-FG-domains across three proteins in figure 2k, we believe that the argument that the interaction is mediated by the phenylalanines to be highly compelling.

4-10 *Line #95 "FG-repeat proteins are known to phase-separate", please add here the caveat "when overexpressed and are not assembled into the NPC", otherwise you are misleading the reader.*

We have rephrased as 'FG-repeat proteins are known to phase-separate *in vitro*.'

4-11 Line #192, please correct "an affect" with "an effect".

We have corrected this typographical error.

Arbitrating Referee (comments communicated by Editor):

4A-1 *Regarding novelty and advance, the arbitrating referee tended to agree with our more critical ref. This referee also felt the advance was somewhat incremental, particularly in light of a recently published paper in Nature Communications*

(<https://www.nature.com/articles/s41467-023-39146-5>). The reviewer was also puzzled by a disagreement between your work and the work presented in that paper, where knocking down Nup98 didn't have much effect, while knocking down Nup35 had a more pronounced effect. That paper must be cited in the revision and the discrepancy addressed.

The recently published Nature Communications paper (Xue *et al.*) has a title which, unfortunately, does not convey the content of the paper. The approach taken in that study is wholly different from ours and we disagree that we are making an incremental advance relative to this work.

In our present study, we:

- Comprehensively and systematically explore FG *interactions* with CA across all FG-containing nucleoporins in the NPC. Our FFS approach avoids the limitation of pulldown assays (such as false-positives through Nup-Nup interactions or membrane association) and variations due to cell type.
- Systematically describe the FG:CA interactions specificity
- Demonstrate that FG:CA interactions are individually weak and multivalent, a critical property for karyopherin mobility.
- Directly observe penetration of permeability barriers, and can disrupt this affect through mutation of the FG-binding pocket on CA.
- Experimentally demonstrate the parallels between HIV and karyopherins.

In contrast, Xue *et al.*:

- Use HIV reporter virus (single cycle, VSV-pseudotyped) to infect HeLa cells treated with nucleoporin-targeting siRNA. Importantly, they present no data that the nucleoporins

have indeed been depleted in this screen. The screen does not attempt to be systematic, but rather identify those nucleoporins that lend themselves to further study by this approach.

- Focus on Nup35, Nup153, and POM121 as these were the nucleoporins that, when subjected to siRNA, gave a phenotype for wild-type virus, but not N74D or P90A. These residues do not mediate CA:FG interactions and these mutants do not demonstrate binding between capsid and NPC.
- Report for the first time that Nup35 depletion influences HIV infection.
- Claim to have discovered the relevance of POM121 to HIV infection, but do not cite Saito 2017 or Guo 2018 who showed this prior.
- Provide no experimental evidence directly linking FG:CA binding to nuclear entry.
- Describe a model in the introduction and discussion that is bolted on to unrelated experimental data.

To address the comment that Nup98 knockdown had little effect, Xue *et al.* only show Nup98 in the initial screen, where it has a modest effect on HIV infection for the two mutants tested. We would expect such a result from a partial knockdown. Importantly, they provide no evidence as to the degree with which it was knocked down, nor do they discuss it anywhere in the text. In the absence of additional data or commentary on their part, there should be no concerns that there is any discrepancy between their work and ours. Nup98 simply wasn't a focus of their study.

With respect to Nup35, Xue *et al.* are the only group to report its relevance to HIV (6 other studies found no phenotype or interaction, including ours – see Table S1). They report an interaction by overexpressing Nup35 in HEK239T cells and pulling it down on CA-NC tubes. They claim that the terminal FG binds to the capsid, but inspection of the relevant western blot (Supplementary figure 9b) reveals that mutation of this FG has not abolished the interaction as claimed in the text. Such a pulldown, where other proteins are not accounted for (eg other components of the NPC), and where the mutational data is at odds with the description of results, does not make a compelling case for an interaction between Nup35 and CA. Likewise, there is no mutational data on the capsid (eg N57D) to support this interaction - a result that is conspicuous by its absence. Furthermore, structural information on the complete NPC shows that Nup35 is placed in the inner ring subcomplex, where it is not accessible to the capsid. It is also not considered an FG-repeat nucleoporin as two of its three FG motifs are in structured domains, and karyopherins do not interact with it. Xue *et al.* make no comment on these structural realities. While their study does show that depletion (to an unspecified degree) of Nup35 results in an infection phenotype in HeLa cells, the evidence for Nup35 interacting with the capsid is weak, and the involvement of FG's is not supported. Their claim that 'the HIV core is an opportunistic nuclear import receptor' is conjecture that is uncoupled from their data, especially given that nuclear transport receptors do not bind Nup35.

Referee #1 recommended that we include a summary of studies that have sought to address the interactions between HIV and specific nucleoporins (see 1-1 above). We have generated this table (Table S1) and have included the Xue *et al.* paper in this context as it is the most recent of many papers

attempting to shed light on this issue. Despite similar traditional approaches (mostly knockdowns and/or pulldowns) there is considerable inconsistency in the findings between the groups. Our approach is unlike any previous study in the HIV field as it allows us to be truly systematic and gives unprecedented mechanistic insight into the HIV-NPC relationship.

4A-2 *Regarding physiological relevance, the arbitrating expert was somewhat equivocal on the question of whether in vitro FG condensates are sufficient to recapitulate the in vivo NPC, but the expert did acknowledge that they can be useful transport models, when used to complement cell-based studies.*

We have approached this study from an HIV capsid-centric perspective in order to understand how it engages with the NPC. The prior cell-based literature supporting the physiological relevance of HIV nuclear entry is vast. Table S1 shows a subset of those studies that have attempted to determine which components of the NPC are relevant to HIV. Other studies include the work of Zila *et al* 2021 who imaged HIV capsids within NPCs by cryoET, and there are numerous studies imaging capsids in the nucleus. Capsid engagement with the NPC during infection is well-established. Ours is the first study to demonstrate that the HIV capsid is capable of mediating its own entry into the permeability barrier. The model system we chose is currently being used for this purpose in other high-profile studies. Indeed, Yu *et al.* Nature 2023 (from the group of Edward Lemke) recently published the use of Nup98 condensates for the reason that they ‘recapitulated the function of the permeability barrier’. The observation that the HIV capsid can independently penetrate these condensates is remarkable. It represents a significant shift in our understanding of the HIV life cycle, and has broad implications for viral nuclear access as it is now clear that viruses need not be constrained to the canonical nuclear import pathways.

Referee #5:

Dickson and colleagues present a powerhouse, biochemistry-based study that significantly informs the mechanism of HIV nuclear transport. Leveraging in vitro translation and fluorescently-labeled capsid like particles (CLPs) made from purified CA protein, they detect Nup-CA binding via fluorescence fluctuation spectroscopy (FFS). From this they rank ability of various human Nups to bind CLPs. They furthermore leverage Nup98 condensates to gauge penetration distance of various substrates (CA, karyopherins), culminating in rather clear images of internalized CLPs. Much of the work is carefully controlled by the use of CA/CLPs mutants as well as peptide competition. The overall conclusion that the HIV capsid mimics human karyopherin behavior to gain NPC entry/passage is novel and will predictably be of interest to wide, general readership. The study overall was quite impressive.

There is no virology data provided. Nevertheless, appropriate prior studies were in general cited, and the writing well-framed the important advancements of this in vitro work to the larger body of literature.

The paper will be improved by incorporating some pertinent references and by providing additional transparency in terms of proteins analyzed and related images. Given the full in vitro nature of the work, it is critical to fully inform readers of analyte quality.

This is an accurate account of our study, and we thank the referee for commenting on its broad interest and that they found it impressive. We also thank the referee for acknowledging that virological experiments already exist in the vast body of HIV capsid research literature, and for recognising that we have attempted to place our research in that context. We welcome their suggestions for additional references and have incorporated them. We also agree with presenting the data on analyte quality (see 5-2 and 5-3 below).

5-1 *Was Nup358₃₀₅₈₋₃₂₂₄ protein tested by FFS? Due to the C-terminal Cyp-homology domain, one would predict binding, and it would be informative to compare this to the utilized CypA control. Perhaps I missed this data. If you don't have it, no need to do the experiment. In this case, either remove Nup358₃₀₅₈₋₃₂₂₄ from Table S1 or add footnote to say it was only analyzed by AlphaFold and not studied as protein.*

We thank the referee for pointing this out. We did collect FFS data and have now included it in Fig S1c,d for completeness. Nup358₃₀₆₅₋₃₂₂₄ has a lower affinity for CA than does CypA or CPSF6 (CypA ~10 μ M, CPSF6 ~50 μ M, Nup358 ~100 μ M) (PMID:23902822), and the FFS data reflect these differences.

5-2 *The paper uses numerous proteins, many made via in vitro translation. Although Table S1 provides some information, the paper will be improved by providing additional transparency. Either greatly expand Table S1 or add separate Table that lists all proteins and peptides used in the paper (at construct level). Although Uniprot Acc is useful, additional work is required to track down specific isoforms, which applies for many of the proteins. Please add a separate column indicating aa content of corresponding full length (FL) proteins.*

We have included a spreadsheet of protein sequences in Supplementary Information. Please note that in the revised manuscript, the relevant table has been renumbered as Table S2.

5-3 *Please add SDS-PAGE images of all in vitro translated proteins. Although I realize FFS affords internal label proximity control, it nevertheless is important to inform readers of analyte quality. If the fluorescent proteins are not discernable via Coomassie blue, please perform appropriate western blots to highlight these alongside Coomassie images.*

We share the referee's enthusiasm for quality control. As the fusion proteins are labelled with GFP, we routinely perform in-gel fluorescence, which abrogates the need for western blot or Coomassie staining, as it reports only on the material being observed in the FFS experiment. We have added an additional extended data figure (S5-1) with these gels. We have also included Coomassie-stained SDS-PAGE gels for all purified proteins in Fig S5-2.

5-4 *Along these lines, please provide negative stain images of assemblies CLPs (WT, N57D, N74D, A77V; Fig 2b-g).*

We thank the referee for this suggestion, and have added an additional extended data figure (S5-3) showing the requested CLP electron micrographs. We used CryoEM rather than negative stain to maintain consistency with other micrographs in the manuscript.

5-5 *Intro first paragraph. It is overstatement to say all genera outside lentiviruses require M phase. While this is well-established for Mo-MLV, other gammaretroviruses as well as betaretroviruses are claimed to effectively infect growth arrested cells. While these studies may be one-offs, more careful work with alpharetroviruses indicate these lay somewhere between HIV and Mo-MLV. Please rephrase to more accurately reflect the literature.*

We thank the referee for their insight, as we were unaware of these studies. To more accurately reflect current understanding, we have rephrased this sentence: 'For most retroviruses, integration occurs during mitosis...'

5-6 Page 2 first full paragraph "cleavage and polyadenylation specificity factor 6"

We have corrected the name to ensure consistency with uniprot.

5-7 Ending sentence, refs 28, 29 established that elements of an adjacent CTD contribute to the FG binding pocket.

These references show that the adjacent CTD does not contribute to the binding of the FG motif of the CPSF6 and Nup153 peptides, but contacts regions outside the FG motif (as co-first author, David Jacques solved these structures in Price *et al.* 2014, PMID: 25356722). Strictly, the CTD plays no role in forming the pocket responsible for making the contacts with the phenylalanine and glycine of the FG motif, and the FG binding is essential for the interaction to occur. The sentence 'While these proteins are structurally distinct and found in different cellular compartments, each of their interactions with the HIV capsid depends on an FG-motif which buries itself into a pocket in the CA N-terminal domain.' is accurate.

5-8 Three lines from bottom. Pom121C was also studied as fragment; please add. Table S1 reports Nup214 coordinates as 1209-2090. Please fix.

We have clarified this statement: 'For Nups that did not express as full-length proteins, we either truncated the transmembrane domain (Pom121₂₆₆₋₁₂₄₉) or expressed only the FG-domain (Nup98₁₋₄₉₉, Nup214₁₂₁₀₋₂₀₉₀).' The Nup214 boundaries have been fixed.

5-9 Page 3 bottom full paragraph line 3. Truncations correct? Nup42 in these experiments appears to be FL.

We have rewritten this sentence for clarity and in response to 5-16 below. It now reads: 'To investigate how the capsid recognises FG-repeats, we examined CA mutant binding to our binders (Nup42_{Full-length}, Nup58_{Full-length}, Nup62_{Full-length}, Nup98₁₋₄₉₉, Pom121₂₆₆₋₁₂₄₉ and Nup214₁₂₁₀₋₂₀₉₀).'

5-10 *“These observations may explain why...” overstates. N74D is defective for macrophage infection (ref 68).*

The defect in macrophage infection is only observed in spreading infection and can be restored by blockade of interferon signalling (Figure 1c of Rasaiyaah *et al.* Nature 2013, PMID: 24196705). Therefore, N74D does possess the ability to infect non-dividing cells, and the block is due to innate immune sensing and interferon signalling. We had not cited reference 68 (Ambrose *et al.*, J. Virol 2012) in this context as this block to spreading macrophage infection was not fully elucidated until subsequent studies. Please note that due to editorial restrictions on the number of citations, we have now removed reference 68.

5-11 *Page 4 line 6, place (Fig 2j and S2-3) at end of sentence.*

We have corrected as suggested.

5-12 *Normalized FFS data indicates CypA the most robust binder. Please amend last sentence of second full paragraph.*

The sentence has been amended to read: *‘Importantly, of the nucleoporins, Nup98 was also the clearest CA binder in our FFS assay (Fig 1b,d).’*

5-13 *The N57A mutation had less of an affect on CA hexamer versus CA-mCherry radial avg (Fig 3c-f; S3-1c-f). Although perhaps maybe 2-fold, page 5 line 4 describes the CA hexamer-mCherry affect as “significantly reduced”. Please calculate and report these p values.*

The statistical treatment is given in figure 3g. $p < 0.0001$.

5-14 *Page 5 second full paragraph line 3 cites 46, 47. My reads indicate while ref 47 showed evidence for direct binding, ref 46 claimed a genetic as compared to biochemical interaction.*

Correct. We have removed ref 46 as we are over-budget on citations.

5-15 Many labs have measured HIV infection in Nup knockdown cells, and results of many of these studies are fully consistent with this current work. Discussion line 2, “cannot be easily manipulated”

We have completely rewritten the opening of the discussion to clarify this point, and to incorporate our response to 1-1, 4A-1 and 5-16: “There are several reasons why the HIV: NPC interaction has remained poorly understood. In cellulo approaches can be confounded by both the essentiality of the Nup genes and the complex interactome of the NPC. Protein depletion approaches may be incomplete and/or result in changes in the abundance and/or mislocalisation of other Nups, while pulldown assays can yield false positives due to Nup-Nup interactions. Nevertheless, over one third of nucleoporins have previously been implicated in various aspects of the HIV lifecycle using these methods (see Table S1). Despite similar approaches, the consistency with which specific Nups are identified varies considerably.”

5-16 Line 5, “any molecular details” is disingenuous. E.g., an important Nup153 FG motif was identified 10 years ago (refs 27-29, 35). Moreover, a recent NSMB paper partially reconstituted HIV nuclear import using Nup-docked DNA origami (PMID: 36807645). While the current paper surely adds important dimensionalities, the NSMB paper could be cited. Moreover, this paper revealed context dependent CA-Nup62 binding. It would be informative if you could frame your own Nup62 results to what’s been previously published. After all, it’s the only Nup that fell outside the subset shown to directly engage karyopherins (bottom page 6).

We thank the referee for pointing out this overstatement, which had resulted from a typographical error. We had intended it to read ‘...many molecular details...’. This statement has been removed in the rewriting of the opening of the discussion (see 5-15 above). We agree that our Nup62 results should be better framed within the literature and have provided such context, including the reference to the DNA origami paper, in Table S1 summarising prior studies requested by Referee 1 (see response 1-1 above). Previously we did not mention the Nup62 results at all in our text, focusing rather on ‘top binders’. To be clearer we have now explicitly stated that Nup62 is binding to capsid and have provided CA mutation data demonstrating it binds specifically via the FG-pocket of capsid (Figure 2). We have rephrased the discussion (3rd paragraph, 2nd sentence) accordingly: “A number of karyopherins have been shown to interact directly with FG-motifs of Nup42, Nup62, Nup98, Nup153 and Nup214, and notably all of these Nups have been identified as CA binders in our study.” We have also highlighted Nup62 in figure 1e.

5-17 Paragraph 4 line 5 should cite PMID: 26586435, the first to show this.

We have added this citation.

5-18 *Bottom paragraph line 2: the predominant CPSF6 isoform harbors a single FG. While CPSF6 may phase separate (this has not been strictly demonstrated for the FL protein), it would seem incorrect to refer to this as “FG-phase”.*

We had not intended for CPSF6 to be regarded as an ‘FG-phase’ but rather a protein that is found in phase-separated compartments (such as speckle-associated domains) that happens to have an FG-motif capable of binding the capsid. We agree that the statement was unclear and have reworded: *‘Whether the HIV capsid maintains its structural integrity whilst embedded within FG-containing phases (be it NPC or CPSF6-rich compartments such as speckle-associated domains) remains an open question.’*

5-19 *Please fix ref 39 citation.*

As we are over-budget on citations, we removed this reference.

5-20 *Fig S3-3e lacks x-axis indicators.*

We have added the missing x-axis labels in Figure S3-3e.

Reviewer Reports on the First Revision:

Referees' comments:

Referee #1 (Remarks to the Author):

In this revised manuscript, Dickson and coworkers have improved their study by including a new Table and included results with a mutant capsid-like particle (N57D). The latter result adds some support for the biological relevance of the condensate internalization assay with CLPs. The study remains quite innovative in that it presents a new view of HIV-1 nuclear entry that is likely to be intensively tested in future studies.

Authors declined to test the internalization of native HIV-1 cores, expressing concern about potentially uninformative outcomes. Their worry that cores would disintegrate over the timescale of the experiment seems ungrounded based on the well-documented stabilization of HIV-1 cores upon the addition of physiological concentrations of inositol hexakisphosphate. Several groups have published studies employing purified HIV cores, so this is feasible. I agree that the process needs to be studied carefully, but this experiment seems worth performing to further support the biological relevance of the CLP condensate penetration results.

Authors included new condensate results with one mutant CLP. While this is helpful, it is not a very thorough test of biological relevance, and inclusion of additional mutants known to be impaired for nuclear entry is needed to support the major claim of the study. Moreover, the authors' statement that the condensate entry assay is not quantitative is surprising, given that quantitative results were shown in Fig. 4g.

The authors stated the meaning of the error bars in the respective figure legends. However, though they provided p values for some results, I was unable to find a statement regarding the statistical tests employed. This information should be included.

Minor point: The authors may wish to consider including PMID:20227665 in Table S1. This 2010 paper was a landmark study in the field of HIV-1 nuclear entry.

Referee #3 (Remarks to the Author):

In my initial review of the manuscript by Dickson et.al. I was overall positive, and suggested that the authors address a set of comments and critiques that would then make the paper suitable for publication. I am satisfied with the responses to these points I made previously.

In reading the comments of the other reviewers, I see there is a robust discussion of other aspects of the manuscript, and which seems worth considering, but which I am not qualified to speak about.

Referee #4 (Remarks to the Author):

The authors have still not addressed the key issues properly, largely choosing to argue with these and the significance of published findings rather than provide experimental data to support their points. These issues can be summarized as follows:

The first and perhaps most critical issue is of significance. The authors state “this karyopherin mimicry model resolves a key conceptual challenge for the role of the HIV capsid in nuclear entry”. However, this is not the case - karyopherin mimicry by the capsid is a well-established concept in the full literature. The clearest iteration of this is a recent paper (Xue, 2023, 37355754) which covers most of the same HIV-related topics, in a comprehensive way, rendering much that is in this manuscript of limited additional significance - specifically Xue et al state in their paper’s title that “the HIV-1 capsid core is an opportunistic nuclear import receptor”. Interaction of the capsid with nucleoporins including FG repeats has also been investigated in depth by numerous other papers including e.g. Kane, 2018, 30084827, Matreyek, 2013, 24130490, Price, 2014, 25356722, Bhattacharya, 2014, 25518861, Bichel, 2013, 23902822, Lin, 2013, 23353830 and recently Shen, 2023, 36943880, and Shen, 2023, 36807645; this concept is also not particularly new, and structures of CA binding to FG repeats go back to 2014 (including a paper from some of the same authors). No major conceptual advance is made as to the capsid - FG Nups interaction as a key part of capsid interaction with the NPC, and, put in context with the published literature, the findings in these manuscripts are demonstrably only incremental. A key question in the field - does the capsid pass entirely through the NPC in vivo - is also not answered by this work.

On a related point, this same large body of literature demonstrates that Nup358, Nup214, Nup62, Pom121 and Nup153 are the key FG Nups for transport, and that while Nup98 may also be involved, it has been shown that it is not a key player. There is considerable literature already covering this including Xue et al., 2023, Dharan et al., 2020, Ao et al., 2012, Bichel et al., 2013, Matreyek et al., 2011, Price et al., 2014, Kane et al., 2018, Buffone et al., 2018. The sufficiency of Nup358, Nup153, and Nup62 for capsid entry to the NPC and demonstration that the capsid can enter the FG medium is for example thoroughly explored in a truly NPC-like model in Shen, 2023, 36807645. In contrast, this manuscript does not explore the ability of capsid to penetrate condensates or other media made of or containing these known players, instead they focus on this relatively unimportant player, Nup98. Nup98 is a key player in normal transport, but viruses do not play by the same rules, and many viruses even destroy or remove Nup98 from NPCs during their intracellular life cycle; and its depletion has been shown not to affect viral nuclear entry in vivo, unlike other key FG Nups. The authors included a table (S1) in the new version of their manuscript where they indicate that knock downs of Nup98 do impair infection in several papers. However, in those, the effect was either small to insignificant (Xue et al. 2023) or the papers indicated that the effect was probably not at import (König et al., 2008; Di Nunzio et al., 2012; Di Nunzio et al., 2013), so that table might be somewhat misleading as to the effect of Nup98 mutations in vivo.

Second, while interactions of capsid specifically with FG Nup condensates are shown, this is a model which is well demonstrated to absorb FG binders but remains controversial and largely untested in its role in the structure and facilitated transport in the NPC in vivo. Specifically, the assumption that demonstration of interactions with condensates mimics the in vivo NPC mechanisms is an extreme, unsupported claim. A paper recently showed this specific in vitro system likely bears little relevance

to the actual dynamic structure of the nuclear pore complex in vivo (Yu, 2023, PMID 37100914), also implying that the selectivity suggested in the papers under review is speculative and neither generally observed or of special significance. Instead, published work such as Shen, 2023, 36807645 use models that are far closer structurally to the NPC to show how the capsid can enter the NPC. Moreover, no in vivo data are presented to link the data presented with in vivo relevance of the findings. Thus, in terms of our understanding of nuclear transport and its hijacking by viruses, the findings presented are incremental and the manuscript does not rise to the level of novelty and mechanistic insight that one normally associates with this journal.

“MISREPRESENTATION OF THE LITERATURE”. We do not test monomeric or ‘oligomeric’ CA in our FFS assay. We only test capsid-like particles (CLPs). We also do not use the word ‘oligomeric’ when referring to CA, as this is an ambiguous term that could be misinterpreted to refer to hexamers, pentamers, tubes, spheres or CLPs.

Misrepresentation ... This is a significant criticism of the intent of the reviewer, which was to provide a dispassionate critique of the work. This is a contentious area, and naturally there are different views in it. The broad agreement of the arbiter reviewer with many of our points underscores that this dismissal as “misrepresentation” is unwarranted.

Comment: I am grateful to the authors for clarifying the intent of their language.

MISREPRESENTATION OF THE MANUSCRIPT - The comment that we claim that condensates do not destabilise the CA oligomers is false...

Again, this is not a misrepresentation. This goes to the heart of comments from e.g. the arbitration reviewer and above, as to the relevance of the specific in vitro Nup98 phase being tested to the transport mechanism of the capsid in the in vivo NPC, which the authors must justify.

COMMENT DIRECTED AT ANOTHER MANUCSCRIPT – We present no data on RanGTP. This comment appears to be addressed to the accompanying Fu et al. manuscript.

Agreed, and apologies – both manuscripts were reviewed simultaneously.

MISREPRESENTATION OF THE MANUSCRIPT – The term ‘self-translocating properties’ is not one that we use, and may indicate that the referee has misinterpreted our claims.

Again, the authors have assumed “misrepresentation” when it is simply a matter of clarification and a term used several times by the accompanying paper to describe the kap-like mechanism suggested for the capsid (apparently both manuscripts disagree on this matter). Indeed, the authors have sought to address this point to clarify their terminology usage; additional clarification would be welcome, including to distinguish the apparent different points of view between Dickson et al. and Fu et al..

MISREPRESENTATION OF THE LITERATURE – None of the four papers presented supports the referee’s assertion that the molecular mechanisms are already known.

I stated, “molecular recognition mechanism by which the capsid interacts with FG repeats”. The study cited and co-authored by Jacques presents a structure of FG repeats bound to CA and thus it is not in the least “disingenuous” to suggest the field have an excellent understanding of the molecular mechanism of this interaction – of course the interaction with all FG repeats has not been structurally mapped, as there are hundreds of FG repeat variant sequences. Do the authors believe or have evidence that the repeats of Nup98 interact with CA in a fundamentally different way from that discussed above and more generally in the literature? The current manuscript presents no new evidence for any different molecular mechanism of interaction, and barring other evidence at the structural level, it is fair to presume they are similar. The fact that Stacey, 2023, 37040417 show another structure of this interaction and that it is similar underscores the point that this current manuscript provides “very limited conceptual advance” over what has been already published about the “molecular recognition mechanism by which the capsid interacts with FG repeats”.

UNSUBSTANTIATED CLAIM – The referee states that CA is well understood to interact with a host of other cellular proteins that mediate interaction with the NPC.

Actually, this is a misunderstanding on the part of the authors. I stated “CA is well understood to interact with a host of other cellular proteins that mediate interaction with the NPC” and indeed it does – numerous other Nups (cellular vs viral proteins) mediate NPC interaction. And, many other proteins interact with CA. I did NOT state “the factors mentioned, CPSF6 and Sec24c, mediate interaction with the NPC”. What evidence can the authors provide that among many interactions with other FG repeat proteins, other nucleoporins, and other factors - all published to be key for capsid entry into the NPC - this one interaction with Nup98 is crucial?

FG-binding and Nup358 binding are not mutually exclusive. We could add a comment along these lines, but it is unclear what concern the referee is raising.

No, they are not. This underscores that multiple publications agree that the process of HIV entry into the NPC is a complex, multifactorial process of which functionally, Nup98 is only one and likely a minor at best (Xue, 2023, 37355754) player, unless the authors could provide experimental *in vivo* evidence that shows otherwise, i.e. that Nup98 is a key player in capsid entry to the NPC, which would then strongly support their claim.

MISREPRESENTATION OF THE LITERATURE – The interaction between CA and TNPO3...

MISREPRESENTATION OF THE MANUSCRIPT. It is disingenuous to state...

I agree that there is more recent evidence indicating TNPO3 is important for CPSF6 import; however TNPO3 is still important for HIV import, which was actually our main point and as again underscored in Xue, 2023, 37355754. Again, the overall point I was making is that “all these other host co-factors... have been shown to be involved in the actual mechanism of capsid entry to the NPC”. Crucially, the relative importance of the interaction of CA with Nup98 in general compared to the host of other proteins that have shown to be important *in vivo*, and the relevance of capsid partitioning into Nup98 phases, remains unestablished in this current manuscript. This must be

addressed experimentally, is not properly addressed currently, and so the “disingenuous” comment is unwarranted.

MISREPRESENTATION OF THE MANUSCRIPT – We investigate and/or discuss the relevant cellular proteins...

No experiments are shown indicating whether the Nups functionally most implicated in vivo in HIV entry to the NPC also form a phase, or other state, into which the capsid can partition, or whether this partitioning behavior is crucial in vivo. Thus, the authors have not fully contextualized the importance of their condensate experiments with how important this behavior is in vivo. The arbitration reviewer also raises similar points.

MISREPRESENTATION OF THE MANUSCRIPT – Again...

I have not ignored the author’s discussion and mention of this paper, but as explained below and in agreement with other referees, the authors, even in their manuscript title, gave me the distinct impression of having explained how HIV enters the NPC, and that is not what they are showing here, so proper rewording of their title and text would be suitable.

UNSUBSTANTIATED CLAIM – The referee bases their...

The authors themselves cite some of the “other work”, e.g. as well as Zila et al., 2021, there is Burdick et al. 2020, Li et al. 2021, Müller et al. 2021, and references in e.g. Shen et al., 2021, as well as references cited above, which should be included and contextualized.

4.1 MISREPRESENTATION OF THE LITERATURE – The referee...

Definitely not. I am reflecting the state of the field’s understanding, with which I am familiar. Even in their abstract and to justify their use of single FG Nup condensates, Dickson et al. literally say that ‘This [the nuclear pore’s diffusion] barrier is a phase-separated condensate in the central channel of the nuclear pore’, what it is not at all an accepted concept in the nuclear transport field, as the biophysical nature of the NPC permeability barrier is a still contentious question. In their rebuttal, the authors cherry-picked a sentence from a figure legend of Yu et al. Nature 2023, specifically Fig. 1. However, the findings of this remarkable paper are clearly stated in their Discussion, where they actually interpret and put into a proper context the findings of their study. Thus, the Yu et al., 2023 paper which the authors claim supports their “approach to modeling the permeability barrier” with Nup98 condensates actually states in its Discussion section: “despite having similar permeability barrier properties as the intact NPC, the bulk condensate formed from phase separating NUP98 is an incomplete approximation of the actual permeability barrier, the materials properties of which are modulated by the anchoring of a distinct number of FG-NUPs with 3D precision on a half-toroidal NPC scaffold. In terms of nuclear transport selectivity, there are consequences: whereas a surface condensate would leave a substantial hole at the center, we found the hole to be filled by FG-NUPs at near-critical conditions (Fig. 4b). These results emphasize the importance of interrogating the permeability barrier in situ to reconcile different transport models and understand the molecular basis for nuclear transport.”. It is easy to cherry-pick references that support the use of FG

condensates as a faithful model of transport – but there are many references that contradict this view, and just in recent papers that include literature reviews, the fact that this aspect of the transport mechanism remains unresolved and controversial is expounded e.g. in Hoogenboom, 2021; Cowburn, 2023; Huang, 2020, 32794558; Zheng, 2023, 36757893; Kalita, 2022, 35089308. Are the reviewers arguing against these experts in the field, and that this has actually been settled?

I agree that while the authors could not originally have discussed the Lim group paper as it appeared after submission of this work, however, it has been available during this review period, and the claimed irrelevance of the paper from the Lim group because it is in BioRxiv seems strange. Nature's instructions to authors:

<https://www.nature.com/nature/for-authors/formatting-guide>

clearly state as suitable references for their papers: “articles that have been published or accepted by a named publication, or that have been uploaded to a recognized preprint server (for example, arXiv, bioRxiv)”. The authors themselves submitted their own manuscript previously as a BioRxiv paper:

<https://doi.org/10.1101/2023.03.23.534032>

and they would presumably not argue that there was no worth in doing this, nor that their manuscript should have been ignored by the field. Thus, my critique and the need to respond to these new findings remain relevant.

From the publications listed by the authors to support ‘the reproducibility and broad acceptance of the FG-Nup phase separation model for understanding NPC transport selectivity’, Frey and Görlich, 2007; Hülsmann et al., 2012; Labokha et al., 2013; Mohr et al., 2009; Schmidt and Görlich, 2015; Ng et al., 2021 (6 out of the 7 articles mentioned in the author's rebuttal) are all studies from Prof. Görlich's laboratory, corresponding author in the related submitted manuscript. Although I do not doubt their integrity, the authors must admit that it also clearly shows that the FG condensate system is far from being a widely used and adopted system in the nuclear transport field to recapitulate the properties of a living NPC permeability barrier outside of that specific laboratory, as underscored by the above cited references and references within or related; if so, it would have undoubtedly been extensively used by other groups in the populated nuclear transport field that had more than 15 years to learn and adopt this method; and as I have stated elsewhere, the literature clearly underscores the disputed nature of the condensate as an accurate model for transport in vivo.

The point I am trying to emphasize is that it is clear that transport of HIV capsid through the NPC is far more complex than only a simple partition within a Nup98 condensate – while undoubtedly interactions with Nup98 play a role, e.g. {Di Nunzio, 2013, 23523133} this work and that of e.g. Xue et al. and Matreyek et al. show that other Nups play an extremely important role and that Nup98 (and its propensity to phase separate) may not be a particularly important player. I am questioning how much the in vitro partitioning actually represents what is going on in vivo, and these cited data and other papers cited above support this questioning of the relevance – or at least, significant limitations - of this in vitro model.

So as not to mislead non-experts, at the least the authors should clearly provide in the manuscript the important caveat that while the FG condensates in certain conditions could mimic some

macroscopic behaviors of a permeability barrier, they should not be considered to recapitulate the nanoscopic properties of a true NPC permeability barrier.

4.2 COMMENT DIRECTED AT ANOTHER MANUSCRIPT – We do not use spheres in our study.

I respectfully disagree, this comment is directly relevant to the manuscript originally entitled “Karyopherin mimicry explains how the HIV capsid penetrates nuclear pores”. Penetrate, as defined by the Cambridge dictionary: to move into or through something. Please do not get distracted by the mentioning of the 40 nm capsids used in the accompanying study, only included here as an example. The authors of this paper by Dickson et al. have not provided data showing that the Karyopherin-like behavior of proteins of the HIV capsid mediate the passage of such capsid through the NPC, and disprove other equally valid (if not better supported by their data) alternatives, like that it might just mediate the docking of the capsid to the NPC. That HIV capsids have been found inside of nuclei by other groups is not evidence supporting their title claim, it might be a fair speculation, suitable for a discussion section, but not for the title of the paper unless they could show further evidence. In this sense, I am not asking anything much different from point 1.7 of Referee #1, with whom I agree.

4.3 CONTRADICTION – The referee states...

Clarification is needed as to which features of capsid interaction mimic karyopherins, and which are different - the way they interact with FG repeats, that karyopherins freely shuttle across NPCs whereas karyopherins do not, that capsid presumably has hundreds of FG interaction sites whereas karyopherins do not. In other words, it should be made clear which aspects of transport factors the capsid mimics, and which it does not. HIV, like other viruses, appropriates and even bypasses normal cellular processes, and so assumptions of normal transport mechanisms may not apply.

4.4 Nup98 condensates do indeed age...

The clarification provided is appropriate, and it would be very useful for future reproducibility that the authors indicate in the methods section what is the time point where gelation and potential issues due to amyloid formation start to be observed in this system.

4.5 This is a consequence of experimental conditions ...

It would be helpful if the authors could add the clarification they provide here to the manuscript, as the accompanying manuscript seems to present data going into a different direction, that is, that the presence of transport factors do not affect the partitioning of viral-like particles.

4.6 The FFS approach relies...

The clarification – and experimental limitations - provided here would be useful as qualifiers in the main text.

4.7 UNSUBSTANTIATED CLAIM – Our model...

I might be confused by the idiosyncrasies of the method used by the authors, but let me try to rephrase my question. First, karyopherins in vivo are expected to show multivalency in their interaction with FGs, but not high avidity. High avidity would cause the whole system to collapse in the central channel as karyopherins would not be able to disengage from the FGs fast enough to ensure a fast and dynamic transport. High avidity of Kaps has been shown to be a potential artifact of certain in vitro measurements (e.g., Kapinos et al., 2014 24739174) and that low per-FG affinity and the enthalpy–entropy balance prevent high-avidity interaction between FG Nups and karyopherins (e.g. Hayama et al., 2018 29374059). Thus, a karyopherin establishes multiple, weak, transient interactions that ensure specificity and fast enough transport. The author’s data show, as they clearly state through the manuscript, high avidity in their assay, which would promote binding of the capsid to the NPC, but not a Kap-like behavior allowing transport across the NPC. Their in vitro system could then be showing that Nup affinity, but not Kap-like (or nuclear transport factor-like) behavior is what drives the interaction of HIV particles with the NPC. Something that is, from my point of view (apparently shared by the arbitrating referee) insufficiently ground-breaking or novel enough as it has been repeatedly shown in the literature, as rightly indicated by the authors themselves in the introduction.

Second, Nup153 1407-1423 is described by the authors in their reply as a “single FG-containing peptide (Nup153, residues 1407-1423)”. The FFS signal from that peptide is shown in Fig. 2C, where it is normalized to the wt value (whatever that is). However, the authors claim that “a single FG peptide is below the detection limit for binding” (“below the detection limit for FFS” in the manuscript) when Nup98 fragments are tested (Fig. 2I; Nup9873-88 and 342-356). Why are some peptides containing a single FG (Nup98) below the limit of detection of their method and others (Nup153) are not? Is this indicating some type of method limitation? FG linker difference? As in Fig. 2B-C the ratios are normalized to the wt, it is hard to see what the non-normalized signal from a known single FG binder would be, and that is a reference/control that would be interesting to add in 2I. So I would suggest to redo the measurements in 2I including those known binding peptides (CPSF6313-327 and Nup1531407-1423) as a reference and if their signal is higher than that of the Nup98 single FG peptides maybe the authors could provide an explanation to this behavior. Also, fragments of Nup98 containing the same number of FGs (13, Fig. 2I) show FFS intensity ratios that differ by a factor of 2. Should one expect such a difference if they contain the same number of binding sites? Is it possible that other interactions (not phenylalanine related) might be at play?

4.8 The hexamer represents a model CA system...

Agreed, and this point probably doesn’t need adding to the text.

4.9 We note that phenylalanine mutation...

The argument might be compelling, but not formally demonstrated in the authors’ method and setup and in the specific Nups used in this study. The data shown in Fig. 2K is useful to, under the conditions tested, discard interactions with structured regions of those Nups, but nothing else. I still recommend that to formally prove the “FG-specific” point (manuscript line #157) and strengthen their conclusion the authors should for example generate a phenylalanine-to-serine or phenylalanine-to-tyrosine mutant of at least one FG Nup representative (one of the Nup98 fragments used in figure 2I?) and show that binding is abolished, as it is commonly done to show

phenylalanine-dependent transport factor interactions (see PMID: 17897934; PMID: 17082456 for examples).

4.10 We have rephrased...

Thanks.

4.11 We have corrected...

Thanks.

Referee #5 (Remarks to the Author):

The field of HIV nuclear import has been highly contentious despite decades of research and multiple reports, many of these in the highest impact journals. Seminal work nearly 20 years ago from Emerman and colleagues (ref 3) first highlighted HIV capsid as the key mediator, debunking at the time earlier work focused on Vpr, matrix, integrase, and “the central DNA flap”. The work in this paper and the accompanying Fu manuscript now provide the biochemical and biological basis for capsid-mediated HIV nuclear import. These papers together represent a transformative advance for the field and should be published in Nature. It is unjustified to suggest that these studies advance the field incrementally.

The revised paper admirably addressed the reviewers comments, adding important N57D mutant to Fig 4 and protein quality gels/images (new Fig S5). My remaining comments are comparatively minor.

1) Please provide additional clarity for added S5-1. Methods are lacking for the in vitro translation technique (ref 3 is cited). The FFS method indicates 50 nM GFP-Nup, so authors can track specific GFP-Nup concentration. Why then is there so much variation in Fig S5-1 signal intensities? Most proteins >100 kDa appear under loaded, with Nup153 upper gel barely discernable. How was gel loading normalized? Ideally, this should be equal mass or molarity. Please clarify.

2) Line 29, “their DNA” is technically incorrect. These are RNA viruses.

3) Line 30, “occurs during mitosis” has been directly shown for MLV. To soften the generalization, I would suggest: “...retroviruses, mitosis provides the opportunity to interact with chromosomal DNA during nuclear envelope breakdown”

4) The new ED figure, S5, is cited before the other ED figs lines 78-79. Please renumber the figures to reflect the order by which they are cited in the main text.

5) Lines 135, 136 are garbled. Please rewrite.

6) Please cite ref 15 alongside ref 16 line 152.

7) Add close parenthesis line 221.

8) Please also cite ref 15 line 272.

Author Rebuttals to First Revision:

Please find below a point-by-point response (in blue) to referee comments to manuscript 2023-03-04927B. Please note there was no 'Referee #2'.

Referee #1 (Remarks to the Author):

In this revised manuscript, Dickson and coworkers have improved their study by including a new Table and included results with a mutant capsid-like particle (N57D). The latter result adds some support for the biological relevance of the condensate internalization assay with CLPs. The study remains quite innovative in that it presents a new view of HIV-1 nuclear entry that is likely to be intensively tested in future studies.

We thank the referee for reviewing our manuscript and for their constructive feedback.

Authors declined to test the internalization of native HIV-1 cores, expressing concern about potentially uninformative outcomes. Their worry that cores would disintegrate over the timescale of the experiment seems ungrounded based on the well-documented stabilization of HIV-1 cores upon the addition of physiological concentrations of inositol hexakisphosphate. Several groups have published studies employing purified HIV cores, so this is feasible. I agree that the process needs to be studied carefully, but this experiment seems worth performing to further support the biological relevance of the CLP condensate penetration results.

We agree that this is an important question, and we maintain that the experiment is not trivial. To clarify, the issue of stability was not about whether we can maintain cores in solution - we ourselves are amongst those who have published that they can be stabilised for days in the presence of IP6. Rather, the possibility that FG-Nup immersion may act as an uncoating trigger would be a major discovery in its own right, and deserves a rigorous investigation. In previous studies, we and others have already found that FG-binding site saturation with PF74 or Lenacapavir leads to CA lattice hyperstabilisation at the cost of containment, and we think it is entirely feasibly that Nup saturation could have similarly complex effects on capsid stability. As the editor has not required us to test native cores, we have not presented additional data, but we do intend to follow this up in future work.

Authors included new condensate results with one mutant CLP. While this is helpful, it is not a very thorough test of biological relevance, and inclusion of additional mutants known to be impaired for nuclear entry is needed to support the major claim of the study. Moreover, the authors' statement

that the condensate entry assay is not quantitative is surprising, given that quantitative results were shown in Fig. 4g.

We have focused our efforts on the mutants most well-characterised to affect the interaction between capsid and FG-Nup. We agree that there are other mutations that can influence nuclear entry, but these can be indirect due to the loss of CA flexibility (such as the E45A mutant previously suggested by the referee), or altered cofactor usage. The role of capsid stability/flexibility and cofactor usage will ultimately prove important to fully understanding HIV nuclear entry, but are beyond the scope of this current work. As the editor has not required us to test more capsid mutants, we present no additional data.

The authors stated the meaning of the error bars in the respective figure legends. However, though they provided p values for some results, I was unable to find a statement regarding the statistical tests employed. This information should be included.

This information has been added in the Figure 3 legend and a Statistics and Reproducibility section has been added following the methods.

Minor point: The authors may wish to consider including PMID:20227665 in Table S1. This 2010 paper was a landmark study in the field of HIV-1 nuclear entry.

This citation has been included in Supplementary Table 1.

Referee #3 (Remarks to the Author):

In my initial review of the manuscript by Dickson et.al. I was overall positive, and suggested that the authors address a set of comments and critiques that would then make the paper suitable for publication. I am satisfied with the responses to these points I made previously.

In reading the comments of the other reviewers, I see there is a robust discussion of other aspects of the manuscript, and which seems worth considering, but which I am not qualified to speak about.

We thank the referee for reviewing our manuscript and for their support.

Referee #4 (Remarks to the Author):

In responding to Referee #4's comments below, we have made the following changes to the manuscript:

1) We have adjusted the title to read 'The HIV capsid mimics karyopherin engagement of FG-nucleoporins'. We had intended the original phrase '*explains how the HIV capsid penetrates nuclear pores*' to refer to how our work could be placed in the context of prior attempts at understanding this process. The new title better reflects what was performed in this specific paper.

2) We have explicitly acknowledged the limitations of using Nup98 condensates to probe nuclear entry (see fourth paragraph of Discussion) and acknowledge that real NPCs are immensely complex macromolecular entities. We have adjusted our language throughout the text to be clear about what can be directly concluded from the data, and what is extrapolation or speculation about the process of nuclear entry. We trust that this serves to demonstrate that a complete understanding of HIV nuclear entry *in vivo* was beyond the scope of the current work.

3) We have included reference to the contentious nature of the diffusion barrier in the discussion, accompanied by citations suggested by the referee.

While we disagree with many of the referee's assertions and interpretations of the literature, we believe that science is best served by open peer review. It is our hope that the robust challenges on display here may prompt others to bring additional perspectives and approaches to further explore the process of viral nuclear entry.

The authors have still not addressed the key issues properly, largely choosing to argue with these and the significance of published findings rather than provide experimental data to support their points. These issues can be summarized as follows:

The first and perhaps most critical issue is of significance. The authors state "this karyopherin mimicry model resolves a key conceptual challenge for the role of the HIV capsid in nuclear entry". However, this is not the case – karyopherin mimicry by the capsid is a well-established concept in the full literature. The clearest iteration of this is a recent paper (Xue, 2023, 37355754) which covers most of the same HIV-related topics, in a comprehensive way, rendering much that is in this manuscript of limited additional significance - specifically Xue et al state in their paper's title that

“the HIV-1 capsid core is an opportunistic nuclear import receptor”. Interaction of the capsid with nucleoporins including FG repeats has also been investigated in depth by numerous other papers including e.g. Kane, 2018, 30084827, Matreyek, 2013, 24130490, Price, 2014, 25356722, Bhattacharya, 2014, 25518861, Bichel, 2013, 23902822, Lin, 2013, 23353830 and recently Shen, 2023, 36943880, and Shen, 2023, 36807645; this concept is also not particularly new, and structures of CA binding to FG repeats go back to 2014 (including a paper from some of the same authors). No major conceptual advance is made as to the capsid – FG Nups interaction as a key part of capsid interaction with the NPC, and, put in context with the published literature, the findings in these manuscripts are demonstrably only incremental. A key question in the field – does the capsid pass entirely through the NPC in vivo – is also not answered by this work.

On a related point, this same large body of literature demonstrates that Nup358, Nup214, Nup62, Pom121 and Nup153 are the key FG Nups for transport, and that while Nup98 may also be involved, it has been shown that it is not a key player. There is considerable literature already covering this including Xue et al., 2023, Dharan et al., 2020, Ao et al., 2012, Bichel et al., 2013, Matreyek et al., 2011, Price et al., 2014, Kane et al., 2018, Buffone et al., 2018. The sufficiency of Nup358, Nup153, and Nup62 for capsid entry to the NPC and demonstration that the capsid can enter the FG medium is for example thoroughly explored in a truly NPC-like model in Shen, 2023, 36807645. In contrast, this manuscript does not explore the ability of capsid to penetrate condensates or other media made of or containing these known players, instead they focus on this relatively unimportant player, Nup98. Nup98 is a key player in normal transport, but viruses do not play by the same rules, and many viruses even destroy or remove Nup98 from NPCs during their intracellular life cycle; and its depletion has been shown not to affect viral nuclear entry in vivo, unlike other key FG Nups. The authors included a table (S1) in the new version of their manuscript where they indicate that knock downs of Nup98 do impair infection in several papers. However, in those, the effect was either small to insignificant (Xue et al. 2023) or the papers indicated that the effect was probably not at import (König et al., 2008; Di Nunzio et al., 2012; Di Nunzio et al., 2013), so that table might be somewhat misleading as to the effect of Nup98 mutations in vivo.

Second, while interactions of capsid specifically with FG Nup condensates are shown, this is a model which is well demonstrated to absorb FG binders but remains controversial and largely untested in its role in the structure and facilitated transport in the NPC in vivo. Specifically, the assumption that demonstration of interactions with condensates mimics the in vivo NPC mechanisms is an extreme, unsupported claim. A paper recently showed this specific in vitro system likely bears little relevance to the actual dynamic structure of the nuclear pore complex in vivo (Yu, 2023, PMID 37100914), also implying that the selectivity suggested in the papers under review is speculative and neither generally observed or of special significance. Instead, published work such as Shen, 2023, 36807645 use models that are far closer structurally to the NPC to show how the capsid can enter the NPC. Moreover, no in vivo data are presented to link the data presented with in vivo relevance of the findings. Thus, in terms of our understanding of nuclear transport and its hijacking by viruses, the findings presented are incremental and the manuscript does not rise to the level of novelty and mechanistic insight that one normally associates with this journal.

We disagree with the referee's interpretation of prior HIV literature. We thank Referee #5 for their comment that it is unjustified that our study advances the field only incrementally.

"MISREPRESENTATION OF THE LITERATURE". We do not test monomeric or 'oligomeric' CA in our FFS assay. We only test capsid-like particles (CLPs). We also do not use the word 'oligomeric' when referring to CA, as this is an ambiguous term that could be misinterpreted to refer to hexamers, pentamers, tubes, spheres or CLPs.

Misrepresentation ... This is a significant criticism of the intent of the reviewer, which was to provide a dispassionate critique of the work. This is a contentious area, and naturally there are different views in it. The broad agreement of the arbiter reviewer with many of our points underscores that this dismissal as "misrepresentation" is unwarranted.

We were not shown details of the arbitrating reviewer's assessment and stand by our original comment.

Comment: I am grateful to the authors for clarifying the intent of their language.

MISREPRESENTATION OF THE MANUSCRIPT— The comment that we claim that condensates do not destabilise the CA oligomers is false...

Again, this is not a misrepresentation. This goes to the heart of comments from e.g. the arbitration reviewer and above, as to the relevance of the specific in vitro Nup98 phase being tested to the transport mechanism of the capsid in the in vivo NPC, which the authors must justify.

We were not shown details of the arbitrating reviewer's assessment and stand by our original comment.

COMMENT DIRECTED AT ANOTHER MANUSCRIPT – We present no data on RanGTP. This comment appears to be addressed to the accompanying Fu et al. manuscript.

Agreed, and apologies – both manuscripts were reviewed simultaneously.

We appreciate the referee acknowledging this error.

MISREPRESENTATION OF THE MANUSCRIPT – The term ‘self-translocating properties’ is not one that we use, and may indicate that the referee has misinterpreted our claims.

Again, the authors have assumed “misrepresentation” when it is simply a matter of clarification and a term used several times by the accompanying paper to describe the kap-like mechanism suggested for the capsid (apparently both manuscripts disagree on this matter). Indeed, the authors have sought to address this point to clarify their terminology usage; additional clarification would be welcome, including to distinguish the apparent different points of view between Dickson et al. and Fu et al..

It is important to acknowledge that the two papers are independent concurrent studies, and we have not sought to adjust findings or interpretation based on Fu et al.

MISREPRESENTATION OF THE LITERATURE – None of the four papers presented supports the referee’s assertion that the molecular mechanisms are already known.

I stated, “molecular recognition mechanism by which the capsid interacts with FG repeats”. The study cited and co-authored by Jacques presents a structure of FG repeats bound to CA and thus it is not in the least “disingenuous” to suggest the field have an excellent understanding of the molecular mechanism of this interaction – of course the interaction with all FG repeats has not been structurally mapped, as there are hundreds of FG repeat variant sequences. Do the authors believe or have evidence that the repeats of Nup98 interact with CA in a fundamentally different way from that discussed above and more generally in the literature? The current manuscript presents no new evidence for any different molecular mechanism of interaction, and barring other evidence at the structural level, it is fair to presume they are similar. The fact that Stacey, 2023, 37040417 show another structure of this interaction and that it is similar underscores the point that this current manuscript provides “very limited conceptual advance” over what has been already published about the “molecular recognition mechanism by which the capsid interacts with FG repeats”.

While the *molecular recognition mechanism* of CA with a particular FG-peptide in Nup153 has been shown by Price et al. (co-authored by Jacques) it is indeed disingenuous to assume that this same mechanism applies to all FG-peptides in all FG-Nups. We provide a systematic study investigating the binding mode of all FG-Nups and put these findings into context of previous studies in Supplementary table 1.

We disagree with the referee's interpretation of the prior literature and stand by our original comment.

UNSUBSTANTIATED CLAIM – The referee states that CA is well understood to interact with a host of other cellular proteins that mediate interaction with the NPC.

Actually, this is a misunderstanding on the part of the authors. I stated “CA is well understood to interact with a host of other cellular proteins that mediate interaction with the NPC” and indeed it does – numerous other Nups (cellular vs viral proteins) mediate NPC interaction. And, many other proteins interact with CA. I did NOT state “the factors mentioned, CPSF6 and Sec24c, mediate interaction with the NPC”. What evidence can the authors provide that among many interactions with other FG repeat proteins, other nucleoporins, and other factors— all published to be key for capsid entry into the NPC— this one interaction with Nup98 is crucial?

Our rationale for the use of Nup98 is presented at line 177 and has remain unchanged since original submission.

FG-binding and Nup358 binding are not mutually exclusive. We could add a comment along these lines, but it is unclear what concern the referee is raising.

No, they are not. This underscores that multiple publications agree that the process of HIV entry into the NPC is a complex, multifactorial process of which functionally, Nup98 is only one and likely a minor at best (Xue, 2023, 37355754) player, unless the authors could provide experimental in vivo evidence that shows otherwise, i.e. that Nup98 is a key player in capsid entry to the NPC, which would then strongly support their claim.

We disagree with the referee's interpretation of the prior literature and stand by our original comment.

MISREPRESENTATION OF THE LITERATURE – The interaction between CA and TNPO3...

MISREPRESENTATION OF THE MANUSCRIPT. It is disingenuous to state...

I agree that there is more recent evidence indicating TNPO3 is important for CPSF6 import; however TNPO3 is still important for HIV import, which was actually our main point and as again underscored in Xue, 2023, 37355754. Again, the overall point I was making is that “all these other host co-factors... have been shown to be involved in the actual mechanism of capsid entry to the NPC”. Crucially, the relative importance of the interaction of CA with Nup98 in general compared to the host of other proteins that have shown to be important in vivo, and the relevance of capsid partitioning into Nup98 phases, remains unestablished in this current manuscript. This must be addressed experimentally, is not properly addressed currently, and so the “disingenuous” comment is unwarranted.

We stand by our original comment, we discuss these co-factors and their potential role for nuclear transport throughout the manuscript.

MISREPRESENTATION OF THE MANUSCRIPT – We investigate and/or discuss the relevant cellular proteins...

No experiments are shown indicating whether the Nups functionally most implicated in vivo in HIV entry to the NPC also form a phase, or other state, into which the capsid can partition, or whether this partitioning behavior is crucial in vivo. Thus, the authors have not fully contextualized the importance of their condensate experiments with how important this behavior is in vivo. The arbitrating reviewer also raises similar points.

We acknowledge that we have not explicitly tested the phase-entry behaviour of each FG-Nup. We have, however, shown in figures 1 and 2 that the capsid interacts with them using the same interfaces as it does with Nup98. Our justification for focusing on Nup98 is presented at line 177. We were not shown details of the arbitrating reviewer’s assessment.

MISREPRESENTATION OF THE MANUSCRIPT – Again...

I have not ignored the author’s discussion and mention of this paper, but as explained below and in agreement with other referees, the authors, even in their manuscript title, gave me the distinct impression of having explained how HIV enters the NPC, and that is not what they are showing here, so proper rewording of their title and text would be suitable.

We have adjusted the title (see above) and made text changes throughout the manuscript that qualify that all such details are not yet known.

UNSUBSTANTIATED CLAIM – The referee bases their...

The authors themselves cite some of the “other work”, e.g. as well as Zila et al., 2021, there is Burdick et al. 2020, Li et al. 2021, Müller et al. 2021, and references in e.g. Shen et al., 2021, as well as references cited above, which should be included and contextualized.

We have faithfully and comprehensively cited the HIV literature and placed our study in their context (including Zila et al., 2021 , Burdick et al., 2020, Müller et al. 2021).

MISREPRESENTATION OF THE LITERATURE – The referee...

Definitely not. I am reflecting the state of the field’s understanding, with which I am familiar. Even in their abstract and to justify their use of single FG Nup condensates, Dickson et al. literally say that ‘This [the nuclear pore’s diffusion] barrier is a phase-separated condensate in the central channel of the nuclear pore’, what it is not at all an accepted concept in the nuclear transport field, as the biophysical nature of the NPC permeability barrier is a still contentious question. In their rebuttal, the authors cherry-picked a sentence from a figure legend of Yu et al. Nature 2023, specifically Fig. 1. However, the findings of this remarkable paper are clearly stated in their Discussion, where they actually interpret and put into a proper context the findings of their study. Thus, the Yu et al., 2023 paper which the authors claim supports their “approach to modeling the permeability barrier” with Nup98 condensates actually states in its Discussion section: “despite having similar permeability barrier properties as the intact NPC, the bulk condensate formed from phase separating NUP98 is an incomplete approximation of the actual permeability barrier, the materials properties of which are modulated by the anchoring of a distinct number of FG-NUPs with 3D precision on a half-toroidal NPC scaffold. In terms of nuclear transport selectivity, there are consequences: whereas a surface condensate would leave a substantial hole at the center, we found the hole to be filled by FG-NUPs at near-critical conditions (Fig. 4b). These results emphasize the importance of interrogating the permeability barrier in situ to reconcile different transport models and understand the molecular basis for nuclear transport.”. It is easy to cherry-pick references that support the use of FG condensates as a faithful model of transport – but there are many references that contradict this view, and just in recent papers that include literature reviews, the fact that this aspect of the transport mechanism remains unresolved and controversial is expounded e.g. in Hoogenboom, 2021; Cowburn, 2023; Huang, 2020, 32794558; Zheng, 2023, 36757893; Kalita, 2022, 35089308. Are the reviewers arguing against these experts in the field, and that this has actually been settled?

We agree that our previous description of the NPC's diffusion barrier did not acknowledge all perspectives of its nanoscopic structure. We have included citation of references Hoogenboom, 2021; ; Huang, 2020; Kalita, 2022 in the discussion to better represent the debate ongoing within the NPC field as to the exact nature of diffusion barrier and the underlying physicochemical principles. We have also adjusted the language in the abstract and main text to avoid describing the diffusion barrier definitively as a phase-separated condensate. We have also included a comment discussing that, due to the NPC complexity, any reduced model system will be an approximation and cite Yu et al. 2023.

I agree that while the authors could not originally have discussed the Lim group paper as it appeared after submission of this work, however, it has been available during this review period, and the claimed irrelevance of the paper from the Lim group because it is in BioRxiv seems strange. Nature's instructions to authors:

<https://www.nature.com/nature/for-authors/formatting-guide>

clearly state as suitable references for their papers: "articles that have been published or accepted by a named publication, or that have been uploaded to a recognized preprint server (for example, arXiv, bioRxiv)". The authors themselves submitted their own manuscript previously as a BioRxiv paper:

<https://doi.org/10.1101/2023.03.23.534032>

and they would presumably not argue that there was no worth in doing this, nor that their manuscript should have been ignored by the field. Thus, my critique and the need to respond to these new findings remain relevant.

We stand by our original comment that preprints that have not been subject to peer review should not be used to steer a peer review process or to support a claim of what is generally accepted in the field.

From the publications listed by the authors to support 'the reproducibility and broad acceptance of the FG-Nup phase separation model for understanding NPC transport selectivity', Frey and Görlich, 2007; Hülsmann et al., 2012; Labokha et al., 2013; Mohr et al., 2009; Schmidt and Görlich, 2015; Ng et al., 2021 (6 out of the 7 articles mentioned in the author's rebuttal) are all studies from Prof. Görlich's laboratory, corresponding author in the related submitted manuscript. Although I do not doubt their integrity, the authors must admit that it also clearly shows that the FG condensate system is far from being a widely used and adopted system in the nuclear transport field to recapitulate the properties of a living NPC permeability barrier outside of that specific laboratory, as underscored by the above cited references and references within or related; if so, it would have undoubtedly been extensively used by other groups in the populated nuclear transport field that had

more than 15 years to learn and adopt this method; and as I have stated elsewhere, the literature clearly underscores the disputed nature of the condensate as an accurate model for transport in vivo.

It is important to acknowledge that our study and Fu et al. were conceived, completed and submitted independently, without prior knowledge of the concurrent studies. Cited publications have been chosen by their scientific merit. We were first made aware of Fu et al. after receiving the first round of referees' comment.

The point I am trying to emphasize is that it is clear that transport of HIV capsid through the NPC is far more complex than only a simple partition within a Nup98 condensate – while undoubtedly interactions with Nup98 play a role, e.g. {Di Nunzio, 2013, 23523133} this work and that of e.g. Xue et al. and Matreyek et al. show that other Nups play an extremely important role and that Nup98 (and its propensity to phase separate) may not be a particularly important player. I am questioning how much the in vitro partitioning actually represents what is going on in vivo, and these cited data and other papers cited above support this questioning of the relevance – or at least, significant limitations - of this in vitro model.

So as not to mislead non-experts, at the least the authors should clearly provide in the manuscript the important caveat that while the FG condensates in certain conditions could mimic some macroscopic behaviors of a permeability barrier, they should not be considered to recapitulate the nanoscopic properties of a true NPC permeability barrier.

We have made numerous changes throughout the manuscript and have added a comment as to the limitations of our approach in the discussion along with accompanying references (see above).

COMMENT DIRECTED AT ANOTHER MANUSCRIPT – *We do not use spheres in our study.*

I respectfully disagree, this comment is directly relevant to the manuscript originally entitled “Karyopherin mimicry explains how the HIV capsid penetrates nuclear pores”. Penetrate, as defined by the Cambridge dictionary: to move into or through something. Please do not get distracted by the mentioning of the 40 nm capsids used in the accompanying study, only included here as an example. The authors of this paper by Dickson et al. have not provided data showing that the Karyopherin-like behavior of proteins of the HIV capsid mediate the passage of such capsid through the NPC, and disprove other equally valid (if not better supported by their data) alternatives, like that it might just mediate the docking of the capsid to the NPC. That HIV capsids have been found inside of nuclei by other groups is not evidence supporting their title claim, it might be a fair speculation, suitable for a

discussion section, but not for the title of the paper unless they could show further evidence. In this sense, I am not asking anything much different from point 1.7 of Referee #1, with whom I agree.

We have adjusted the title to remove the word 'penetrate' (see above), and restrict speculation and contextualisation of our work within the broader NPC entry literature to the discussion.

CONTRADICTION – The referee states...

Clarification is needed as to which features of capsid interaction mimic karyopherins, and which are different - the way they interact with FG repeats, that karyopherins freely shuttle across NPCs whereas karyopherins do not, that capsid presumably has hundreds of FG interaction sites whereas karyopherins do not. In other words, it should be made clear which aspects of transport factors the capsid mimics, and which it does not. HIV, like other viruses, appropriates and even bypasses normal cellular processes, and so assumptions of normal transport mechanisms may not apply.

We discuss mimicry of key transport factor properties by CA throughout the discussion and explicitly describe our karyopherin mimicry model in the concluding paragraph.

Nup98 condensates do indeed age...

The clarification provided is appropriate, and it would be very useful for future reproducibility that the authors indicate in the methods section what is the time point where gelation and potential issues due to amyloid formation start to be observed in this system.

We are pleased to have provided clarity on this issue. We have not explicitly explored the precise timing of gelation or possible amyloid formation, but have controlled for the absence of these effects in the methods as described. We also note the absence of any observable amyloid structure in the cryo-electron tomograms of Nup98 condensates.

This is a consequence of experimental conditions ...

It would be helpful if the authors could add the clarification they provide here to the manuscript, as the accompanying manuscript seems to present data going into a different direction, that is, that the presence of transport factors do not affect the partitioning of viral-like particles.

In the main text we have added the phrase '*under our experimental conditions*' at line 232, and add the following explanatory sentence at line 234: '*Notably, a fully assembled capsid with >1000 FG-binding sites will likely be a much stronger competitor for the FG-motifs than CA hexamers.*' It is important to acknowledge that the two papers are independent concurrent studies, and we have not sought to adjust findings or interpretation based on Fu et al.

The FFS approach relies...

The clarification – and experimental limitations - provided here would be useful as qualifiers in the main text.

We have left the text unchanged as the FFS strengths are already presented in the manuscript and we cite the relevant methods paper which explains the method in detail (Lau et al.)

UNSUBSTANTIATED CLAIM – Our model...

I might be confused by the idiosyncrasies of the method used by the authors, but let me try to rephrase my question. First, karyopherins in vivo are expected to show multivalency in their interaction with FGs, but not high avidity. High avidity would cause the whole system to collapse in the central channel as karyopherins would not be able to disengage from the FGs fast enough to ensure a fast and dynamic transport. High avidity of Kaps has been shown to be a potential artifact of certain in vitro measurements (e.g., Kapinos et al., 2014 24739174) and that low per-FG affinity and the enthalpy–entropy balance prevent high-avidity interaction between FG Nups and karyopherins (e.g. Hayama et al., 2018 29374059). Thus, a karyopherin establishes multiple, weak, transient interactions that ensure specificity and fast enough transport. The author's data show, as they clearly state through the manuscript, high avidity in their assay, which would promote binding of the capsid to the NPC, but not a Kap-like behavior allowing transport across the NPC. Their in vitro system could then be showing that Nup affinity, but not Kap-like (or nuclear transport factor-like) behavior is what drives the interaction of HIV particles with the NPC. Something that is, from my point of view (apparently shared by the arbitrating referee) insufficiently ground-breaking or novel enough as it has been repeatedly shown in the literature, as rightly indicated by the authors themselves in the introduction.

We agree that ‘high avidity’ at line 88 in the brief overview of the FFS method is misleading with respect to the apparent affinity, especially given the usage of the term in the NPC field. We have rephrased as ‘... the CLP lattice can engage binders that rely on multivalent CA contacts to make detectable capsid interactions’. To avoid confusion the two other uses of ‘avidity’ in the main text have been replaced with ‘multivalency’ at lines 73 and 162.

Second, Nup153 1407-1423 is described by the authors in their reply as a “single FG-containing peptide (Nup153, residues 1407-1423)”. The FFS signal from that peptide is shown in Fig. 2C, where it is normalized to the wt value (whatever that is). However, the authors claim that “a single FG peptide is below the detection limit for binding” (“below the detection limit for FFS” in the manuscript) when Nup98 fragments are tested (Fig. 2I; Nup9873-88 and 342-356). Why are some peptides containing a single FG (Nup98) below the limit of detection of their method and others (Nup153) are not? Is this indicating some type of method limitation? FG linker difference? As in Fig. 2B-C the ratios are normalized to the wt, it is hard to see what the non-normalized signal from a known single FG binder would be, and that is a reference/control that would be interesting to add in 2I. So I would suggest to redo the measurements in 2I including those known binding peptides (CPSF6313-327 and Nup1531407-1423) as a reference and if their signal is higher than that of the Nup98 single FG peptides maybe the authors could provide an explanation to this behavior.

Also, fragments of Nup98 containing the same number of FGs (13, Fig. 2I) show FFS intensity ratios that differ by a factor of 2. Should one expect such a difference if they contain the same number of binding sites? Is it possible that other interactions (not phenylalanine related) might be at play?

As we show in figure 2a, Nup153₁₄₀₇₋₁₄₂₃ and CPSF6₃₁₃₋₃₂₇ make additional contacts with the capsid that are mediated by residues N-terminal to the FG motif. This results in tighter binding than FG-alone. These details and relevant mutational studies are described in Price 2014. Nup98 has no equivalent ‘high-affinity FG’ region. This is the power of the FFS measurement – it allows us to probe the multivalent interactions between FG-repeat and CA lattice, even though some single FG containing peptides may be low affinity binders and beyond the detection limit. Nup153₁₄₀₇₋₁₄₂₃, CPSF6₃₁₃₋₃₂₇ and the Nup98 peptides were tested side by side as the reviewer suggests, confirming much weaker binding of the Nup98 peptides. We did not represent the data this way as this experiment focused on the dissection of Nup98 into single FG-peptides and FG-containing fragments. The peptide data is publicly available on Dryad.

The Nup98 results are key findings, showing that one cannot assume that the well-studied interaction of CA with Nup153₁₄₀₇₋₁₄₂₃ is representative of all CA-FG interaction. Indeed, Nup153₁₄₀₇₋₁₄₂₃, while widely recognised as a CA binder, is the exception rather than the rule due to its relatively high affinity. The residues between FG motifs can, therefore, influence total binding either through additional interactions, or through modulation of FG accessibility. In the case of the Nup98 fragments, it is remarkable that fragments of such different lengths are so similar in binding. The

observed differences are likely to be due to relative FG-density as well as sequence differences surrounding the FG motifs. As the residual binding is identical (effectively zero) for the N57D mutant, this disfavours the existence of a cryptic binding mode in addition to direct FG interaction. Importantly, all three fragments bind wild-type CLPs, suggesting that multiple FG-repeats contribute to binding rather than there being a single high-affinity binding site.

The hexamer represents a model CA system...

Agreed, and this point probably doesn't need adding to the text.

No further changes made.

We note that phenylalanine mutation...

The argument might be compelling, but not formally demonstrated in the authors' method and setup and in the specific Nups used in this study. The data shown in Fig. 2K is useful to, under the conditions tested, discard interactions with structured regions of those Nups, but nothing else. I still recommend that to formally prove the "FG-specific" point (manuscript line #157) and strengthen their conclusion the authors should for example generate a phenylalanine-to-serine or phenylalanine-to-tyrosine mutant of at least one FG Nup representative (one of the Nup98 fragments used in figure 2I?) and show that binding is abolished, as it is commonly done to show phenylalanine-dependent transport factor interactions (see PMID: 17897934; PMID: 17082456 for examples).

We stand by our original argument that this control is unnecessary considering the internal consistency of our results and prior work in this space.

We have rephrased...

Thanks.

No further changes made.

We have corrected...

Thanks.

No further changes made.

Referee #5 (Remarks to the Author):

The field of HIV nuclear import has been highly contentious despite decades of research and multiple reports, many of these in the highest impact journals. Seminal work nearly 20 years ago from Emerman and colleagues (ref 3) first highlighted HIV capsid as the key mediator, debunking at the time earlier work focused on Vpr, matrix, integrase, and “the central DNA flap”. The work in this paper and the accompanying Fu manuscript now provide the biochemical and biological basis for capsid-mediated HIV nuclear import. These papers together represent a transformative advance for the field and should be published in Nature. It is unjustified to suggest that these studies advance the field incrementally.

The revised paper admirably addressed the reviewers comments, adding important N57D mutant to Fig 4 and protein quality gels/images (new Fig S5). My remaining comments are comparatively minor.

We thank the referee for their statements of support and for their constructive feedback.

Please provide additional clarity for added S5-1. Methods are lacking for the in vitro translation technique (ref 3 is cited). The FFS method indicates 50 nM GFP-Nup, so authors can track specific GFP-Nup concentration. Why then is there so much variation in Fig S5-1 signal intensities? Most proteins >100 kDa appear under loaded, with Nup153 upper gel barely discernable. How was gel loading normalized? Ideally, this should be equal mass or molarity. Please clarify.

We have expanded the cell-free expression method description. The gels presented in Supplementary Figure 1 are not intended to be quantitative and have not been normalised. Rather, they represent the maximum possible loading from the cell-free extract in order to demonstrate sample purity. The different fluorescence levels reflect the different expression levels (large proteins generally express to lower levels, as the reviewer has noted). All samples were diluted to 50 nM for the FFS measurement, but this concentration is too low to reliably interpret by in-gel fluorescence.

Line 29, “their DNA” is technically incorrect. These are RNA viruses.

We have clarified this sentence, which now reads: ‘All retroviruses establish infection by converting their ssRNA into dsDNA and integrating it into host chromatin within the nucleus’

Line 30, "occurs during mitosis" has been directly shown for MLV. To soften the generalization, I would suggest: "...retroviruses, mitosis provides the opportunity to interact with chromosomal DNA during nuclear envelope breakdown"

We have made this suggested change.

The new ED figure, S5, is cited before the other ED figs lines 78-79. Please renumber the figures to reflect the order by which they are cited in the main text.

Figure order corrected.

Lines 135, 136 are garbled. Please rewrite.

The sentence now reads: *'To investigate how the capsid recognises FG-repeats, we examined CA mutant interactions with our identified binders (Nup42_{Full-length}, Nup58_{Full-length}, Nup62_{Full-length}, Nup98₁₋₄₉₉, Pom121₂₆₆₋₁₂₄₉ and Nup214₁₂₁₀₋₂₀₉₀).*'

Please cite ref 15 alongside ref 16 line 152.

Reference cited.

Add close parenthesis line 221.

Parentheses closed.

Please also cite ref 15 line 272.

Reference cited.